# LINOCS: Lookahead Inference of Networked Operators for Continuous Stability

**Noga Mudrik**                                                    *nmudrik1@jhu.edu*
*Biomedical Engineering, Kavli NDI, CIS*
*The Johns Hopkins University*
*Baltimore, MD, 21218.*

**Eva Yezerets**                                                    *eyezere1@jhu.edu*
*Biomedical Engineering, CIS*
*The Johns Hopkins University*
*Baltimore, MD, 21218*

**Yenho Chen**                                                      *yenho@gatech.edu*
*Department of Biomedical Engineering*
*Georgia Institute of Technology*
*Atlanta, GA 30332.*

**Christopher J. Rozell**                                           *crozell@gatech.edu*
*School of Electrical and Computer Engineering*
*Georgia Institute of Technology*
*Atlanta, GA 30332.*

**Adam S. Charles**                                                 *adamsc@jhu.edu*
*Biomedical Engineering, Kavli NDI, CIS*
*The Johns Hopkins University*
*Baltimore, MD, 21218.*

**Reviewed on OpenReview:** *https://openreview.net/forum?id=A6D3PYSyqJ*

## Abstract

Identifying hidden interactions within complex systems is key to unlocking deeper insights into their operational dynamics, including how their elements affect each other and contribute to the overall system behavior. For instance, in neuroscience, discovering neuron-to-neuron interactions is essential for understanding brain function; in ecology, recognizing interactions among populations is key to understanding complex ecosystems. Such systems, often modeled as dynamical systems, typically exhibit noisy high-dimensional and non-stationary temporal behavior that renders their identification challenging. Existing dynamical system identification methods typically yield operators that accurately capture short-term behavior but fail to predict long-term trends, suggesting an incomplete capture of the underlying process. Methods that consider extended forecasts (e.g., recurrent neural networks) lack explicit representations of element-wise interactions and require substantial training data, thereby failing to capture interpretable network operators. Here we introduce **L**ookahead-driven **I**nference of **N**etworked **O**perators for **C**ontinuous **S**tability (LINOCS), a robust learning procedure for identifying hidden dynamical interactions in noisy time-series data. LINOCS integrates several multi-step predictions with adaptive weights during training to recover dynamical operators that can yield accurate long-term predictions. We demonstrate LINOCS' ability to recover the ground truth dynamical operators underlying synthetic time-series data for multiple dynamical systems models (including linear, piecewise linear, time-changing linear systems' decomposition, and regularized linear time-varying

systems) as well as its capability to produce meaningful operators with robust reconstructions through various real-world examples.

## 1 Introduction

Uncovering the dynamics underlying high-dimensional time-series data is crucial for deciphering the fundamental principles that govern temporally evolving systems. This is apparent across significant scientific domains, including neuroscience (where neurons or ensembles interact over time, e.g., Vaadia et al. (1995); D'Aleo et al. (2019); Mudrik et al. (2024c)), immunology (where cells regulate immune responses, e.g., Savill et al. (2002)), and ecology (where understanding population interactions yields insights into ecosystem dynamics, e.g., Stein et al. (2013)). Hence, scientific research necessitates the development of procedures adept at learning dynamic operators that can accurately capture the non-linear and non-stationary evolution of real systems.

Existing approaches for dynamical systems identification, though, often rely on either "black-box" deep learning methods, which while powerful, yield uninterpretable representations, or on simple learning procedures that maximize reconstruction between consecutive samples, and thus fail to accurately predict the system's behavior for longer scales. Specifically, common dynamical system identification models regularly rely on optimizing dynamics by minimizing the prediction error for each time point based on projecting the preceding one through dynamics. Consequently, when using such procedures to learn the operators, post-learning long-term predictions of the system's values (by iteratively estimating the system's state at the next time point) usually result in undesired divergence away from the real system's values. This difficulty in long-term predictions, importantly, implies that operators recovered by these models that are based on local cost functions may not capture the underlying system correctly. The challenge in identifying such underlying operators, therefore, lies in the need to incorporate long-term predictions directly into the learning procedure, which can be especially challenging in cases where the dynamics are non-stationary, non-linear, or otherwise constrained in ways that reflect real-world system behavior.

To address this challenge, we present a learning procedure that introduces Lookahead Inference of Networked Operators for Continuous Stability (LINOCS). LINOCS bridges the gap between minimizing reconstruction costs based on single time-step projections (which typically result in operators that quickly diverge in long-term forecasts), and optimizing multi-step training that relies on reconstructions from past time points (which can lead to unstable predictions). Particularly, LINOCS achieves this by integrating adaptive re-weighted multi-step reconstructions into the dynamics inference, progressively building up the cost over training iterations while simultaneously considering several multi-step reconstruction terms to identify operators that enable stable, long-term reconstruction post-training. Through this process, LINOCS also avoids relying on massive amounts of data (like other methods, including RNNs, for example, require). We demonstrate the effectiveness and adaptability of LINOCS in a variety of dynamical systems models, including linear, switching-linear, decomposed systems, and smoothly linear time-varying systems, achieving significantly improved accuracy in operator identification and long-term predictions compared to 1-step optimization approaches (Fig. 4E, 3A).

The main advantages of LINOCS over existing approaches is its integration of adaptive weights for different reconstruction orders. This adaptive nature allows LINOCS to stably regulate noisy measurements and extract highly accurate dynamics that can predict far into the future. Moreover, while other methods primarily focus on specific dynamical system architectures (e.g., RNNs), LINOCS can be implemented into various dynamic structures, as we demonstrate through applications to switching linear systems and decomposed linear systems. Finally, LINOCS improves the identification of operators that expose pairwise element interactions, which is important for scientific interpretation.

Our contributions in this paper notably include:

- We propose LINOCS, a novel learning procedure that incorporates re-weighting multi-step predictions into the cost for operator identification.

- We demonstrate that applying LINOCS improves the ability to recognize ground-truth operators.

- We show LINOCS' efficacy across a diverse range of dynamical systems, including linear, periodically linear, linear time-varying (LTV), and decomposed linear.

- Finally, we demonstrate LINOCS' ability to work on real-world brain recordings, resulting in better long-term reconstruction compared to baselines.

## 2 Background and Terminology

Consider a system with $p$ interacting elements (e.g., neurons in the brain) whose time-changing state $\boldsymbol{X} \in \mathbb{R}^{p \times T}$ evolves over discrete time points $t = 1 \ldots T$ as $\boldsymbol{x}_{t+1} = g(\boldsymbol{x}_t, \boldsymbol{b}_t, t)$, where $\boldsymbol{x}_t \in \mathbb{R}^p$ refers to the state at time $t$, $\boldsymbol{b}_t \in \mathbb{R}^p$ represents an offset at time $t$, and $g$ is a function $g : \mathbb{R}^p \times \mathbb{R}^p \times \mathbb{Z} \to \mathbb{R}^p$. For example, in neuroscience, $\boldsymbol{x}_t$ can represent the time-evolving activity of $p$ recorded neurons during a recording session with $T$ time points, or $\boldsymbol{x}_t$ can represent the activation levels of $p$ immune cells when modeling the immune system.

In this paper, we focus on linear, piece-wise linear, and locally linear time-varying systems. Specifically, we limit our analysis to functions $g(\cdot)$ that can be written as:

$$\boldsymbol{x}_{t+1} = g(\boldsymbol{x}_t, \boldsymbol{b}_t, t) := \boldsymbol{A}_t \boldsymbol{x}_t + \boldsymbol{b}_t, \tag{1}$$

where $\boldsymbol{A}_t \in \mathbb{R}^{p \times p}$ represent the transition matrices at each time $t$. Our focus on locally linear dynamics is supported by the fact that even highly nonlinear functions can be well approximated over small time intervals using local linearization (Khalil, 2001; Sastry, 2013). Importantly, this formulation's advantage lies in its "network interpretability"—the retention of the ability to easily extract the system's pairwise interactions, including non-stationary changes in $\boldsymbol{A}_t$ and $\boldsymbol{b}_t$ over time. Specifically, any operator entry $[\boldsymbol{A}_t]_{i,j}$ for $i, j = 1 \ldots p$ in every $t = 1 \ldots T$ can be interpreted as the effect of element $j$ on element $i$ at time $t$.

In practice, however, robustly recovering operators that can accurately describe the system's evolution in non-stationary and non-linear settings, faces numerous computational challenges. Chiefly, if we adopt a naive approach to identify the operators as: $\widehat{\boldsymbol{A}}_t, \widehat{\boldsymbol{b}}_t = \arg\min_{\boldsymbol{A}_t, \boldsymbol{b}_t} \|\boldsymbol{x}_t - \boldsymbol{A}_t \boldsymbol{x}_t - \boldsymbol{b}_t\|_2^2$ for every $t = 1 \ldots T$, this problem is statistically unidentifiable. Specifically, the problem has $p^2 + p$ unknowns for each time point, but only $p$ equations.

One approach to improve inference in these settings is to introduce additional structure via a prior over $\boldsymbol{A}_t$ and $\boldsymbol{b}_t$, that constrains the solution space and is commonly grounded in application-driven assumptions. For instance, in many scientific settings, it is reasonable to assume that interactions change smoothly over time. Therefore, adding a temporal smoothness constraint on $\boldsymbol{A}_t$ and $\boldsymbol{b}_t$ (e.g., $\|\boldsymbol{A}_t - \boldsymbol{A}_{t-1}\|_F^2 < \epsilon_A$ and $\|\boldsymbol{b}_t - \boldsymbol{b}_{t-1}\|_2^2 < \epsilon_b$) can be beneficial for both interpretability and accuracy. In addition, the inclusion of such constraints can be crucial, particularly in noisy settings, to prevent overfitting. The addition of such constraints transforms the problem to:

$$\widehat{\boldsymbol{A}}_t, \widehat{\boldsymbol{b}}_t = \arg\min_{\boldsymbol{A}_t, \boldsymbol{b}_t} \|\boldsymbol{x}_{t+1} - \boldsymbol{A}_t \boldsymbol{x}_t - \boldsymbol{b}_t\|_2^2 + \mathcal{R}(\boldsymbol{A}_t, \boldsymbol{b}_t), \tag{2}$$

where $\mathcal{R}(\boldsymbol{A}_t, \boldsymbol{b}_t)$ can represent regularization on the dynamic operators.

Although these solutions may offer good short-term predictions (i.e., predicting $\boldsymbol{x}_t$ from $\boldsymbol{x}_{t-1}$) with minor errors, they often struggle to fully reconstruct the dynamics over longer time-scales due to the build-up of estimation errors over multiple iterative predictions.

The challenge of operator identification is further complicated by the practical issue that we can typically only observe noisy measurements of $\boldsymbol{x}_t$:

$$\widetilde{\boldsymbol{x}}_t = \boldsymbol{x}_t + \boldsymbol{\epsilon}_t \text{ where } \boldsymbol{\epsilon} \in \mathbb{R}^{p \times T} \text{ represents some noise, e.g.,} \boldsymbol{\epsilon}_t \sim \mathcal{N}(0, \sigma^2).$$

Particularly, if $\widehat{\boldsymbol{A}}_t$ is obtained from $\arg\min_{\boldsymbol{A}_t} \|\boldsymbol{x}_{t+1} - \boldsymbol{A}_t \widehat{\boldsymbol{x}}_t\|_2^2$, the distances between the real and the estimated operators $\{\|\boldsymbol{A}_t - \widehat{\boldsymbol{A}}_t\|_F^2\}_{t=1}^T$ increase with the noise $\boldsymbol{\epsilon}_t$—further hindering operator recovery. For instance, given $\epsilon_t = \epsilon \sim \mathcal{N}(0, \sigma^2)$ *i.i.d* Gaussian noise, i.e., $\widetilde{\boldsymbol{x}}_t | \widetilde{\boldsymbol{x}}_{t-1}$ has a covariance of

$\sigma^2 \boldsymbol{I}$, then $\widetilde{\boldsymbol{x}}_{t+1} = \boldsymbol{A}\widetilde{\boldsymbol{x}}_t + \epsilon = \boldsymbol{A}(\boldsymbol{A}\widetilde{\boldsymbol{x}}_{t-1} + \epsilon) + \epsilon = \boldsymbol{A}^2\widetilde{\boldsymbol{x}}_{t-1} + \boldsymbol{A}\epsilon + \epsilon$, meaning the new noise term is $\boldsymbol{A}\epsilon + \epsilon \sim \mathcal{N}(0, \sigma^2(\boldsymbol{A}\boldsymbol{A}^T + I))$. For higher orders, the accumulated noise variance over $K$ steps will result in: $\sum_{k=0}^{K-1} \sigma^2 \boldsymbol{A}^k (\boldsymbol{A}^T)^k$, which can quickly increase based on the eigenspectra of $\boldsymbol{A}$.

As the accuracy of the fit of a dynamical system to data often needs to be evaluated based on its ability to predict future values accurately (Tabar, 2019), an inability to capture long-term predictions suggests that the learned operators demonstrate limited capacity to fully describe the system.

In this context, we define below three types of prediction styles that will be used to evaluate and compare methods throughout this paper:

- **1-Step Prediction $(\boldsymbol{x}_{t+1}|\boldsymbol{x}_t)$:** 1-Step prediction involves using the state at each time point $t$ to estimate the state at the next time point $(t + 1)$.

- **Iterative Multi-Step Prediction (IMS) of Order $K \in \mathbb{N}$ $(\boldsymbol{x}_{t+k}|\boldsymbol{x}_t)$:** IMS involves iteratively, for $K$ times, forecasting 1-step ahead values and using these forecasts as inputs for further 1-step ahead forecasts. Namely, predicting $\widehat{\boldsymbol{x}}_{t+k}|\widehat{\boldsymbol{x}}_{t+k-1}$ $\forall k = 0 \ldots K - 1$, where $\widehat{\boldsymbol{x}}_{t-1} := \widetilde{\boldsymbol{x}}_{t-1}$. Here, we will notate an IMS prediction of order $K$ by $\widehat{\boldsymbol{x}}_t^K$. We chose to name this prediction style "IMS" as to be consistent with the literature (Chevillon, 2007).

- **Full Lookahead Prediction $(\boldsymbol{x}_k|\boldsymbol{x}_0)$:** This method enhances IMS by forecasting the state at each time point $\boldsymbol{x}_t$ starting from the initial observations $(\widetilde{\boldsymbol{x}}_0)$. It achieves this by sequentially applying transition matrices to the estimation from the previous time point $\widehat{\boldsymbol{x}}_t|\widehat{\boldsymbol{x}}_{t-1}$, starting from $\widetilde{\boldsymbol{x}}_0$, resulting in: $\widehat{\boldsymbol{x}}_t = \widehat{\boldsymbol{A}}_{t-1} \ldots \widehat{\boldsymbol{A}}_0 \widetilde{\boldsymbol{x}}_0$ $\forall t = 1 \ldots T$.
  (Note: the formula above is presented without offsets $\{\boldsymbol{b}_t\}_{t=0}^T$ for simplicity, though they may be included).

Importantly, IMS Prediction and Full Lookahead Prediction typically exhibit instability due to the accumulation of errors in the sequential reconstructing process (Fig. 1A).

## 3 Prior relevant approaches:

Theoretical literature on long-term prediction instability traces back to Cox (1961) and Klein (2019), who, respectively, introduced exponential smoothing and direct estimation of distant future states. Subsequent studies, including Findley (1983; 1985); Weiss (1991); Tiao and Xu (1993); Lin and Granger (1994); Kang (2003), evaluated the effectiveness of dynamical system identification methods in producing long-term predictions. These approaches, however, build on either "1-step training", which focuses on identifying dynamical operators by minimizing the reconstruction error when projecting the state from one time point to the next, or on "direct forecasting", which aims to identify a mapping function $\boldsymbol{F}_{kt} : \mathbb{R}^p \to \mathbb{R}^p$ that predicts future states by $\boldsymbol{x}_{t+k} = \boldsymbol{F}_{kt}(\boldsymbol{x}_t)$, bypassing explicit identification of intermediate dynamical operators.

While the "direct estimation" approach naturally results in more stable long-term predictions compared to 1-step optimization, such approach fails to provide an interpretable "network" meaning to the operators, i.e., the ability to interpret each entry $(i, j)$ of the operator $(\boldsymbol{A}_{t,[i,j]})$ as the effect of element $j$ on element $i$ at time $t$. This is important e.g., in neuroscience, where understanding the brain's interactions entails discerning the time-changing fast interactions of neurons (Sussillo, 2014), or in epidemiology where tracking disease spread dynamics matters (Aguiar et al., 2020). In addition, when Marcellino et al. (2006) compared between iterated and direct estimates using macroeconomic data, they found that in contrast to previous assumptions, iterated forecast methods outperform direct forecast methods, especially when models can auto-select long-lag specifications—raising questions about which approach is more suitable for learning dynamical operators.

Markov and Hidden Markov Models (Florian et al., 2011; Ou et al., 2013; Bilmes, 2006) are widely used to model time series data and capture temporal dependencies in dynamical systems. However, HMMs struggle with long-term predictions due to their reliance on the Markovian assumption, and require extensions

for modeling non-stationary systems (e.g. Sin and Kim (1995)). Other models, including low-rank auto-aggressors (Harris et al., 2021; Basu et al., 2019) and other low-dimensional linear models (e.g.,DMD and its variants Tu (2013); Askham and Kutz (2018); Sashidhar and Kutz (2022); Ferré et al. (2023); Kutz et al. (2016) are widely used for capturing and predicting complex dynamical systems in various fields such as fluid dynamics, neuroscience, and financial modeling. However, they do not specifically address the issue of long-term divergence.

More recently, Venkatraman et al. (2015) proposed a general approach called DAD that reuses training data to build a no-regret learner with multi-step prediction that includes "fixing" and updating the model itself based on every step within the multi-step prediction. However, the authors presented it as a general, nonspecific, approach to consider without specific implementation details. More importantly, DAD does not discuss the possibility for including priors over the operators (e.g., temporal smoothness between consecutive operators) during the training, nor did they consider the need to find operators that are not only expressive but also interpretable.

Other approaches are based on multiple shooting, include, e.g. Jordana et al. (2021), who proposed proposed learning dynamics with multiple shooting and multi-step training under each sub-trajectory (i.e., between shooting nodes). However, their method requires matching boundary conditions as they start only from the shooting nodes, which can be prone to noise sensitivity. Additionally, they give similar weights to different multi-step orders, which may be hard to tune if some orders present high errors at different periods during training.

Generalized Linear Model (GLM) with multi-step inference were proposed in the context of learning neural spikes from spiking data (Hocker and Park, 2017). Their Poisson GLM model maximizes a log-likelihood cost that incorporate the spiking multiple steps ahead in the future, thus addressing runaway self-excitation of neuron activity. However, their idea is specific to the Poisson GLM model and is not designed for non-stationary systems, thus limiting its applicability under varying applications.

Full forward and/or backward passes through e.g., Backpropagation Through Time (BPTT) as in Recurrent Neural Networks (RNNs), can partially handle long-term prediction instability. Nevertheless, approaches relying on this process may prone to the vanishing or exploding gradient. More advanced RNN models that incorporate future steps integration to improve RNN forecasting include (Hess et al., 2023), that addresses the vanishing/exploding gradients through a modification in RNN training to promote bounded gradients in training on chaotic systems and in piecewise linear RNNs. Pal et al. (2021), on the other hand, proposed using particle flow to approximate the posterior distribution of future RNN states using a spatial graph of element similarities. Other RNN models, including Unni et al. (2023); Xiao et al. (2023); Yeung et al. (2019), leverage the Koopman operator for improved long-term prediction, yet, while powerful, they yield operators that cannot be directly interpreted as pairwise interactions between our elements of interest. Moreover, all above RNN models are currently limited by their architecture and do not support handling more nuanced dynamics (e.g., decomposed systems as in Mudrik et al. (2024a)), or effectively capturing smoothly-changing non-stationary behaviors. Another limitation of these methods is that they do not incorporate constraints on the operators (e.g., sparsity), which can be important for enhancing the operators' interpretability.

Hafner et al. (2019) proposed a model of reinforcement learning with future steps learning that learns the environment dynamics and chooses actions through online planning in a latent space. For the dynamics modeling, they use both stochastic and deterministic components based on a generalized variational objective that encourages multi-step predictions. However, their latent model is action-based and does not explicitly limit the dynamics to locally linear functions, potentially complicating the understanding of evolution in the latent space. Additionally, they use gradient descent to learn the parameters, which may be prone to convergence local minima.

An additional approach to understanding dynamical systems involves identifying a sparse set of functions that jointly decompose the observations. For example, SINDy (Sparse Identification of Nonlinear Dynamics, Brunton et al. (2016)) employs a data-driven approach to discover governing fundamental equations from data using sparse regression. Although SINDy and its extensions (e.g., Kaheman et al. (2020)) are promising for discovering governing data equations, such representation does not provide explicit insight into the time-changing *interactions* between the state elements.

# 4 Specific models considered in this work:

Of particular interest in this work is improving the model fit of a core set of linear dynamical systems with different temporal constraints on the system evolution, including 1) Time-Invariant Linear Dynamical Systems (LDS), 2) Switching Linear Dynamical Systems (SLDS) (Ackerson and Fu, 1970; Bar-Shalom and Li, 1990; Hamilton, 1990; Ghahramani and Hinton, 1996; Murphy, 1998; Fox et al., 2008; Linderman et al., 2017), 3) decomposed Linear Dynamical Systems (dLDS, Mudrik et al. (2024a); Chen et al. (2024); Mudrik et al. (2024b)), and 4) regularized Linear Time-Varying (LTV) Dynamical Systems.

**Time-Invariant Linear Dynamical Systems (LDS).** In linear systems analysis, the evolution of a general state $\boldsymbol{X}$ over $T + 1$ time points can be typically represented as $\boldsymbol{x}_{t+1} = \boldsymbol{A}\boldsymbol{x}_t + \boldsymbol{b}$, where $\boldsymbol{A} \in \mathbb{R}^{p \times p}$ is the time-invariant dynamics matrix and $\boldsymbol{b} \in \mathbb{R}^{p \times 1}$ remain constant over time. One common method for determining $\boldsymbol{A}$ (and $\boldsymbol{b}$ if it is assumed that an unknown offset exists) involves a 1-step optimization approach that includes applying least squares across all time points. This entails solving

$$\widehat{\boldsymbol{A}}, \widehat{\boldsymbol{b}} = \arg\min_{\boldsymbol{A},\boldsymbol{b}} \|\widetilde{\boldsymbol{X}}_{[:,1:T]} - \boldsymbol{A}\widetilde{\boldsymbol{X}}_{[:,0:T-1]} - [1]_{1 \times T} \otimes \boldsymbol{b}\|_F^2, \tag{3}$$

where $\widetilde{\boldsymbol{X}}_{[:,1:T]}$ and $\widetilde{\boldsymbol{X}}_{[:,0:T-1]}$ represents the noisy observations of the state from the second time point ($t = 1$) up to the last time point ($T$) and from the first time point ($t = 0$) up to $T - 1$, respectively. $[1]_{1 \times T} \otimes \boldsymbol{b}$ represents the horizontal concatenation of the column vector $\boldsymbol{b}$, for $T$ times. Here, $\boldsymbol{A}$ captures the average influence from $\boldsymbol{x}_{t-1}$ to $\boldsymbol{x}_t$ for all $t = 1 \ldots T$.

While such linear (time-invariant) systems are simple and therefore interpretable in terms of capturing element interactions—referred to here as "network" interpretability—their overly simplistic nature often prevents them from adequately representing the complexities of real-world time series.

**Switching Linear Dynamical Systems (SLDS).** SLDS models (Ackerson and Fu, 1970; Bar-Shalom and Li, 1990; Hamilton, 1990; Ghahramani and Hinton, 1996; Murphy, 1998; Fox et al., 2008; Linderman et al., 2017) along with other piece-wise stationary models (e.g., Song et al. (2021)) aim to provide interpretable representations of dynamics by identifying operators that govern periods of stationary behavior, with the system transitioning between these operators over time. Variations of SLDS include, e.g., recurrent SLDS (rSLDS), which introduces an additional dependency between discrete switches and the previous state's location in space (Linderman et al., 2017); and tree-structured recurrent SLDS, which extends rSLDS by incorporating a generalized stick-breaking procedure (Nassar et al., 2018).

While SLDS models usually involve transitioning from an observed to a latent low-dimensional space, here we chose to focus on the case where switches occur within the observation space, essentially enforcing the transition to the latent space to be the identity operator. If we denote $\widetilde{\boldsymbol{X}} \in \mathbb{R}^{N \times T}$ as the noisy observations subjected to *i.i.d* Gaussian noise, SLDS models the evolution of $\widetilde{\boldsymbol{x}}_t$ using a set of $J$ discrete states ($j = 1 \ldots J$), where each state $j$ is associated with its own linear dynamical system $\boldsymbol{f}_j$. These discrete states switch between them abruptly at certain time points following an HMM model. During each inter-switch period, if the system is in the $j$-th discrete state, SLDS models the evolution of the state linearly as $\boldsymbol{x}_t = \boldsymbol{f}_j\boldsymbol{x}_{t-1} + \boldsymbol{b}_j$, where $\boldsymbol{f}_j$ represents the linear transition matrix for the $j$-th discrete state and $\boldsymbol{b}_j$ denotes a constant offset term for that discrete state. SLDS can be trained by an alternating set of steps between dynamic learning and the HMM update of the operators. This way, SLDS tackles the crucial task of capturing non-stationarities while preserving interpretability; however it inherently lacks the capability to distinguish between multiple co-occurring processes or overlapping subsystems. More information about the model assumptions, limitations, and parameter selection can be found in (Linderman et al., 2016; 2017).

**decomposed Linear Dynamical Systems (dLDS).** The Decomposed Linear Dynamical Systems (dLDS, Mudrik et al. (2024a)) model relaxed the time-invariant or piece-wise linear limitation of LDSs and SLDSs to support the discovery of co-occurring processes while maintaining interpretability. To emphasize, here, for simplicity, we focus on the case where the dynamics evolution is described directly in the observation space, while the full model presented in (Mudrik et al., 2024a) supports learning the dynamics within an identified latent state. Specifically, dLDS models the dynamics evolution ($\boldsymbol{A}_t\widetilde{\boldsymbol{x}}_{t-1} \rightarrow \widetilde{\boldsymbol{x}}_t$) using a sparse time-changing decomposition of linear dynamical operators such that $\boldsymbol{A}_t = \left(\Sigma_{j=1}^J \boldsymbol{f}_j c_{jt}\right)$, resulting in $\left(\Sigma_{j=1}^J \boldsymbol{f}_j c_{jt}\right) \widetilde{\boldsymbol{x}}_{t-1} \rightarrow \widetilde{\boldsymbol{x}}_t$. These dynamical operators ($\{\boldsymbol{f}_j\}_{j=1}^J$) are global, i.e., not time dependent, and hence

are interpretable globally. However, their time-changing weights ($c_t$) enable modeling non-stationary and non-linear dynamics (Fig. 2 right). Notably, dLDS is trained through an Expectation-Maximization (EM) procedure where the global dynamic operators $\{f_j\}_{j=1}^J$ and their time-changing coefficients $\{\{c_{jt}\}_{t=1}^T\}_{j=1}^J$ are updated iteratively to maximize the posteriors as:

$$\{\{\widehat{c_{jt}}\}_{t=1}^T\}_{j=1}^J = \arg\max_{\{c_{jt}\}} P(\{c_{jt}\}_{j=1,t=1}^{J,T} | \widetilde{X}, \{f_j\}) \tag{4}$$

$$\{\widehat{f}_j\}_{j=1}^J = \arg\max_{\{f_j\}} P(\{f_j\} | \widetilde{X}, \{c_{jt}\}_{j=1,t=1}^{J,T}). \tag{5}$$

Interestingly, dLDS can also capture linear or switching behaviors described earlier, by fixing the dLDS coefficients over time (for linear behavior, Fig. 2 far left) and by allowing abrupt change of coefficients in specific time points (for switching behaviors, Fig. 2 middle left). More information about the model assumptions, limitations, and parameter selection can be found in (Mudrik et al., 2024a).

While dLDS presents several dynamic modeling advantages (e.g., captures non-stationarities, promotes interpretability), as it estimates the parameters for each time $t$ solely based on the values of the preceding state at time $t-1$, it does not address the issue of inaccurate long-term predictions' divergence.

**Smooth or Sparse Linear Time-Varying Systems (LTV).** In this paper, we refer to LTV systems that can be described by: $x_{t+1} = A_t x_t$ for all $t = 1 \ldots T$. We further assume that a regularization $\mathcal{R}(A_t)$ may be applied to the operators $\{A_t\}_{t=1}^T$. This regularization can be inspired by the application (e.g., smoothness of operators over time, $\|A_t - A_{t-1}\|_F^2 < \epsilon_2$ or operator sparsity $\|\text{vec}(A_t)\|_0 < \epsilon_1$) and can mitigate the ill-posed nature of finding $A_t$ separately for each time point.

## 5 LINOCS

In LINOCS we aim to learn the unknown dynamic operators $\{\widehat{A}_t\}_{t=1}^T$ by integrating several multi-step predictions simultaneously into the inference procedure. This approach yields not only a more accurate full-lookahead post-learning reconstruction but also operators that are more closely aligned with the ground truth. Particularly, for every $t = 1 \ldots T$, LINOCS finds the most likely estimate of $\{A_t, b_t\}$ given $K+1$ ($K \in \mathbb{Z}_{\geq 0}$ hyper-parameter) multi-step reconstructions of orders $k = 0 \ldots K$ with different weights $\{w_k\}_{k=0}^K$:

$$\widehat{A}_t = \arg\min_{A_t} \sum_{k=0}^K w_k \|\widetilde{x}_{t+1} - A_t \widehat{x}_t^k\|_F^2, \tag{6}$$

where $\widetilde{x}_{t+1}$ are the observations at time $t+1$ and $\widehat{x}_t^k$ is the $k$-order lookahead estimation, defined by the recursive rule for predicting the state at time $t$, starting from the observations at $t-k$. Namely,

$$\widehat{x}_t^k = \widehat{A}_{t-1}\widehat{A}_{t-2}\ldots\widehat{A}_{t-k}\widetilde{x}_{t-k}. \tag{7}$$

The weights $\{w_k\}_{k=0}^K$ associated with the orders $k = 0 \ldots K$ are dynamically adjusted throughout the inference process (Fig. 1B). This adjustment considers both the order number ($k$) and the current reconstruction error related to that order, $e_k$ (e.g., the $\ell_2$ norm, $e_k = \|\widetilde{x}_t - \widehat{x}_t^k\|_2^2$). Unlike other multi-step methods (e.g., Venkatraman et al. (2015)), LINOCS adapts the weights of the reconstruction orders to prioritize the minimization of large errors in lower orders before considering higher orders (Fig. 1B). Specifically, it gradually increases the weight of the best lookahead reconstruction until convergence conditions are satisfied. In our implementations, the weights can be chosen from a list of built-in choices such as uniform, linearly decreasing, and exponentially decreasing weights. Additionally, our framework allows custom weight functions that suit their specific needs. In the experiments presented in this paper, we focus on showcasing three specific options for the weights:

- Adapting the weights to sequentially introduce higher multi-step reconstruction orders once the error for each preceding order falls below a designated threshold, while continuing to maintain the activation of lower orders. Specifically, in the initial iterations, only $w_0 > 0$, with all other weights

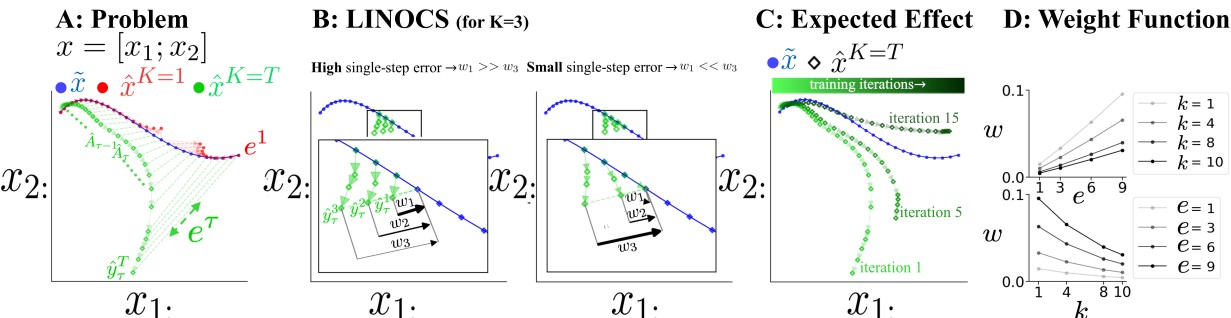

Figure 1: **Schematic of the problem and approach. A:** Models that perform well on 1-step prediction (i.e., prediction order 0, in red), commonly fail in higher orders reconstruction (e.g., order-$T$, in green). **B:** LINOCS (e.g., for training order $K = 3$) integrates weighted multi-step reconstructions for all $k = 0 \ldots K$ orders. It adapts the weights of these reconstruction orders to prioritize minimizing large errors at lower orders before addressing higher orders. The system gradually increases the weight of the most effective lookahead reconstruction until convergence conditions are met. **C:** LINOCS improves long-term reconstruction during training iterations, such that the predictions during training are expected to increasingly align with the real system. Green color indicates full-lookahead predictions across different iterations. **D:** Weights of different lookahead training orders ($k$). $w_k : \mathbb{R}^2 \to \mathbb{R}$ is a function of the order $k$ and the $k$-th order reconstruction error $e$. Top: Illustration of an exemplary effect of $e$ on $w$ when fixing $k$. Bottom: Illustration of an exemplary effect of $k$ on $w$ when fixing $e$.

$w_j = 0$ for $j \in [1, K]$. As the error of each step's reconstruction falls below this threshold, the subsequent weight $w_{j+1}$ is activated. For instance, if $w_0$ is the last activated weight and the error for the first (1-step) reconstruction falls below the threshold, $w_1$ becomes active ($w_1 > 0$), and this sequence of activation continues for higher-order weights as each subsequent step achieves the required accuracy.

- Constant weights over iterations with an exponential decay over $k$, defined as $w_k = \exp^{-\sigma k}$ for some scalar $\sigma \in \mathbb{R}_{>0}$.

- A weight function that considers both $k$ and $e_k$, exhibiting a monotonic decrease in $k$ and an increase in $e$, with $k$ decreasing faster than $e$ increases (Fig. 1 D).

Importantly, throughout this paper, we distinguish between two concepts: training order and prediction order. We denote "training order" ($K_{\text{train}}$) as the maximum order considered *during* LINOCS training. In line with this, "$K$-step optimization" or "$K$-step training" specifically refer to the use of the $K$-step cost (i.e. estimating $\boldsymbol{x}_t | \{\boldsymbol{x}_{t-k}\}_{k=1}^K$) *during* training, with "1-step optimization/training" being a special case in which the training considers only consecutive time points for inference.

In contrast, prediction order refers to *post-training* predictions that involve iteratively propagating the identified operators for $K_{\text{pred}}$ steps into the future.

Here, we demonstrate the contribution of LINOCS for accurate long-scale predictions in four types of systems: 1) time-invariant linear; 2) switching linear; 3) decomposed linear; and 4) LTV systems. Importantly, in our experiments, we assume that we observe the underlying system under additive *i.i.d* Gaussian noise conditions, however LINOCS can be easily adjusted to other noise statistics.

## 5.1 LINOCS for Linear Dynamics

We first present the LINOCS learning rule for the simplest case of time-invariant linear dynamical systems (Fig. 2 leftmost subplot). Let $\widetilde{\boldsymbol{X}} \in \mathbb{R}^{p \times T}$ be the observations of state $\boldsymbol{X}$, such that $\widetilde{\boldsymbol{X}} = \boldsymbol{X} + \eta$, with $\eta$ being an *i.i.d* Gaussian noise ($\eta \sim \mathcal{N}(0, \sigma^2)$). In this time-invariant linear case, $\boldsymbol{X}$ evolves linearly as $\boldsymbol{x}_{t+1} = \boldsymbol{A}\boldsymbol{x}_t + \boldsymbol{b}$ for all $t = 1 \ldots T$, where $\boldsymbol{b} \in \mathbb{R}^{p \times 1}$ is a constant offset.

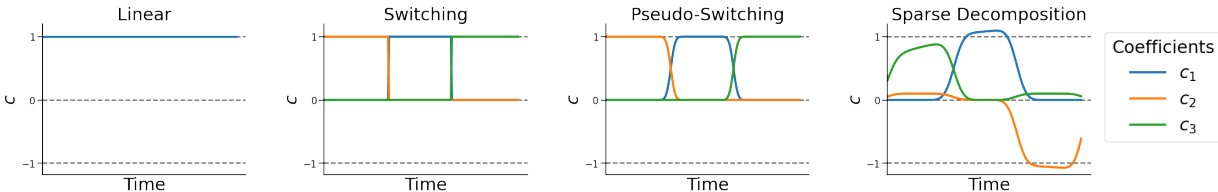

Figure 2: **Example time-varying LDS systems A:** The baseline linear time-invariant dynamical system will present dynamical operator constant over time. **B:** Switching linear dynamical systems (SLDS) jump between different linear operators that are time-invariant between jumps. **C:** "Pseudo-Switching" dynamics is similar to SLDS with the inclusion of smoother transitions between periods of constant linear dynamics. **D:** The decomposed linear dynamical systems (dLDS) model is a generalization of SLDS to sparse time-changing linear combinations of linear operators. dLDS can also model negative coefficients.

In this case, LINOCS estimates $\boldsymbol{A}$ and $\boldsymbol{b}$ by

$$\widehat{\boldsymbol{A}}, \widehat{\boldsymbol{b}} = \arg\min_{\boldsymbol{A},\boldsymbol{b}} \sum_{k=0}^{K} \boldsymbol{w}_k \|\widetilde{\boldsymbol{x}}_{t+1} - \boldsymbol{A}^{k+1}\widetilde{\boldsymbol{x}}_{t-k} - \sum_{j=0}^{k} \boldsymbol{A}^j \boldsymbol{b}\|_2^2, \tag{8}$$

where $\boldsymbol{A}^{k+1}$ is taking the transition matrix $\boldsymbol{A}$ to the power of $k+1$, $K$ is an hyperpameter that dictates the maximum reconstruction order, and the set $\{\boldsymbol{w}_k\}_{k=0}^{K}$ can be either predefined or automatically adapted over training based on each order error. Please refer to Algorithm 1 and Appendix A.5 for further details on the inference of the operator and offset for the linear case.

---

**Algorithm 1** Linear-LINOCS

---

**Inputs:** Observations $\widetilde{X}$ and maximum order $K$.
**Build** $K$ Multi-step reconstructions to get the set of $\{\boldsymbol{\psi}_k\}$                    $\triangleright$ as described in Appendix A.5.
**Infer** Transition $\boldsymbol{A}$                    $\triangleright$ via Eq. (8) as described in Sec. A.5
**Infer** Offset $\boldsymbol{b}$                    $\triangleright$ via Eq. (16)

---

### 5.2   LINOCS for Switching Linear Dynamical Systems (SLDS)

For SLDS (Fig. 2, middle-left), we integrate LINOCS into the SLDS operator inference stage (to infer $\{\boldsymbol{f}\}_{j=1}^{J}$, $\{\boldsymbol{b}\}_{j=1}^{J}$, see Sec. 4) using the SSM framework proposed by Linderman et al. (2020), which is the current framework for running SLDS/rSLDS as described in e.g. Linderman et al. (2016; 2017). We maintain the existing SLDS approach to estimating switch times that delineate the boundaries of the linear periods between switches (i.e., we kept the switching times inference step as implemented by Linderman et al. (2020)). To recover the operators within these identified linear periods, we integrate the learning rule for the time-invariant linear case described above (Sec. 5.1). Please refer to Algorithm 2 for the procedural steps.

---

**Algorithm 2** SLDS-LINOCS

---

**Input:** Observations $\widetilde{\boldsymbol{X}}$, maximum order $K \in \mathbb{R}^+$, number of iterations.                    $\triangleright$ Initial parameters
**Initialize:** $\{\boldsymbol{f}\}_{j=1}^{J} \sim$ Uniform                    $\triangleright$ Start with a uniform distribution
**for** each iteration until a defined number of iterations **do**
       **find** switching time                    $\triangleright$ Based on current SSM framework Linderman et al. (2020)
       **infer** operator in each period                    $\triangleright$ Use linear inference as described in Alg. 1
**end for**

---

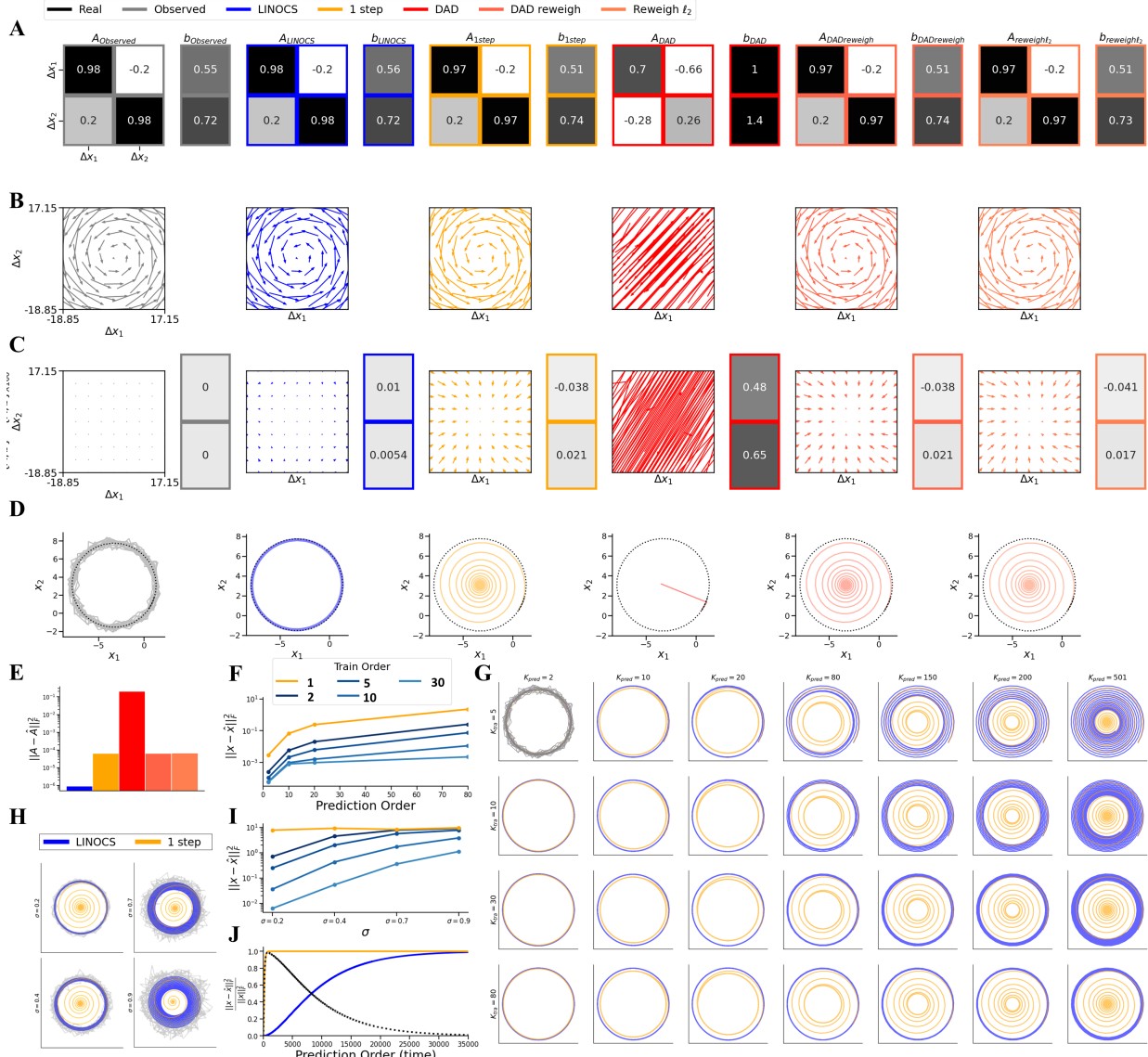

Figure 3: **Linear System Experiment. A:** Real vs. identified operators and offsets. **B:** Quiver plots of real and identified operators. **C:** Highlighted differences in effects between real operators and inferred operators showing how small differences in dynamic operators gain prominence during lookahead reconstruction (calculation details in Section A.1). **D:** Full lookahead reconstructions (ground truth vs. baselines) show swift convergence to the origin for the 1-step optimization (yellow) and divergence for DAD-based results (three most right subplots). **E:** Frobenius norm of the differences between the ground truth operators ($A$) and the identified operators ($\widehat{A}$). **F:** MSE under increasing prediction orders. For all orders, LINOCS achieves better (lower) MSE compared to 1-step optimization. **G:** Full lookahead post-training predictions using operators identified by 1-step optimization (yellow) vs. the predictions using those identified by LINOCS (blue) under various training orders (rows) and prediction orders (columns). **H, I:** LINOCS reconstruction compared to 1-step optimization under increasing noise levels ($\sigma$) demonstrates its robustness. **J:** Propagating the identified operators until reaching a relative reconstruction error of $\sim 1$. LINOCS identifies operators that can accurately predict $\sim 35,000$ time points ($\sim 1380$ full rotations), much higher than 1-step training that decay immediately.

### 5.3 LINOCS for decomposed Linear Dynamical Systems (dLDS)

For dLDS (Fig. 2, two rightmost subplots), as in SLDS, we incorporate LINOCS into the dynamical systems update step. Let $\widehat{x}_{t+1}^k$ denote the $k$-th order reconstruction of $x_{t+1}$, calculated by iteratively propagating the dLDS reconstruction $k+1$ times, starting from $x_{t-k}$. Furthermore, let $x_{t+1} \approx \Sigma_{j=1}^J \widehat{c}_{jt} \widehat{f}_j x_t$, where $\widehat{c}_t$ represents the current estimate of the dLDS coefficients and $\{\widehat{f}_j\}_{j=1}^J$ denotes the current estimate of the basis operators. We can now write the $k$-th order reconstruction $(\widehat{x}_{t+1}^k)$ as

$$\left(\sum_{j=1}^J \widehat{c}_{jt} \widehat{f}_j\right) \left(\sum_{j=1}^J \widehat{c}_{j(t-1)} \widehat{f}_j\right) \cdots \left(\sum_{j=1}^J \widehat{c}_{j(t-k)} \widehat{f}_j\right) \widetilde{x}_{t-k} \to \widehat{x}_{t+1}^k,$$

as follows with the recursive update rule

$$(\sum_{j=1}^J \widehat{c}_{jt} \widehat{f}_j) \widehat{x}_t^{k-1} \to \widehat{x}_{t+1}^k, \text{ where } \widehat{x}_t^0 := \widetilde{x}_t. \tag{9}$$

To effectively integrate LINOCS into dLDS, we incorporate multi-step predictions into the training procedure of dLDS itself. Specifically, in each iteration, we start with the update of the dynamics coefficients $c_t$. For this, we first define $F_{x_t^k} \in \mathbb{R}^{p \times J}$ as the horizontal concatenation

$$F_{x_t^k} := [f_1 \widehat{x}_t^k, f_2 \widehat{x}_t^k, \ldots, f_J \widehat{x}_t^k] \text{ for some } k \in 0 \ldots K.$$

Next, we define a new matrix $\widetilde{F}_{x_t}^K$ that extends $F_{x_t}$ to all $K+1$ reconstructions stacked vertically, resulting in $\widetilde{F}_{x_t}^K \in \mathbb{R}^{(K+1)p \times J}$:

$$\widetilde{F}_{x_t}^K = \begin{bmatrix} w_0 F_{x_t} \\ w_1 F_{x_t^1} \\ \vdots \\ w_{K-1} F_{x_t^K} \end{bmatrix}, \tag{10}$$

where $w_k$ is the weight of the $k$-th multi-step order. This matrix can then be used to infer the coefficients $(c_t)$ while simultaneously considering different reconstruction orders with varying weights.

To mirror this concatenated matrix of dynamics with the observations $(\widetilde{x}_{t+1})$, we further define a matching concatenated state vector $(\widetilde{x}_{t+1})_{vert} \in \mathbb{R}^{p(K+1) \times 1}$. This vector $(\widetilde{x}_{t+1})_{vert}$ is obtained by vertically stacking $K+1$ times the observations $\widetilde{x}_{t+1} \in \mathbb{R}^{p \times 1}$ at time $t+1$ weighted by their corresponding $w_k$ values, resulting in $(\widetilde{x}_{t+1})_{vert} = [w_0 \widetilde{x}_{t+1}; w_1 \widetilde{x}_{t+1}; \cdots; w_K \widetilde{x}_{t+1}])$. In simplified terms, $(\widetilde{x}_{t+1})_{vert} := w \otimes \widetilde{x}_{t+1}$ where $w = [w_0; w_1; \cdots; w_K] \in \mathbb{R}^{(K+1) \times 1}$ are the training orders' currents weights, and $\otimes$ denotes the Kronecker product.

The coefficients, $c_t$, are thus updated in every iteration by minimizing the squared $\ell_2$ norm

$$\widehat{c}_t = \arg\min_{c_t} \|(\widetilde{x}_{t+1})_{vert} - \widetilde{F}_{x_t}^K c_t\|_2^2 + \lambda_c \|c_t\|_1. \tag{11}$$

where $\lambda_c$ is the weight of the $\ell_1$ sparsity-promoting regularization on the coefficients.

Note that the multiplication in the first term $(\widetilde{F}_{x_t}^K c_t \in \mathbb{R}^{(K+1)p \times 1})$ produces a vector of estimates of the observations at $t+1$ $(\widehat{x_{t+1}})$ computed from all different $K+1$ past states. This way, the estimator in Equation (11) seeks the $c_t$ vector that best predicts $x_{t+1}$ considering all $K+1$ lookaheads.

The next step within every iteration includes updating the dynamic operators $\{f_j\}_{j=1}^J$. One additional modification we make (compared to the original learning of dLDS as presented by Mudrik et al. (2024a)) is that rather than updating each $f_j$ using gradient descent, we infer the dLDS' basis dynamics operators $\{f_j\}_{j=1}^J$ by fully and directly minimizing the cost. Specifically, let

$$F_{all} := [f_1, f_2, \ldots, f_J] \in \mathbb{R}^{p \times pJ},$$

be the concatenated matrix of all $\{\boldsymbol{f}_j\}_{j=1}^J$.

Also, let $(\boldsymbol{x}_c)_t := ([1]_{J\times 1} \otimes \boldsymbol{x}_t) \circ (\boldsymbol{c}_t \otimes [1]_{p\times 1}) \in \mathbb{R}^{pJ\times 1}$, where $\otimes$ denoted the Kronecker product and $\circ$ denotes element-wise multiplication, and let $\boldsymbol{X}_c \in \mathbb{R}^{pJ\times T}$ be the horizontal concatenation of all $\{(\boldsymbol{x}_c)_t\}_{t=0}^{T-1}$.

With these definitions, the dLDS operators $\{\boldsymbol{f}_j\}_{j=1}^J$ are updated by

$$\widehat{\boldsymbol{F}}_{all} = \arg\min_{\boldsymbol{F}_{all}} \|\widetilde{\boldsymbol{X}}_{:,2:} - \boldsymbol{F}_{all}(\boldsymbol{X}_c)\|_F^2, \tag{12}$$

with each $\{\boldsymbol{f}_j\}_{j=1}^J$ then being extracted from $\widehat{\boldsymbol{F}}_{all}$ and normalized to a Frobenius norm of 1. Please see Algorithm 3 for the procedural steps.

---

**Algorithm 3** dLDS-LINOCS

---

**Input:** Observations $\widetilde{\boldsymbol{X}}$, maximum order $K \in \mathbb{R}^+$
**Initialize:** $\boldsymbol{C} \in \mathbb{R}^{J\times T}$ and $\{\boldsymbol{f}\}_{j=1}^J \sim$ Uniform
$\boldsymbol{f}_j \leftarrow \frac{\boldsymbol{f}_j}{\max(\lambda(\boldsymbol{f}_j))} \forall j = 1, \dots, J$         $\triangleright$ Normalizing each $\boldsymbol{f}_j$ to unit spectral norm
**while** not converged **do**
  Compute $k$ lookaheads of $\widetilde{\boldsymbol{x}}$ using current estimates of $\{\boldsymbol{c}_t\}_{t=1}^T$ and $\{\boldsymbol{f}\}_{j=1}^J$.    $\triangleright$ via Eq. 9
  Construct $\widetilde{\boldsymbol{F}}_{x_t}^K$.                   $\triangleright$ via Eq. 10
  Update $c$ using the LASSO method.           $\triangleright$ via Eq. 11
  Update $\boldsymbol{F}_{all}$.                  $\triangleright$ via Eq. 12
  Split $\boldsymbol{F}_{all} \in \mathbb{R}^{p\times pJ}$ into $\{\boldsymbol{f}_j\}_{j=1}^J$ (each $\boldsymbol{f}_j \in \mathbb{R}^{p\times p}$).
**end while**

---

### 5.4 LINOCS for regularized Linear Time-Varying Systems (LTV)

Finally, we focus on the more general case of regularized linear time varying systems that are not necessarily switching or decomposed. In particular, we focus on two types of regularizations, 1) the operators change smoothly over time, i.e., $\|\boldsymbol{A}_t - \boldsymbol{A}_{t-1}\|_F^2$ is small, and 2) the operators are sparse, i.e., $\|\boldsymbol{A}_t\|_0$ is small.

For the LTV case, we apply LINOCS to find the time-changing operators $\{\boldsymbol{A}_t\}_{t=1}^T$ by iteratively integrating multi-step reconstruction with the appropriate regularization. The operators are initialized with a regularized 1-step optimization

$$\widehat{\boldsymbol{A}}_t = \arg\min_{\boldsymbol{A}_t} \|\widetilde{\boldsymbol{x}}_t - \boldsymbol{A}_t\widetilde{\boldsymbol{x}}_{t-1}\|_F^2 + \mathcal{R}(\boldsymbol{A}_t) = \arg\min_{\boldsymbol{A}_t} \|\widetilde{\boldsymbol{x}}_t - \boldsymbol{A}_t\widetilde{\boldsymbol{x}}_{t-1}\|_F^2 + \lambda\|\boldsymbol{A}_t - \boldsymbol{A}_{t-1}\|_F^2 \tag{13}$$

where $\lambda$ is the regularization weight. While in this paper we chose to focus on $\mathcal{R}(\boldsymbol{A}_t) = \lambda\|\boldsymbol{A}_t - \boldsymbol{A}_{t-1}\|_F^2$ other regularization terms can be used in a more general sense.

We integrate LINOCS into the estimation process by iteratively updating the operator estimates one at a time. Specifically, during each round of updates, we loop over every time point $t = 0 \dots T$ and hold all operators of times $\tau \neq t$ ($\{\boldsymbol{A}_\tau\}_{\tau\neq t}$) fixed at their former estimates. We then update $\boldsymbol{A}_t$ by

$$\widehat{\boldsymbol{A}}_t = \arg\min_{\boldsymbol{A}_t} \sum_{k=0}^K \left[ w_k \sum_{k_i=1}^{k+1} \|\widetilde{\boldsymbol{x}}_{t-k_i+k+1} - \widehat{\boldsymbol{A}}_{t-k_i+k-1}\widehat{\boldsymbol{A}}_{t-k_i+k-2}\cdots\boldsymbol{A}_t\cdots\widehat{\boldsymbol{A}}_{t-k_i+1}\widehat{\boldsymbol{A}}_{t-k_i}\widetilde{\boldsymbol{x}}_{t-k_i}\|_2^2 \right] + \mathcal{R}(\boldsymbol{A}_t)$$

where $k = 0\dots K$ denotes the order of the reconstruction and $t - k_i$ denotes the starting point of the reconstruction. The weights $\boldsymbol{w}^k$ are set as in the previous models. Please see Algorithm 4 for the procedural steps.

## 6 Results

To showcase LINOCS' ability to capture the dynamics in multiple models, we applied LINOCS to the above systems under diverse settings. The hyper-parameters used in each experiment are summarized in Section A.3.

---

**Algorithm 4** LTV-LINOCS

---

**Input:** Observations $\widetilde{\boldsymbol{X}}$, maximum order $K \in \mathbb{R}^+$, number of iterations $M$.
**Initialize:** $\{\boldsymbol{A}_t\}_{t=1}^T \sim \text{Uniform}$          ▷ Initialize from uniform distribution
**for** each iteration $m = 1 \ldots M$ **do**
    **for** each time point $t = 1 \ldots T$ **do**     ▷ To infer $\boldsymbol{A}_t$, consider $K + 1$ windows with shifts up to $K$
    Define the following quantities:
    $\boldsymbol{x}^+ := \text{diag}(\{\{\boldsymbol{x}_{t+k-k_i-1}\}_{k_i=1}^{k+1}\}_{k=0}^K) \in \mathbb{R}^{pK^* \times K^*}$
    $\boldsymbol{x}^- := \text{diag}(\{\{\boldsymbol{x}_{t-k_i}\}_{k_i=1}^{k+1}\}_{k=0}^K) \in \mathbb{R}^{pK^* \times K^*}$
    $\boldsymbol{B} := \text{concat}(\{\{\prod_{j=1}^{k_i} \widehat{A}_{t-j}\}_{k_i=1}^{k+1}\}_{k=0}^K, \text{axis} = 1) \in \mathbb{R}^{p \times pK^*}$        ▷ horizontal concatenation of $\boldsymbol{A}$
estimates for windows
    $\boldsymbol{C} := \text{concat}(\{\{\prod_{j=1}^{k-k_i} \widehat{A}_{t-k_i+k-j}\}_{k_i=1}^{k+1}\}_{k=0}^K, \text{axis} = 0) \in \mathbb{R}^{pK^* \times p}$     ▷ vertical concatenation of $\boldsymbol{A}$
estimates for shifts
    $\boldsymbol{\Psi} := \boldsymbol{B}\boldsymbol{x}^- \in \mathbb{R}^{p \times K^*}$
    $\widehat{\boldsymbol{A}_t} = \arg\min_{\boldsymbol{A}_t} \|\widetilde{\boldsymbol{x}^+} - \boldsymbol{C}\boldsymbol{A}_t\boldsymbol{\Psi}\|_F^2 + \lambda\|\boldsymbol{A}_t - \boldsymbol{A}_{t-1}\|_F^2$     ▷ Solve for $\boldsymbol{A}_t$ through Ridge regression
optimization
    **end for**
**end for**

---

## 6.1 LINOCS more accurately identifies ground truth linear systems under noisy observations

We first test LINOCS' ability to robustly learn time-invariant linear dynamical systems from noisy observations. We then simulate the dynamics $\boldsymbol{A} \in \mathbb{R}^{2 \times 2}$ as a rotational transition operator and a random offset $\boldsymbol{b} \in \mathbb{R}^{2 \times 1}$, where each $\boldsymbol{b}_i \sim \text{Uniform}(0, 1)$. We build the synthetic state $\boldsymbol{x}_t \in \mathbb{R}^{2 \times 1}$ for $T = 500$ time points, as $\boldsymbol{x}_t = \boldsymbol{A}\boldsymbol{x}_{t-1} + \boldsymbol{b}$ starting from random initial value $\boldsymbol{x}_0 \in \text{Uniform}(0, 1)^{2 \times 1}$, such that the noisy observations are $\widetilde{\boldsymbol{x}} = \boldsymbol{x} + \eta$, where the noise $\eta \sim \mathcal{N}(0, \sigma^2) = \mathcal{N}(0, 0.3^2)$ (Fig. 3A,B).

We compare the learned operators using LINOCS against four baselines. First we compare to traditional 1-step optimization (Eqn. (3)). We further compare the linear LINOCS to our implementations of the DAD approach (Venkatraman et al., 2015), as it is the approach closest to LINOCS in terms of integrating multi-step predictions into model training.

Our implementation of DAD integrates expert and non-expert demonstrations for model training, inspired by the Dataset-Aggregation (DAgger) approach (Ross et al., 2011). Specifically we test three implementations of DAD. For each implementation we initialized the transition matrix ($\boldsymbol{A}_{init}$) and the offset ($\boldsymbol{b}_{init}$) using the optimal estimate from 1-step optimization. We then, in each DAD implementation, train the model through 100 iterations where at each iteration, we used our last estimates of $\boldsymbol{A}$ and $\boldsymbol{b}$ to perform full lookahead reconstruction, starting from time $t = 0$. We then update our estimates of $\boldsymbol{A}$ and $\boldsymbol{b}$ using the optimal 1-step optimization while considering both the observations and the above full lookahead reconstruction.

Particularly, for the comparisons to DAD, we tested all three options outlined below:

- *DAD with full model update:*
  At each iteration, we update $\boldsymbol{A}$ and $\boldsymbol{b}$ based on the lookahead reconstruction of the state ($\widehat{\boldsymbol{x}}_t$) calculated based on the last operators estimate. Namely, $\{\widehat{\boldsymbol{A}}_{\text{iter}+1}, \widehat{\boldsymbol{b}}_{\text{iter}+1}\} = \arg\min_{\{\boldsymbol{A}, \boldsymbol{b}\}} \frac{1}{T} \sum_{t=0}^{T-1} \|\widetilde{\boldsymbol{x}}_{t+1} - (\boldsymbol{A}\widehat{\boldsymbol{x}}_t + \boldsymbol{b})\|_2^2$.

- *DAgger-inspired DAD (reweighed DAD):*
  For reweighted DAD, we estimate $\boldsymbol{A}$ and $\boldsymbol{b}$ at each iteration using both the observations and the lookahead reconstruction from the last estimates of $\boldsymbol{A}$ and $\boldsymbol{b}$. In particular, let $[\widehat{\boldsymbol{x}}_t, \widetilde{\boldsymbol{x}}_t] \in \mathbb{R}^{p \times 2}$ be a horizontal concatenation of the lookahead reconstruction and of the observations at time $t$. Then we iteratively solve: $\{\widehat{\boldsymbol{A}}_{\text{iter}+1}, \widehat{\boldsymbol{b}}_{\text{iter}+1}\} = \arg\min_{\{\boldsymbol{A}, \boldsymbol{b}\}} \frac{1}{T} \sum_{t=0}^{T-1} \| [\widehat{\boldsymbol{x}}_{t+1}, \widetilde{\boldsymbol{x}}_{t+1}] - (\boldsymbol{A}[\widehat{\boldsymbol{x}}_t, \widetilde{\boldsymbol{x}}_t] + \boldsymbol{b})\|_2^2$.

- *DAgger with $\ell_2$ constraint (reweighted DAD with $\ell_2$):*
  This option is solved similarly to the reweighted DAD, with the addition of the Frobenius norm ($\|\cdot\|_F$) on the operators ($\boldsymbol{A}$) and on ($\boldsymbol{b}$) during training.

We find that LINOCS identifies operators that yield accurate dynamics in long-time scale predictions (Fig. 3D). The other methods we tested, including 1-step optimization (Fig. 3C,E cyan) and DAD-based implementations (Fig. 3D, reds), instead indeed decay to zero (away from the real system), indicating a less accurate estimation of the dynamics. The improved accuracy of the operators identified by LINOCS (Fig. 3E) becomes apparent when examining the effects of the operators' estimation errors (Fig. 3C). These errors are larger for the other methods, showcasing that those methods accumulate more errors over shorter time spans than LINOCS' does.

We next investigated the effect of the training order in LINOCS on long-term reconstruction. We trained LINOCS on the noisy observations with increasing *training* orders and then tested the performance under increasing *prediction* orders (Fig. 3G). Compared to the baselines tested, LINOCS exhibits increased performance even with very low training orders (e.g., 5), with higher orders resulting in almost perfect reconstruction (Fig. 3G bottom-right subplot).

Additionally, exploring LINOCS's robustness to noise reveals that, unlike one-step reconstruction, LINOCS is robust even under very high levels of noise (Fig. 3G, H, blue). The resulting MSE compared to the ground truth dynamics is much lower in LINOCS, even under very high $\sigma$ noise levels (Fig. 3H, I blue vs. orange-red).

When examining the duration for which LINOCS remains robust without converging, we observe that our approach accurately predicts approximately 35,000 time points into the future before deviating from the real system and decaying to 0—demonstrating stability over exceptionally long time scales (Fig. 3J).

We further tested LINOCS on linear systems with structured noise (Fig. 13) as well as on a simulation of 3-dimensional cylinder (Fig. 14), yielding similar results. For structured noise, we modeled the observation as $\widetilde{\boldsymbol{x}}_t = \boldsymbol{x}_t + \sigma \sin(\gamma t)$ with $\sigma = 0.5$ and $\gamma = 3$ for $t = 1 \ldots 501$. Unlike other methods, LINOCS found operators that led to accurate long-term predictions (Fig. 13 D). Moreover, when examined under increasing training and prediction orders, we found that LINOCS is robust for long-term predictions, even for full lookahead reconstructions ($k_{\text{pred}} = 501$, Fig. 13 E,H). When evaluating its robustness to increasing structured noise levels ($\sigma$), we found that even for very high noise levels ($\sigma = 0.9$), LINOCS achieved much more robust results than 1-step optimization (Fig. 13 F,G). Additionally, when exploring how far into the future it enables robust reconstruction before converging, we found that it is capable of full lookahead for approximately 70,000 time points—a testament to its ability to find more robust operators that can adequately describe the system (Fig. 13 I).

For the 3D cylinder case (Fig. 14), with Gaussian noise ($\sigma = 0.4$), we similarly demonstrate that LINOCS recovers more accurate operators, leading to significantly more robust long-term predictions and enabling full recovery of the process (Fig. 14 A,B,C,D), both under increasing prediction orders (Fig. 14 E, G) and noise ($\sigma$) levels (Fig. 14 H). It further exhibits an impressive ability to reconstruct lookahead predictions (starting from $\boldsymbol{x}_0$) for very long periods (approximately 70,000 time points) before converging to similar error as of 1-step (Fig. 14 I). In contrast, 1-step optimization yields high-error within a few prediction orders.

## 6.2 LINOCS identifies accurate interactions in switching systems

We next tested LINOCS-driven SLDS as detailed in Section 5.2 on simulated data comprising of $J = 3$ discrete states. The transition operators for each of the distinct states was set to a $3 \times 3$ rotational matrix oriented in a different direction. Additionally, the offset for each state ($\boldsymbol{b}_j \in \mathbb{R}^{3\times 1}$) was set to be the same random vector drawn from a uniform distribution between 0 and 1 (Fig. 15D).

Notably, since the method is invariant to the order of the operators, to compare the identified operators to the ground truth operators, we sorted the operators using the "linear sum assignment" problem (SciPy's implementation, by Crouse (2016)), with the cost function being the Frobenius norm between each pair of $\boldsymbol{f}$s (ground truth vs. estimated for each model). As baselines, we compare the results of LINOCS-augmented SLDS with standard SLDS and recurrent SLDS (Linderman et al., 2016) with varying numbers of iterations.

When comparing LINOCS-SLDS to the baselines (Fig. 4), LINOCS consistently outperformed the other approaches across multiple metrics including operator recovery (Fig. 4C,E), switching times recovery (Fig. 4A,D), and dynamics reconstruction (Fig. 4B, Fig. 21B). In particular, LINOCS-SLDS accurately

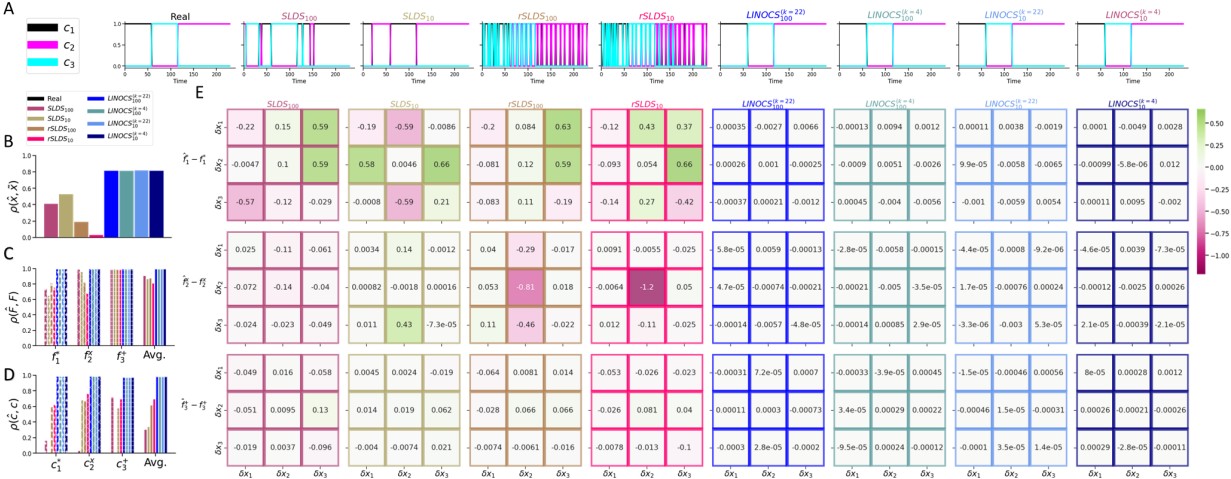

Figure 4: **Results on switching systems. A:** Active discrete states for LINOCS (blue) compared to baselines, including SLDS and rSLDS with 10 or 100 training epochs. **B:** Correlation ($\rho$) between the ground truth dynamics ($\boldsymbol{x}$) and the full-lookahead reconstructed dynamics ($\widehat{\boldsymbol{x}}$). **C:** Correlation ($\rho$) between the ground truth operators ($\boldsymbol{f}$) and the identified operators ($\widehat{\boldsymbol{f}}$). **D:** Correlation ($\rho$) between the ground truth coefficients ($\boldsymbol{c}$) and the identified coefficients ($\widehat{\boldsymbol{c}}$). **E:** Difference between the ground-truth sub-dynamics ($\widehat{\boldsymbol{F}}$) and reconstructed basis dynamics by different models. LINOCS was able to achieve sub-dynamics that are much closer to the ground truth than the other baselines.

identified switching times, whereas classical SLDS and rSLDS tended to introduce additional redundant switches (Fig. 4A). Moreover, the discrepancies between the ground truth operators and those identified by LINOCS (Fig. 4E, right-most four columns) were substantially smaller than the differences observed with classical SLDS/rSLDS (Fig. 4 E, left columns), as evidenced by the higher correlations between LINOCS' operators and the ground truth (Fig. 4C). Furthermore, when examining the eigenvalues of the identified operators compared to the ground truth (Fig. 16), the eigenspectrum derived from the LINOCS-driven solver closely resembled the ground truth eigenspectrum more than the classical SLDS and rSLDS cases, highlighting the effectiveness of LINOCS in capturing the underlying dynamics.

### 6.3 LINOCS finds dLDS operators that yield accurate dLDS lookahead predictions

Next, we applied LINOCS to dLDS, as described in Section 5.3. First we generated ground-truth data that represent a "pseudo-switching" (Fig. 2) process—i.e. linear dynamics that switch more smoothly (in our case between $J = 3$ systems) compared to SLDS where operators switch abruptly. This creates overlap periods where two dynamical systems are active at once as they trade off (Fig. 17). LINOCS-dLDS demonstrated significantly improved stability in full lookahead reconstruction compared to single-step dLDS (Fig. 5). Notably, training with orders approximately greater than 35 ($K_{\text{train}} > 35$) on our synthetic dataset (containing 1000 time points) resulted in highly accurate full reconstruction (Fig. 5A). Additionally, when comparing MSE and correlation of the time-evolving operator $\boldsymbol{F}_t = \sum_{j=1}^{J} c_{jt} \boldsymbol{f}_j$ to the ground truth, we observed a monotonic decrease in MSE with increasing maximal LINOCS training orders ($K_{\text{train}}$), while the correlation showed a monotonic increase (Fig. 5B).

Interestingly, although the 1-step prediction (post-training) is seemingly good also for non-LINOCS dLDS (or low-order LINOCS-dLDS) (Fig. 5D left), the advantage of LINOCS is revealed in Figure 5D right, under the full lookahead reconstruction. This implies that evaluating dynamical models not only based on their ability to predict immediate steps but also on their performance in further steps (i.e., under multistep reconstructions) is critical for more nuanced evaluations, as reconstruction errors could be obscured in 1-step predictions. Additionally, this comparison also highlights the importance of integrating multiple orders simultaneously during training.

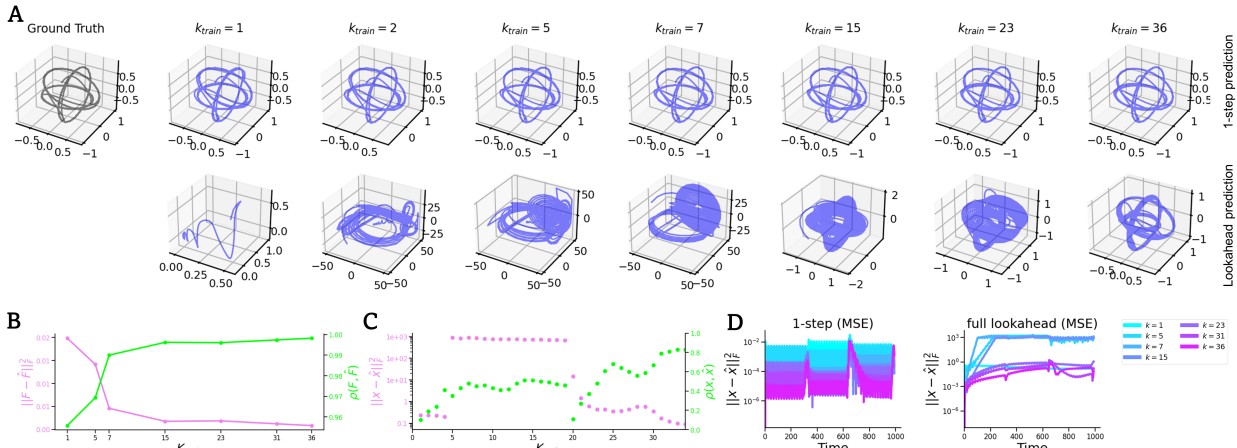

Figure 5: **Decomposed linear dynamical systems results. A:** Ground truth dynamics compared to 1-step (top) and full lookahead (bottom) reconstructions for non-LINOCS dLDS ($K_{train} = 1$) and LINOCS-dLDS with different training orders ($K_{train} \in [2, 5, 7, 15, 23, 36]$). **B:** MSE (pink) and correlation (green) between ground truth operators and the operators identified by LINOCS under different orders. **C:** MSE (pink) and correlation (green) between ground truth dynamics and full lookahead reconstructions using the different LINOCS training orders. **D:** Local MSE for 1-step (left) and for full lookahead reconstruction (right) over the time points of the dynamics.

We further extended our study to encompass more nuanced dLDS settings, exhibiting prolonged time scales and recurring patterns of identical active operators across distinct intervals (Fig. 18). We found analogous enhancements of LINOCS over the traditional 1-step dLDS implementation. Specifically, LINOCS demonstrates robust accurate long-term predictions, including full lookahead prediction (Fig. 6B), in contrast to 1-step optimization, which yield high lookahead error (Fig. 6C, last three subplots). Furthermore, also for this more complex example, upon comparing the identified time-varying transition operators $\boldsymbol{F}_t = \sum_j^J c_{jt} \boldsymbol{f}_j$ to the ground truth, LINOCS revealed operators with eigenvalues significantly more correlated with the real operators' evaluations (Fig. 6D, E) compared to the 1-step optimization results. Additionally, when comparing the operators' values themselves against the ground truth, those identified by LINOCS exhibited higher correlation and smaller MSE with the ground truth compared to these identified by 1-step dLDS (Fig. 6F,G,H).

### 6.4 LINOCS finds interactions that yield robust lookahead predictions in Linear Time-Varying (LTV) systems

To test the applicability of LINOCS to more general LTV systems, we implemented LINOCS-LTV to capture the chaotic behavior of the Lorenz attractor (Sec. A.4) through a smoothly changing LTV approximation (Fig. 7). We compared LINOCS-LTV with several other LTV solvers with varying constraints, including smoothness and sparsity ($\tau = 6, 7, 8$ and smoothness with weights $\lambda = 2, 20$, refer to Sec. A.2 for details). Unlike methods relying on 1-step optimization, LINOCS, despite similar regularization constraints, achieved superior full lookahead reconstruction (Fig. 7A bottom).

Also here, while different methods performed satisfactorily in the 1-step (post-training) prediction (Fig. 7A top, B red, C red), disparities emerged in higher-orders lookahead predictions where alternative methods failed. While all methods, including LINOCS, achieved commendable 1-step reconstruction, LINOCS demonstrated a markedly lower full lookahead error (Fig. 7B green, 5 most right bar pairs) and superior full-lookahead reconstruction correlation with the ground truth (Fig. 7C green, five most right bar pairs).

In addition, we analyzed operators identified across various training iterations of LINOCS to assess their proficiency in achieving lookahead reconstruction (Fig. 19). For this analysis, we used the Lorenz attractor with 900 time points with intervals of 0.1/9 arbitrary units (a.u.), and applied a smoothness constraint with

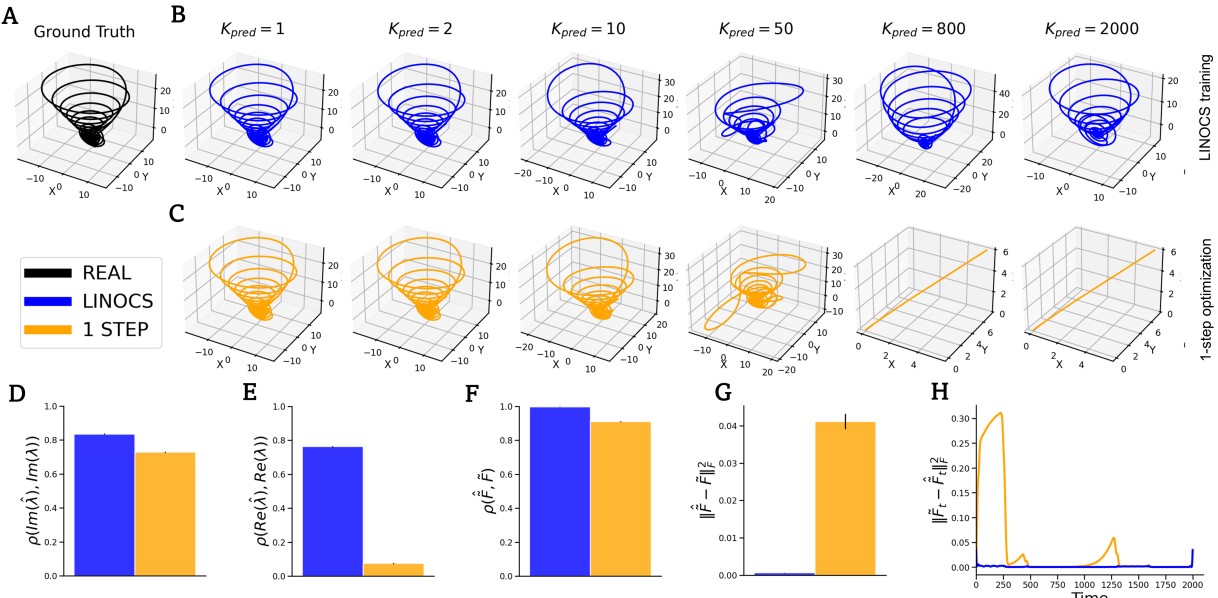

Figure 6: **Additional decomposed linear dynamical systems results ($K_{\mathbf{train}} = 50$). A:** Ground truth dynamics compared to (**B**) LINOCS results and (**C**) 1-step optimization results, for increasing prediction orders. **D,E:** Correlation between the eigenvalues of the ground truth transition matrix ($\boldsymbol{F}_t = \sum_j^J c_{jt}\boldsymbol{f}_j$) and the eigenvalues of the one identified by LINOCS. **D:** imaginary part; **E:** real part. Results display the average correlation over time. Eigenvalues were matched using the "linear sum assignment problem" (Scipy's Crouse (2016)). **F,G:** Comparing the identified time-changing transition matrix ($\boldsymbol{F}_t$) identified by LINOCS vs. 1-step optimization in terms of correlation (**F**) and MSE (**G**). **H:** Comparing the MSE of the identified $\boldsymbol{F}_t$ over time.

a weight of $\lambda = 0.1$. We observed that over training iterations, LINOCS adaptively influenced the predicted lookahead dynamics to gradually converge towards the ground truth dynamics (Fig. 19 C), with a monotonic decrease MSE (Fig. 19A, B).

When analyzing which time points of the dynamics contributed to higher MSEs in the full post-training lookahead prediction, we noticed that early training iterations tended to produce higher full lookahead prediction errors at later time points of the dynamics (Fig. 19B, top right). However, over subsequent iterations, the effect of LINOCS managed to mitigate the accumulation of errors at these late time points (Fig. 19B, bottom right).

## 6.5 LINOCS finds robust interactions in real-world neural data

Finally, we applied LINOCS to real-world dataset described by Kyzar et al. (2024), which consists of high density electrode array of populations of single units in the human medial temporal and medial frontal lobes while subjects were engaged in a screening task. We applied linear LINOCS, SLDS, dLDS-LINOCS, and LTV-LINOCS to a single recording session that includes recordings from five brain areas (amygdala left and right, cingulate cortex, hippocampus, pre-supplementary motor area). All the dynamical systems models were trained on the firing rate data, which we inferred from the spike-sorted electrophysiology via a Gaussian kernel convolution (see Sec. C.6 for details).

We investigated several LINOCS models to showcase their distinct characteristics. First, we examined the linear case for each brain area individually and explored the mean field interactions between areas (Fig. 8). Importantly, while typical real-world brain dynamics are assumed to be non-linear and non-stationary, our aim in starting with the linear model was to demonstrate how LINOCS can identify the fundamental background neural interactions under linear assumptions and check how its identified interactions defer from

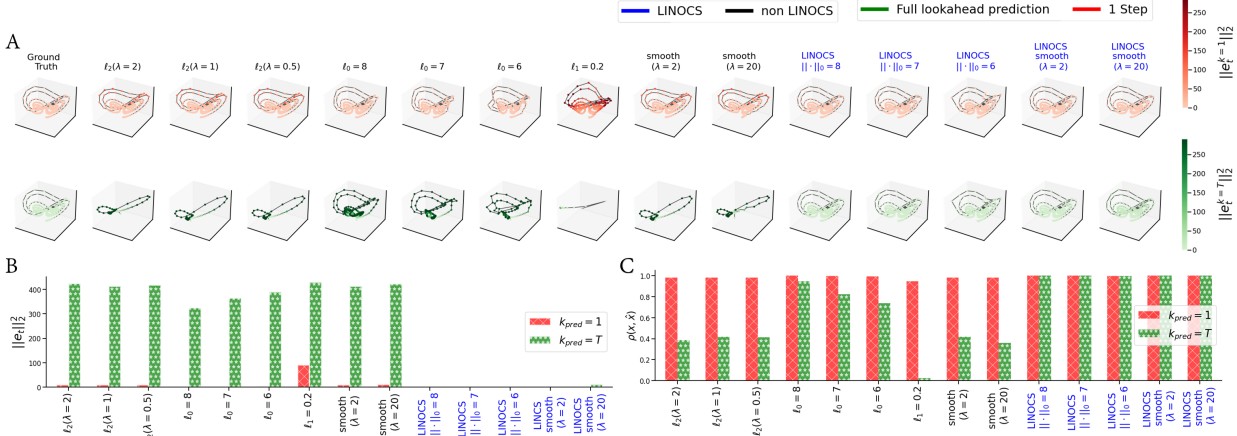

Figure 7: **LTV approximation of the Lorenz attractor. A:** 1-step post-training prediction (top, pink-red) vs. full lookahead prediction (bottom, green) for different baselines, including LINOCS with various smoothness levels or $\ell_0$ regularization. Color indicates the local MSE. $\lambda$ refers to the regularization weight. **B:** Squared $\ell_2$ of the error between the ground truth and 1-step prediction (red) vs. full lookahead predictions (starry green), for the different methods. **C:** Correlations between the ground truth and 1-step prediction (red) vs. full lookahead predictions (starry green), for the different methods.

these identified by the 1-step approach. We first applied the linear LINOCS on the firing rate activity from all neurons within each region to identify between-region interactions. We observed that LINOCS identified different linear interactions within areas compared to the 1-step optimization. Drawing from our conclusions based on synthetic data linear results, this suggests that LINOCS may provide a more nuanced linear approximation of brain activity compared to the common 1-step optimization (Fig. 8 A), though further comparisons with established neuroscience methods are needed to fully validate this advantage.

Then, we also applied the linear LINOCS on the mean activity of each region, and found that when examining the full lookahead reconstructions (Fig. 8 C) the 1-step optimization, in contrast to LINOCS, decayed to zero activity due to small accumulated deviations in operator values. In contrast, LINOCS managed to maintain activity closer to the average values of the dynamics. However, due to linear enforcement, neither approach could capture fluctuations in dynamics. Moreover, the full lookahead reconstruction error for LINOCS-linear was overall much smaller compared to the classical 1-step (Fig. 8B).

We next applied LINOCS-SLDS with three discrete states and compared it with regular SLDS using the same number of iterations. LINOCS identified operators that exhibit slight differences compared to those found by classical SLDS (Fig. 9B vs Fig. 22; Fig. 23 A vs. B) as well as slightly different switching patterns between the two approaches (Fig. 23 C,D). These operators resulted in significantly more robust lookahead predictions. Specifically, differences are evident in both connection presence, weights, and distribution among global operators. For example, in the "Amygdala left" region, both classical SLDS and LINOCS-driven SLDS identify a connection from neuron 10 to neuron 2 as part of $\boldsymbol{f}_3$, albeit with varying weights. Additionally, both methods identify connections from 5 to 3 (in $\boldsymbol{f}_1$ for classical SLDS and in $\boldsymbol{f}_2$ for LINOCS-SLDS) as well as from 6 to 3 (in $\boldsymbol{f}_2$ for classical SLDS and in $\boldsymbol{f}_1$ for LINOCS-SLDS), but with differing weights. Similar discrepancies are observed in other regions. Furthermore, LINOCS-SLDS and classical-SLDS each identify connections that the other overlooks; for instance, in the "Amygdala right" region, LINOCS-SLDS identifies a strong connection from 11 to 6, whereas classical SLDS does not. Conversely, classical SLDS identifies a connection from 3 to 6 (in $\boldsymbol{f}_3$), which LINOCS-SLDS does not recognize.

Importantly, the operators found by LINOCS enable full lookahead reconstruction without diverging, in contrast to regular classical SLDS that diverge to extreme values in full lookahead prediction (Fig. 9C). Moreover, the reconstruction error for the full lookahead prediction was overall much smaller for LINOCS-SLDS compared to the classical SLDS (Fig. 9A). These observations suggest that if the real neural process

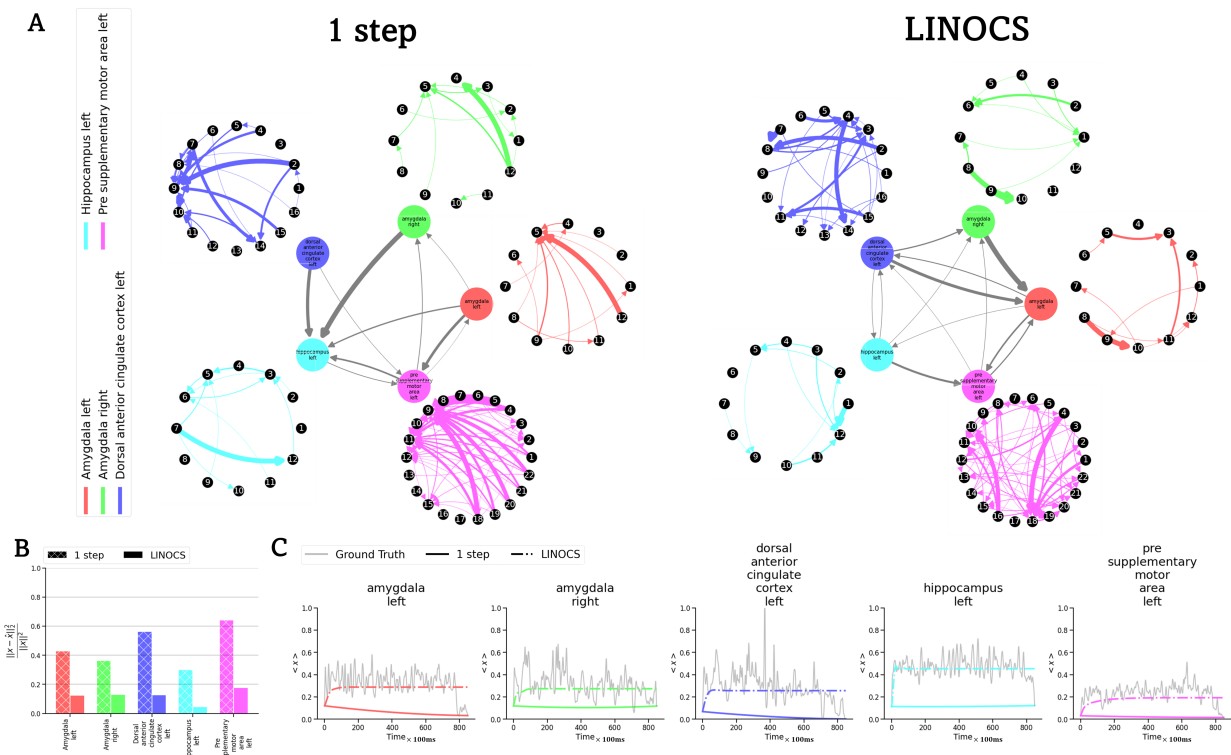

Figure 8: **Application of linear-LINOCS to multi-region neural recordings. A:** The per-region and between-region mean field linear dynamics operators identified by 1-step linear optimization vs. LINOCS with a linear time-invariant system. Each within-region network describes the linear operator $\widehat{\boldsymbol{A}}_{region}$ derived by applying LINOCS to the firing rate matrix constrained to include only the neurons from that region. Each node represents a unit of the spike sorted data, ideally corresponding to a single neuron. Colors of nodes represent their estimated brain region. Interactions between areas are found by applying LINOCS to the mean activity per area. Edges width indicate connection strength. **B:** Full-lookahead reconstruction error for 1-step linear optimization vs. LINOCS-linear approach. **C:** Ground truth mean activity per region compared to the lookahead prediction trace of the mean field activity by 1-step linear optimization vs. LINOCS-linear.

follows switching dynamics, LINOCS may capture the underlying dynamics more effectively, as inferred from our analysis of the synthetic case.

We observed similar patterns using dLDS-LINOCS, which revealed underlying global brain interactions potentially fundamental to brain function (Fig. 10 A). When examining their dynamic activations ($\boldsymbol{c}_t$), we noted a "background" interaction consistently active, with slight modulations over time (Fig. 10 B, brown), alongside gradually changing activities of other interactions (Fig. 10 B, gray-blue-purple). Importantly, these results provided lookahead predictions that did not decay and maintained a high correlation with the observations (Fig. 10 C).

Finally, employed the LTV-LINOCS on all neurons from all regions simultaneously while imposing a smoothness constraint on consecutive operators (with regularization of $\lambda = 0.1$ on $\|\boldsymbol{A}_t - \boldsymbol{A}_{t-1}\|_2^2$, Fig. 11). Our findings reveal that LINOCS identifies operators capable of producing full lookahead reconstructions without divergence, closely approximating observed data. Comparative analysis against 1-step optimization with various smoothness levels (Fig. 11B,C,E,F) underscores LINOCS' ability to achieve better reconstructions than other approaches. Additionally, examination of error evolution over time suggests a monotonic increase in error for non-LINOCS approaches (Fig. 11C). Moreover, we observed notable discrepancies between the operators identified by by LINOCS and the baselines (Fig. 11D). These results highlight the efficacy of LTV-LINOCS in capturing complex temporal dynamics in real world data while maintaining data fidelity.

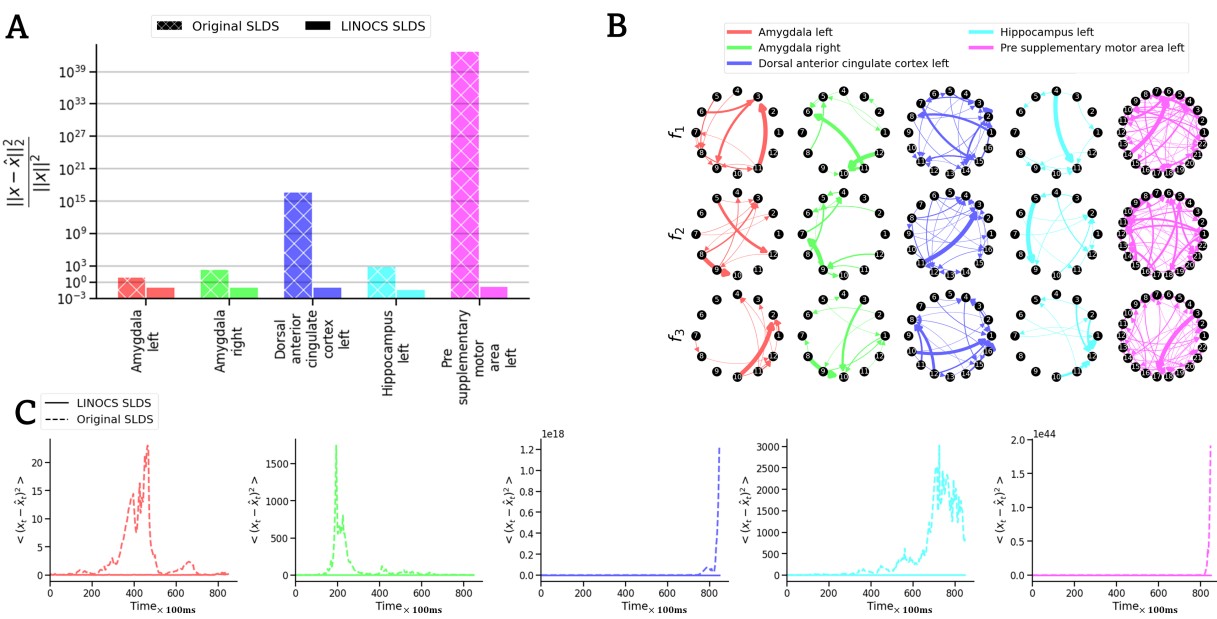

Figure 9: **SLDS results on real data. A:** Relative error of the full lookahead prediction compared to the ground truth, for both LINOCS-SLDS vs. classical SLDS, for each brain area. **B:** The networks identified by LINOCS-SLDS (see Fig. 22 for the networks identified by the classical-SLDS). **C:** Lookahead reconstruction using LINOCS-SLDS (solid curve) vs. classical SLDS (dashed curve). Classical-SLDS diverge to extreme values in the full lookahead reconstruction.

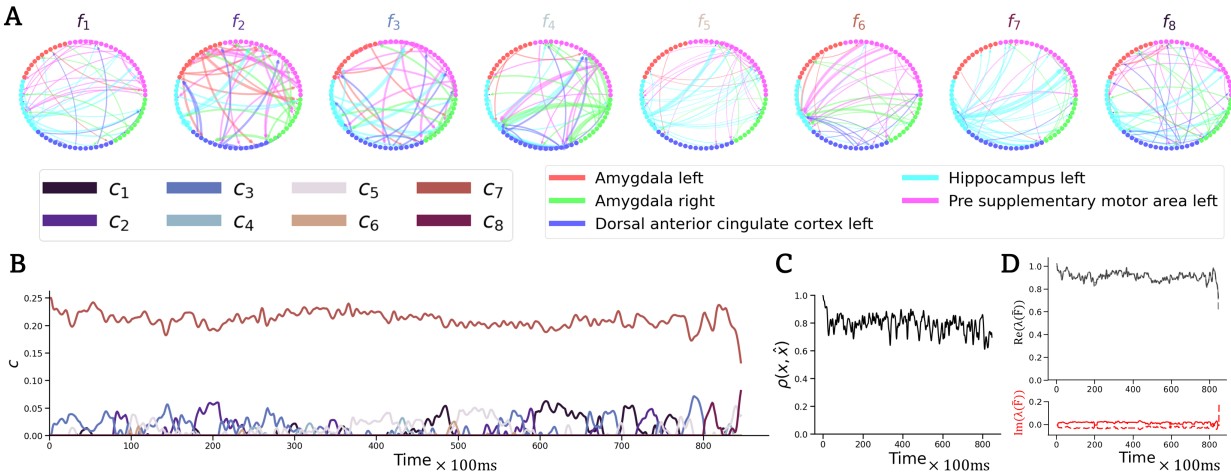

Figure 10: **dLDS-LINOCS results on the real neural data. A:** The identified operators $\{\boldsymbol{f}_j\}_{j=1}^{J=8}$. **B:** The identified sparse coefficients $\boldsymbol{c}_t$. **C:** Correlation between the full-lookahead reconstruction results and the observations. **D:** The two largest eigenvalues ($\lambda$) of the time-varying transition operator $\widetilde{\boldsymbol{F}}_t = \sum_{j=1}^{8} c_{jt} \boldsymbol{f}_j$. Black (top): real part. Red (bottom): imaginary part.

Overall, we showed that in all these real-world neural versions, LINOCS was able to recover more robust descriptions of the dynamic evolution for the long run, which, based on our synthetic results, may imply that these are closer to the real unknown interactions.

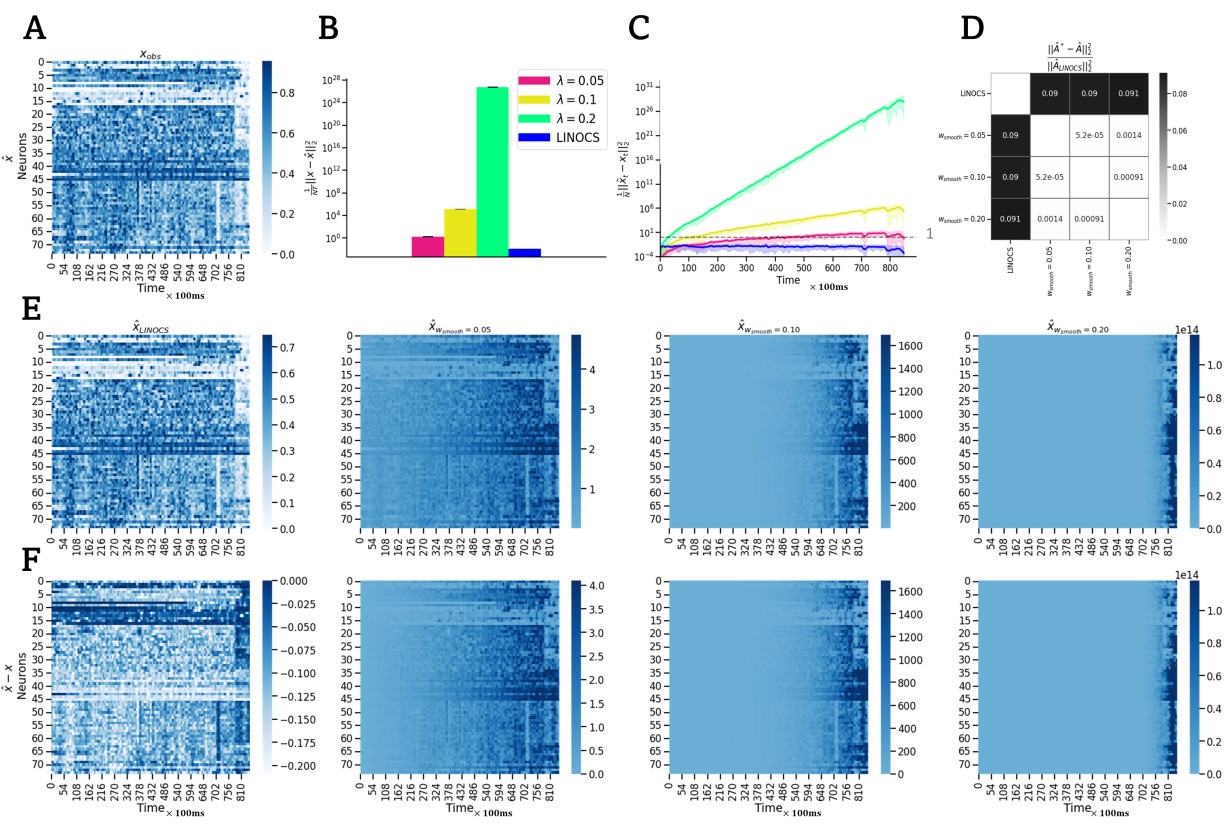

Figure 11: **LTV-LINOCS results on real neural data. A:** The ground truth data. **B:** MSE between the ground truth data and the full lookahead reconstructions. **C:** MSE of the reconstruction over time points, compared to 1-step optimization with different smoothness levels. **D:** Frobenius norm of the differences between **A**s identified by the different models, normalized by the magnitude of the operator identified by LINOCS. **B:** Full-lookahead predictions produced by LINOCS-LTV (left) and 1-step optimization with increasing smoothness constraints ($\lambda$). **F:** Difference between the observations and the full lookahead reconstruction for LINOCS-LTV vs. 1-step optimization with increasing smoothness levels.

## 7 Discussion

In this paper, we introduced LINOCS (Lookahead Inference of Networked Operators for Continuous Stability), a learning procedure to improve stability and accuracy of dynamical system inference that leverages multiple lookahead estimations. By iteratively integrating re-weighted multi-step reconstructions with additional constraints on the operators, LINOCS enables robust inference of networked operators in dynamical systems, even in the presence of noise and nonlinearity.

Our experimental results highlight LINOCS' effectiveness across various dynamical systems, including Linear Systems (LDSs), Switching Linear Systems (SLDS), decomposed LDSs (dLDS) (Mudrik et al., 2024a), and Linear Time-Varying Systems (LTV) in both simulated and real-world neural data. LINOCS achieves more precise full lookahead reconstruction and more accurately retrieves ground truth operators in synthetic data compared to baseline methods, highlighting its ability to better capture nuanes in the underlying system. These findings suggest that LINOCS holds greater potential than alternative approaches for accurately identifying unknown hidden interactions also in real-world data, where the real underlying interactions are typically obscured but pivotal for robust scientific interpretation.

Looking ahead, several promising avenues exist for future work. These include applying LINOCS to a broader range of datasets to uncover new scientific discoveries about component interactions (e,g,. using LINOCS-dlds to identify functional interaction motifs in *C. elegans*, as done for 1-step inference in Yezerets et al.

(2024)). Additionally, from a computational standpoint, LINOCS could be extended to improve operator inference and reconstruction robustness in models like TVART Harris et al. (2021), or be integrated into RNN training to enhance learning and prediction stability. Particularly, if extending LINOCS to deep networks, the integration of multi-step reconstructions into the networks' training, may be used to address issues such as vanishing or exploding gradients. Another future step would be to integrate an offset term identification into the dLDS application (e.g., as addressed in Chen et al. (2024) for 1-step inference). Additionally, extending LINOCS to handle non-linear local transformations (i.e., $\boldsymbol{x}_t = g(\boldsymbol{A}_t \boldsymbol{x}_{t-1})$, where $g$ is a non-linear activation), as well as non-Gaussian noise, could enhance its applicability to a wider range of real-world scenarios.

## 8 Code and Data Availability

The code is shared on GitHub and available at `https://github.com/NogaMudrik/LINOCS`, with tutorial notebooks available **here** (for the Linear-LINOCS example) and at `https://github.com/NogaMudrik/LINOCS/blob/main/run_dLDS_Lorenz_example.ipynb` for the dLDS-LINOCS example. The human neural recordings data we used to exemplify LINOCS is available at (Kyzar et al., 2024).

A video briefly explaining the paper is available at `https://youtu.be/5XVYRHd5wGs`.

## 9 Acknowledgments

We thank our funding sources for supporting us during this project. N.M. was funded by The Kavli Foundation NeuroData Discovery award. Y.C. and C.R. were funded by James S. McDonnell Foundation grant number 22002039, with Y.C. being further funded by National Institutes of Health grant number 2T32EB025816, and C.R. being further funded by the Julian T. Hightower Chair. A.S.C. and E.Y. were partially supported by the NSF CAREER Award 2340338 and a Johns Hopkins Bridge Grant.

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

# A    Appendix

## A.1    calculation details for operators differences (Fig. 3C)

We computed the operator differences illustrated in Figure 3C using the expression:

$$((\widehat{\boldsymbol{A}} - \boldsymbol{I})\boldsymbol{x} - (\boldsymbol{A} - \boldsymbol{I})\boldsymbol{x}) \times \text{factor}$$

Here, $\widehat{\boldsymbol{A}}$ represents the operators identified by the different methods, $\boldsymbol{A}$ denotes the ground-truth operators, and "factor" is a scalar used for visualization purposes only (identical for all methods).

## A.2    Regularization options in LTV system experiment

we applied the following regularization options in Sec. 6.4.

- **Smoothness:** The objective function becomes: $\boldsymbol{A}_t = \arg\min \|\widetilde{\boldsymbol{x}}_t - \boldsymbol{A}_t\widetilde{\boldsymbol{x}}_t\|_F^2 + \lambda\|\boldsymbol{A}_t - \boldsymbol{A}_{t-1}\|_F^2 \forall t = 2 \ldots T$

- **Sparsity Regularization:** $\boldsymbol{A}_t = \arg\min \|\widetilde{\boldsymbol{x}}_t - \boldsymbol{A}_t\widetilde{\boldsymbol{x}}_t\|_F^2$ s.t. $\|\boldsymbol{A}_t\|_0 <= \tau$

## A.3    Hyperparamters used in experiments

Table 1: Hyperparameter settings for DAD baseline in linear experiment

| Parameter | Value | Additional Info |
|---|---|---|
| seed | 0 | random seed |
| T | 500 | number of time points |
| $w_{\ell_2}$ | 0 | weight of $\ell_2$ regularization on dynamics |
| $w_{decay}$ | 1 | decay of regularization coefficient over iterations |
| $N_{iterations}$ | 100 | number of iterations |
| A_init_type | step | initialize $A$ with 1-step optimization |
| reweight | False | whether to reweight the observations and the lookahead during training. |

Table 2: Hyperparameter settings for "DAD reweigh" baseline in linear experiment

| Parameter | Value | Additional Info |
|---|---|---|
| seed | 0 | random seed |
| T | 500 | number of time points |
| $w_{\ell_2}$ | 0 | weight of $\ell_2$ regularization on dynamics |
| $w_{decay}$ | 1 | decay of regularization coefficient over iterations |
| $N_{iterations}$ | 100 | number of iterations |
| A_init_type | 'step' | initialization type for matrix A |
| reweight | True | whether to reweight the observations and the lookahead during training. |

Table 3: Hyperparameter settings for "reweigh $\ell_2$" baseline in linear experiment

| Parameter | Value | Additional Info |
|---|---|---|
| seed | 0 | random seed |
| T | 500 | number of time points |
| $w_{\ell_2}$ | 1 | weight of $\ell_2$ regularization on dynamics |
| $w_{decay}$ | 1 | decay of regularization coefficient over iterations |
| $N_{iterations}$ | 100 | number of iterations |
| A_init_type | 'step' | initialization type for matrix A |
| reweight | True | whether to reweight the observations and the lookahead during training. |

Table 4: Hyperparameter settings for LINOCS in linear experiment

| Parameter | Value | Additional Info |
|---|---|---|
| K | 80 | maximum training order |
| with offset | True | whether to look for an offset term $\boldsymbol{b}$ |
| cal_offset | True | whether to calculate offset |
| weights_style | exponential | weight style for orders |
| $\sigma_w$ | exponential parameter for the weights | 0.01 |
| infer_b_way | 'each' | approach to infer the onset. Based on each order. |
| $K_b$ | 20 | maximum lookahead training order for the offsets $\boldsymbol{b}$ |
| weights_style_b | 'exponential' | weights style for the offsets $\boldsymbol{b}$ |

### A.4 Lorenz equations

The Lorenz attractor follows:

$$\frac{dx}{dt} = \sigma(y - x),$$
$$\frac{dy}{dt} = x(\rho - z) - y,$$
$$\frac{dz}{dt} = xy - \beta z,$$

where $x$, $y$, and $z$ represent the state variables, and $\sigma$, $\rho$, and $\beta$ are the system parameters, set to $\sigma = 10$, $\rho = 28$, and $\beta = \frac{8}{3}$.

### A.5 Calculation of Operator and Offset for Linear Experiment

We begin by assuming that $\boldsymbol{X}$ represents the observations in this subsection. Our analysis involves two main objectives: estimating the transition operator $\boldsymbol{A}$ and the offset $\boldsymbol{b}$.

We define a set of matrices $\{\boldsymbol{\psi}_k\}_{k=1}^K$, where each $\boldsymbol{\psi}_k$ is found by solving:

$$\widehat{\boldsymbol{\psi}_k} = \arg\min_{\boldsymbol{\psi}_k} \|\boldsymbol{X}_{:,k:} - \boldsymbol{\psi}_k[\boldsymbol{X}_{:,:T-k}, [1]_{1 \times T-k}]\|_F^2$$

Here, $[1]_{1 \times T-k}$ is a row vector of ones, and $[\boldsymbol{X}_{:,:T-k}, [1]_{1 \times T-k}]$ is the vertical concatenation. Each $\boldsymbol{\psi}_k$ captures $\boldsymbol{A}^{k+1} + \sum_{k_i=0}^k \boldsymbol{A}^{k_i}\boldsymbol{b}$, with the last column of $\boldsymbol{\psi}_k$ specifically capturing $\sum_{k_i=0}^k \boldsymbol{A}^{k_i}\boldsymbol{b}$.

We extract all but the last column from each $\boldsymbol{\psi}_k$:

$$\widehat{\boldsymbol{A}} = \arg\min_{\boldsymbol{A}} \|\boldsymbol{A}^k - (\boldsymbol{\psi}_k)_{:,:p}\|_F^2$$

Table 5: Hyperparameter settings for dLDS experiment

| Parameter | Value | Additional Info |
|---|---|---|
| K | 50 | maximum lookahead order |
| additional_update | True | whether to include an additional update step |
| $\ell_1$_init | 0 | value of $\ell_1$ on the coefficients for the 1-st iteration |
| max_iters | 200 | maximum number of iterations |
| $\|F\|_2$ | 0 | $\ell_2$ norm on the basis of dLDS dynamics |
| $freq_{update_F}$ | 5 | frequency of updating $\{\boldsymbol{f}_j\}_{j=1}^J$ |
| $\ell_{2decay}$ | 0.99 | decay of the $\ell_2$ norm on the coefficients |
| $\ell_{1decay}$ | 0.9999, | decay of the $\ell_1$ norm on the coefficients |
| $w_{smooth decay}$ | 1.01, | decay of smoothing weight on coefficients |
| l_smooth_time | 1.1, | regularization weight on coefficients smoothing $\boldsymbol{c}_t$. |
| $\ell_2$ | 0, | $\ell_2$ on the coefficients |
| $w_{smooth_{time}}$ | 0.1, | smoothness on $\boldsymbol{c}_t$ over time |
| $\sigma_{noisy_c}$ | 0.05 | std of the noise to add to coefficients during training. |
| $\ell_1$ | 2.5, | $\ell_1$ regularization on $\boldsymbol{c}_t$ |
| decor | False | wether to decorrelate the basis dynamics |
| max_interval_k | 1 | maximum interval to increase $k$ during training |
| to_norm_F | True | whether to normalize the basis dynamics |
| $\boldsymbol{x}_0$ | [0.2160895, 0.97627445, 0.00623026] | ground truth initial state (at $t = 0$) |
| J | 3 | number of basis dynamics operators ($\boldsymbol{f}$s) |

Table 6: Hyperparameter setting for the LTV-Lorenz experiment

| Parameter | Value | Additional Info |
|---|---|---|
| $K$ | 5 | maximum lookahead order |
| $w_{smooth}$ | 2 (or 20) | smoothness weight |
| $estimate_{thres}$ | 2 | threshold to increase $k$ value |
| with_future_cost_or | True | consider both fast and future reconstructions |
| $with_{future}$ | True | wether to apply time smoothness to future weights |
| $w_{smooth_{future}}$ | 0.05 | weights future smoothness |
| $weights_{style}$ | 'uni' | Uniform weight style. |
| $\|w\|_F$ | 0 | Frobenius norm weights |
| init_style | 'step' | initialize based on optimal 1-step |
| $max_{iters}$ | 40 | maximum number of iterations |
| $error_{thres}$ | 8 | threshold to increase error |
| $with_{reverse}$ | False | whether to include reverse update |
| $N_{zeros}$ | 0 or 1 or 2 or 3 | how many zeros (relevant only to the sparse networks) |

This operation isolates the transition operator $\boldsymbol{A}$ by minimizing the Frobenius norm of the differences across all k levels.

Using the last columns of the original $\boldsymbol{\psi}$ matrices, we estimate $\boldsymbol{b}$, considering our previously estimated $\boldsymbol{A}$. We model each observation $\boldsymbol{x}_t$ as:

$$\boldsymbol{x}_t \approx \boldsymbol{A}\boldsymbol{x}_{t-1} + \boldsymbol{b} \tag{14}$$

Expanding recursively for $k$ steps, we derive:

$$\boldsymbol{x}_t \approx \boldsymbol{A}^k \boldsymbol{x}_{t-k} + \sum_{k_i=0}^{k-1} \boldsymbol{A}^{k_i} \boldsymbol{b} \tag{15}$$

To organize our data for analysis, we define shifted observation matrices $\boldsymbol{X}^{+k} := \boldsymbol{X}_{:,k:}$ and corresponding estimated matrices $\widehat{\boldsymbol{X}^{+k}} := \boldsymbol{A}^k \boldsymbol{X}_{:,:T-k}$. These are concatenated vertically for all $k = 1 \ldots K$, adjusting the dimensions as necessary with NaN padding.

Additionally, we construct $\widetilde{\boldsymbol{A}} \in \mathbb{R}^{kp \times p}$ as a vertical concatenation of $\left\{ \sum_{k_i=0}^{k-1} \boldsymbol{A}^{k_i} \right\}_{k=1}^{K}$.

We solve for the offset $\boldsymbol{b}$ by minimizing:

$$\widehat{\boldsymbol{b}} = \arg \min_{\boldsymbol{b}} \| \boldsymbol{X}^{+K} - \widehat{\boldsymbol{X}^{+K}} - \widetilde{\boldsymbol{A}}\boldsymbol{b} \|_F^2 \tag{16}$$

This minimization is achieved via simple least squares, and $\boldsymbol{b}$ is then averaged across times.

# B  Explanation of Baselines for Comparing LTV-LINOCS

We compared the LTV-LINOCS system against two other multi-step approaches, each with various parameter settings:

1. **SSM-RNN Jordana et al. (2021)**: We used the code from the GitHub repository linked to the original paper Jordana et al. (2021). We experimented with different numbers of epochs, network architectures (fully connected vs. locally linear), seeds, and numbers of shooting points. Details are provided in Table 9.

2. **GTF-shPLRNN Hess et al. (2023)**: We utilized the code available in the GitHub repository here, which is linked to the original paper Hess et al. (2023) and written in Julia. We based the parameters on the values defined in the file `GTF-shPLRNN/paper/_experiments/EEG/Table1/shPLRNN/_aGTF.jl` but modified the observation model to the identity.

It is important to note that while we did not perform exhaustive hyperparameter tuning, we examined 12-16 hyperparameter combinations around the default settings. Details of the hyperparameters are listed in Tables 7, 8, and 9.

For reference, we used the ground truth version of the Lorenz attractor, shown in Figure 7, and present the results in Figure 20.

# C  Computational Complexity

## C.1  Linear System

Given that $\boldsymbol{x} \in \mathbb{R}^{N \times T}$, and $\boldsymbol{A} \in \mathbb{I} \times \mathbb{I}$, computing the highest power $K$ of $\boldsymbol{A}$ ($\boldsymbol{A}^K$) through repeated squaring involves $K$ matrix multiplications, each with a complexity of $\mathcal{O}(N^3)$, resulting in $\mathcal{O}(KN^3)$.

If the optimizer is set to a maximum of $M$ iterations, then the total complexity is $\mathcal{O}(MKN^3)$. In the linear case, the addition of the offset term is integrated into the inference by extending the size of $\boldsymbol{A}$ by 1, which does not affect the computational complexity scale (assuming $N >> 1$).

Table 7: Parameters for GTF baseline options

| | option_1 | option_2 | option_3 | option_4 | option_5 | option_6 | option_7 | option_8 | option_9 | option_10 | option_11 | option_12 |
|---|---|---|---|---|---|---|---|---|---|---|---|---|
| partial_forcing | FALSE | FALSE | FALSE | FALSE | FALSE | FALSE | FALSE | FALSE | FALSE | FALSE | FALSE | FALSE |
| latent_dim | 16 | 16 | 16 | 16 | 16 | 16 | 16 | 16 | 16 | 16 | 16 | 16 |
| scalar_saving_interval | 500 | 500 | 500 | 500 | 500 | 500 | 500 | 500 | 500 | 500 | 500 | 500 |
| teacher_forcing_interval | 16 | 16 | 16 | 16 | 16 | 16 | 16 | 16 | 16 | 16 | 16 | 16 |
| gaussian_noise_level | 0 | 0 | 0 | 0 | 0 | 0 | 0 | 0 | 0 | 0 | 0 | 0 |
| use_gtf | TRUE | TRUE | TRUE | TRUE | TRUE | TRUE | TRUE | TRUE | TRUE | TRUE | TRUE | TRUE |
| optimizer | RADAM | RADAM | RADAM | RADAM | RADAM | RADAM | RADAM | RADAM | RADAM | RADAM | RADAM | RADAM |
| batch_size | 16 | 16 | 16 | 16 | 16 | 16 | 16 | 16 | 16 | 16 | 16 | 16 |
| start_lr | 0.001 | 0.001 | 0.001 | 0.001 | 0.001 | 0.001 | 0.001 | 0.001 | 0.001 | 0.001 | 0.001 | 0.001 |
| path_to_inputs | | | | | | | | | | | | |
| gtf_alpha | 1 | 1 | 1 | 0.8 | 0.8 | 0.8 | 1 | 1 | 1 | 0.9 | 0.9 | 0.9 |
| batches_per_epoch | 50 | 50 | 50 | 50 | 50 | 50 | 50 | 50 | 50 | 50 | 50 | 50 |
| gtf_alpha_decay | 0.999 | 0.999 | 0.999 | 0.999 | 0.999 | 0.999 | 0.999 | 0.999 | 0.999 | 0.999 | 0.999 | 0.999 |
| observation_model | Identity | Identity | Identity | Identity | Identity | Identity | Identity | Identity | Identity | Identity | Identity | Identity |
| lat_model_regularization | 0.0001 | 0.0001 | 0.0001 | 0.0001 | 0.0001 | 0.0001 | 0.0001 | 0.0001 | 0.0001 | 0.0001 | 0.0001 | 0.0001 |
| D_stsp_bins | 30 | 30 | 30 | 30 | 30 | 30 | 30 | 30 | 30 | 30 | 30 | 30 |
| PE_n | 20 | 20 | 20 | 20 | 20 | 20 | 20 | 20 | 20 | 20 | 20 | 20 |
| end_lr | 1e-06 | 1e-06 | 1e-06 | 1e-06 | 1e-06 | 1e-06 | 1e-06 | 1e-06 | 1e-06 | 1e-06 | 1e-06 | 1e-06 |
| device | cpu | cpu | cpu | cpu | cpu | cpu | cpu | cpu | cpu | cpu | cpu | cpu |
| gradient_clipping_norm | 0 | 0 | 0 | 0 | 0 | 0 | 0 | 0 | 0 | 0 | 0 | 0 |
| D_stsp_scaling | 1 | 1 | 1 | 1 | 1 | 1 | 1 | 1 | 1 | 1 | 1 | 1 |
| image_saving_interval | 500 | 500 | 500 | 500 | 500 | 500 | 500 | 500 | 500 | 500 | 500 | 500 |
| num_bases | 50 | 50 | 50 | 50 | 50 | 50 | 50 | 50 | 50 | 50 | 50 | 50 |
| hidden_dim | 512 | 512 | 512 | 512 | 512 | 512 | 512 | 512 | 512 | 512 | 512 | 512 |
| sequence_length | 50 | 50 | 50 | 50 | 50 | 50 | 50 | 50 | 50 | 50 | 50 | 50 |
| MAR_ratio | 0 | 0 | 0 | 0 | 0 | 0 | 0 | 0 | 0 | 0 | 0 | 0 |
| obs_model_regularization | 1e-06 | 1e-06 | 1e-06 | 1e-06 | 1e-06 | 1e-06 | 1e-06 | 1e-06 | 1e-06 | 1e-06 | 1e-06 | 1e-06 |
| alpha_update_interval | 5 | 5 | 5 | 5 | 5 | 5 | 5 | 5 | 5 | 5 | 5 | 5 |
| epochs | 5000 | 5000 | 5000 | 5000 | 5000 | 5000 | 5000 | 5000 | 5000 | 5000 | 5000 | 5000 |
| MAR_lambda | 0 | 0 | 0 | 0 | 0 | 0 | 0 | 0 | 0 | 0 | 0 | 0 |
| run | 1 | 1 | 1 | 1 | 1 | 1 | 1 | 1 | 1 | 2 | 2 | 2 |
| PSE_smoothing | 20 | 20 | 20 | 20 | 20 | 20 | 20 | 20 | 20 | 20 | 20 | 20 |
| epoch number | 500 | 1000 | 1500 | 500 | 1000 | 1500 | 500 | 1000 | 1500 | 500 | 1000 | 1500 |

Table 8: GTF Model of Each option

| | model |
|---|---|
| **option_1** | clippedShallowPLRNN |
| **option_2** | clippedShallowPLRNN |
| **option_3** | clippedShallowPLRNN |
| **option_4** | clippedShallowPLRNN |
| **option_5** | clippedShallowPLRNN |
| **option_6** | clippedShallowPLRNN |
| **option_7** | shallowPLRNN |
| **option_8** | shallowPLRNN |
| **option_9** | shallowPLRNN |
| **option_10** | shallowPLRNN |
| **option_11** | shallowPLRNN |
| **option_12** | shallowPLRNN |

## C.2 dLDS

The direct dLDS version we propose in the paper works directly on the observations rather than on a latent low-dimensional space. This process involves both identifying the dynamic operators $\{\boldsymbol{f}_j\}_{j=1}^J$ and their coefficients ($\{\boldsymbol{c}_t\}_{t=1}^T$).

We will start with the complexity for the coefficients inference. Particularly, the inference of $\boldsymbol{c}_t$ for each $t = 1 \ldots T$, involves: $\widehat{\boldsymbol{c}}_t = \arg\min_{\boldsymbol{c}_t} \|\widetilde{\boldsymbol{x}_{t+1}}_{\text{vert}} - \widetilde{\boldsymbol{F}}_{x_t}^K \boldsymbol{c}_t\|_F^2$ where $\widetilde{\boldsymbol{F}}_{x_t}^K \in \mathbb{R}^{(K+1)p \times J}$ and $(\widetilde{\boldsymbol{x}}_{t+1})_{vert} \in \mathbb{R}^{p(K+1) \times 1}$. Hence, assuming $(K+1)p >> J$, performing the above least squares to infer each $\boldsymbol{c}_t$ requires $\mathcal{O}(J^2(K+1)p) = \mathcal{O}(J^2 Kp)$ assuming $K > 1$ for the pseudo-inverse and $\mathcal{O}(J(K+1)p) = \mathcal{O}(JKp)$ for the matrix-vector multiplication, resulting in overall $\mathcal{O}(J^2 Kp)$ for each time point, and $\mathcal{O}(MTJ^2Kp)$ for $T$ time points and $M$ overall model iterations.

To infer the elements of the set $\{\boldsymbol{f}_j\}_{j=1}^J$

$$\widehat{\boldsymbol{F}}_{all} = \arg\min_{\boldsymbol{F}_{all}} \|\widetilde{\boldsymbol{X}}_{:,2:} - \boldsymbol{F}_{all}(\boldsymbol{X_c})\|_F^2,$$

Table 9: Parameters for RNN Multiple-Shooting baseline options

|  | num_shooting | len_filt | max_it | argT | alpha | log_name | bs | seed | num_epochs |
|---|---|---|---|---|---|---|---|---|---|
| **Option 1** | 5 | 100 | 50 | 10000 | 100 | locally_linear | 40 | 1 | 5 |
| **Option 2** | 5 | 100 | 50 | 210 | 100 | locally_linear | 40 | 4 | 1000 |
| **Option 3** | 5 | 100 | 50 | 210 | 100 | locally_linear | 40 | 4 | 1000 |
| **Option 4** | 200 | 100 | 50 | 10000 | 100 | 5 | 40 | 4 | 5 |
| **Option 5** | 5 | 100 | 50 | 210 | 100 | locally_linear | 40 | 4 | 1000 |
| **Option 6** | 2 | 100 | 50 | 210 | 100 | locally_linear | 40 | 4 | 1000 |
| **Option 7** | 5 | 100 | 50 | 210 | 100 | locally_linear | 40 | 4 | 100 |
| **Option 8** | 2 | 100 | 50 | 210 | 100 | locally_linear | 40 | 40 | 100 |
| **Option 9** | 2 | 100 | 50 | 210 | 100 | locally_linear | 40 | 480 | 100 |
| **Option 10** | 2 | 100 | 50 | 210 | 100 | locally_linear | 40 | 480 | 100 |
| **Option 11** | 2 | 100 | 50 | 210 | 20 | locally_linear | 40 | 480 | 100 |
| **Option 12** | 2 | 100 | 50 | 210 | 5 | locally_linear | 40 | 480 | 100 |
| **Option 13** | 2 | 100 | 50 | 210 | 1 | locally_linear | 40 | 480 | 100 |
| **Option 14** | 2 | 100 | 10 | 210 | 1 | fully_connected | 40 | 480 | 100 |
| **Option 15** | 5 | 100 | 10 | 210 | 1 | fully_connected | 40 | 480 | 1000 |
| **Option 16** | 2 | 100 | 10 | 210 | 100 | fully_connected | 40 | 480 | 1000 |

Table 10: Parameters for LINOCS baseline options for comparison figure

|  | **Option 1** | **Option 2** | **Option 3** | **Option 4** | **Option 5** | **Option 6** | **Option 7** |
|---|---|---|---|---|---|---|---|
| $\boldsymbol{\lambda}_{\text{smooth}}$ | 0 | 0 | 0 | 0 | 0 | 2 | 20 |
| **num_zeros** | 0 | 0 | 1 | 2 | 3 | 0 | 0 |
| **max_K** | 8 | 3 | 4 | 4 | 4 | 5 | 5 |
| **max_iters** | 15 | 15 | 25 | 25 | 25 | 40 | 40 |

where $\widetilde{\boldsymbol{X}} \in \mathbb{R}^{p \times (T+1)}$ and $\boldsymbol{X}_c \in \mathbb{R}^{pJ \times T}$. Assuming $pJ < T$, the overall complexity will be $\mathcal{O}(p^2 J^2 T)$ for the the pseudo-inverse step and $\mathcal{O}(TpJp) = \mathcal{O}(Tp^2 J)$. Hence, for each iteration $\mathcal{O}(p^2 J^2 T)$ and overall $\mathcal{O}(Mp^2 J^2 T)$.

Hence, assuming $K > p$, the overall complexity of the dLDS direct LINOCS extension would be $\mathcal{O}(MTJ^2 p * max(p, K)) = \mathcal{O}(MTJ^2 pk)$.

## C.3  Locally Linear System (Time Invariant)

Assuming we limit the number of iterations to $M$. For each iteration $m = 1 \ldots M$, we iterate over $t = 1 \ldots T$ time points to infer $\boldsymbol{A}_t$. For each $\boldsymbol{A}_t$ we consider $K + 1$ windows ($k = 0 \ldots K$) with $k_i = 1 \ldots k + 1$ shifts. Hence, under each combination of time $t$ and iteration $m$, we get an overall

$$K^* = \sum_{k=0}^{K} k + 1 = \frac{(K+1)(1+K+1)}{2} = \frac{(K+1)(K+2)}{2}$$

equations for each $\boldsymbol{A}_t$. For simplicity, we will neglect the weights $w_k$ in the following computational complexity analysis since they do not add complexity and do not nee to be considered here.

Let:

- $\boldsymbol{x}^+ := \text{diag}\big(\{\{\boldsymbol{x}_{t+k-k_i-1}\}_{k_i=1}^{k+1}\}_{k=0}^{K} \in \mathbb{R}^{pK^* \times K^*}\big)$ (each $\boldsymbol{x}_\tau \in \mathbb{R}^{p \times 1}$).

- $\boldsymbol{x}^- := \text{diag}\big(\{\{\boldsymbol{x}_{t-k_i}\}_{k_i=1}^{k+1}\}_{k=0}^{K} \in \mathbb{R}^{pK^* \times K^*}\big)$.

- $\boldsymbol{C} \in \mathbb{R}^{pK^* \times p}$ be a vertical concatenation of the set: $\{\{\Pi_{j=1}^{k-k_i} \widehat{A}_{t-k_i+k-j}\}_{k_i=1}^{k+1}\}_{k=0}^{K}$.

- $\boldsymbol{B} \in \mathbb{R}^{p \times pK^*}$ be an horizontal concatenation of the set: $\{\{\Pi_{j=1}^{k_i} \widehat{A}_{t-j}\}_{k_i=1}^{k+1}\}_{k=0}^{K}$.

- $\boldsymbol{\Psi} := \boldsymbol{B} \boldsymbol{x}^- \in \mathbb{R}^{p \times K^*}$

Figure 12: Zoom in for the reconstruction of the linear experiment.

- $\widetilde{\boldsymbol{\Psi}} \in \mathbb{R}^{p \times (k+p)}$ be $[\boldsymbol{\Psi}, \boldsymbol{I}_{p \times p}]$, namely, an horizontal consternation of $\boldsymbol{\Psi}$ and $\boldsymbol{I}_{p \times p}$ where $\lambda$ is the weight of smoothness regularization on $\boldsymbol{A}_t$.

- $\widetilde{\boldsymbol{x}^+}$ be an horizontal consternation of $\boldsymbol{x}^+$ and $\lambda \mathbf{1}_{K^* \times 1} \otimes \boldsymbol{A}_{t-1}$ where $\otimes$ is the Kronecker product, $\mathbf{1}_{K^* \times 1}$ is a vector of ones with $K^*$ elements, and $\lambda \in \mathbb{R}^+$ is the weight of smoothness regularization on $\boldsymbol{A}_t$.

Each $\boldsymbol{A}_t$ can thus be solved via extended least squares:

$$\widehat{\boldsymbol{A}_t} = \arg\min_{\boldsymbol{A}_t} \|\boldsymbol{x}^+ - \boldsymbol{C}\boldsymbol{A}_t\boldsymbol{B}\boldsymbol{x}^-\|_F^2 + \lambda\|\boldsymbol{A}_t - \boldsymbol{A}_{t-1}\|_F^2$$
$$= \arg\min_{\boldsymbol{A}_t} \|\boldsymbol{x}^+ - \boldsymbol{C}\boldsymbol{A}_t\boldsymbol{\Psi}\|_F^2 + \lambda\|\boldsymbol{A}_t - \boldsymbol{A}_{t-1}\|_F^2$$
$$= \arg\min_{\boldsymbol{A}_t} \|\widetilde{\boldsymbol{x}^+} - \boldsymbol{C}\boldsymbol{A}_t\widetilde{\boldsymbol{\Psi}}\|_F^2$$

Assuming $K^* > p$, the overall process thus involves the following computational steps:

1. Calculating $\boldsymbol{\Psi}$: $\mathcal{O}(p^2(K^*)^2)$

2. Pseudo-Inverse of $\boldsymbol{\Psi}$: $\mathcal{O}((p^2(K^* + p)) = \mathcal{O}(p^2 K^*)$.

3. Pseudo Inverse of $\boldsymbol{C}$: $\mathcal{O}(p^2 K p) = \mathcal{O}(K p^3)$.

4. Matrix multiplication of $\text{pinv}(\boldsymbol{C})\boldsymbol{x}^+\text{pinv}(\boldsymbol{\Psi})$: $\mathcal{O}(p^2(K^*)^2)$.

Overall, the most computational complexity in each iteration and time step is $\mathcal{O}(p^2(K^*)^2)$ Since we have $T$ time points and assuming that we limit the number of iterations to $M$, the overall complexity would be:

$$\mathcal{O}(MTp^2K^2)$$

## C.4 Observation vs. Latent Spaces in Dynamical Systems

In many dynamical systems models, the observation space, denoted as $\mathcal{X} \subseteq \mathbb{R}^N$, contains the directly measured variables. For a system observed over time, we represent these observations as $\boldsymbol{x}_t \in \mathcal{X}$, where $t$ denotes the time index. The latent (low-dimensional) space, denoted as $\mathcal{Z} \subseteq \mathbb{R}^M$ with $M \leq N$, contains hidden variables that are not directly observed but inferred from the observations. Let $\boldsymbol{z}_t \in \mathcal{Z}$ represent the latent state at time $t$. The relationship between the observation space and the latent space is often modeled by an observation function $h : \mathcal{Z} \to \mathcal{X}$, such that:

$$\boldsymbol{x}_t = h(\boldsymbol{z}_t) + \epsilon_t$$

where $\epsilon_t \sim \mathcal{N}(0, \sigma^2 I)$ represents observation noise. Several models, including SLDS and dLDS, include such transition to a latent state to capture the underlying dynamics in a low-dimensional space. In this paper, by fixing the observation function to the identity operator, we in essence learn the dynamic evolution directly on the observations.

### C.5 More Information about Neural Data

The data we used was collected by the Rutishauser lab at Cedars-Sinai Medical Center Kyzar et al. (2024), with detailed descriptions in Kyzar et al. (2024); Kamiński et al. (2017; 2020). The dataset includes electro-physiological recordings from 21 epileptic patients who were implanted with depth electrodes and Behnke-Fried microwires in the human medial temporal lobe and medial frontal cortex. The recordings were obtained while the subjects performed a memory task ("Sternberg task"). In the Sternberg task, participants were required to memorize a set of 1–3 images. These images were pseudo-randomly chosen from a group of five that elicited the strongest selective responses during the screening task. Following a maintenance period, participants were shown a probe image and asked to identify whether it was included in the initial set. The images used in these tasks represented a broad array of subjects and complex natural scenes. Please see Kyzar et al. (2024) for more details.

We loaded the data from the DANDI Archive in an NWB (Neurodata Without Borders) format ( Rübel et al. (2022)), and used a single session from it. This session includes recordings of a 63-year-old male subject (Subject 10 in the data) recorded in June 2023 while performing the Sternberg task.

### C.6 Pre-processing to Neural Data

To process the data, we took the spike times of the $p = 74$ neurons within the chosen session. We then convolved the spike times with a 100-ms-width Gaussian kernel to get a firing rate estimation. We then normalized the data by dividing each neurons' estimated firing rate by its top 99% firing rate, and used it on the initial 850 samples.

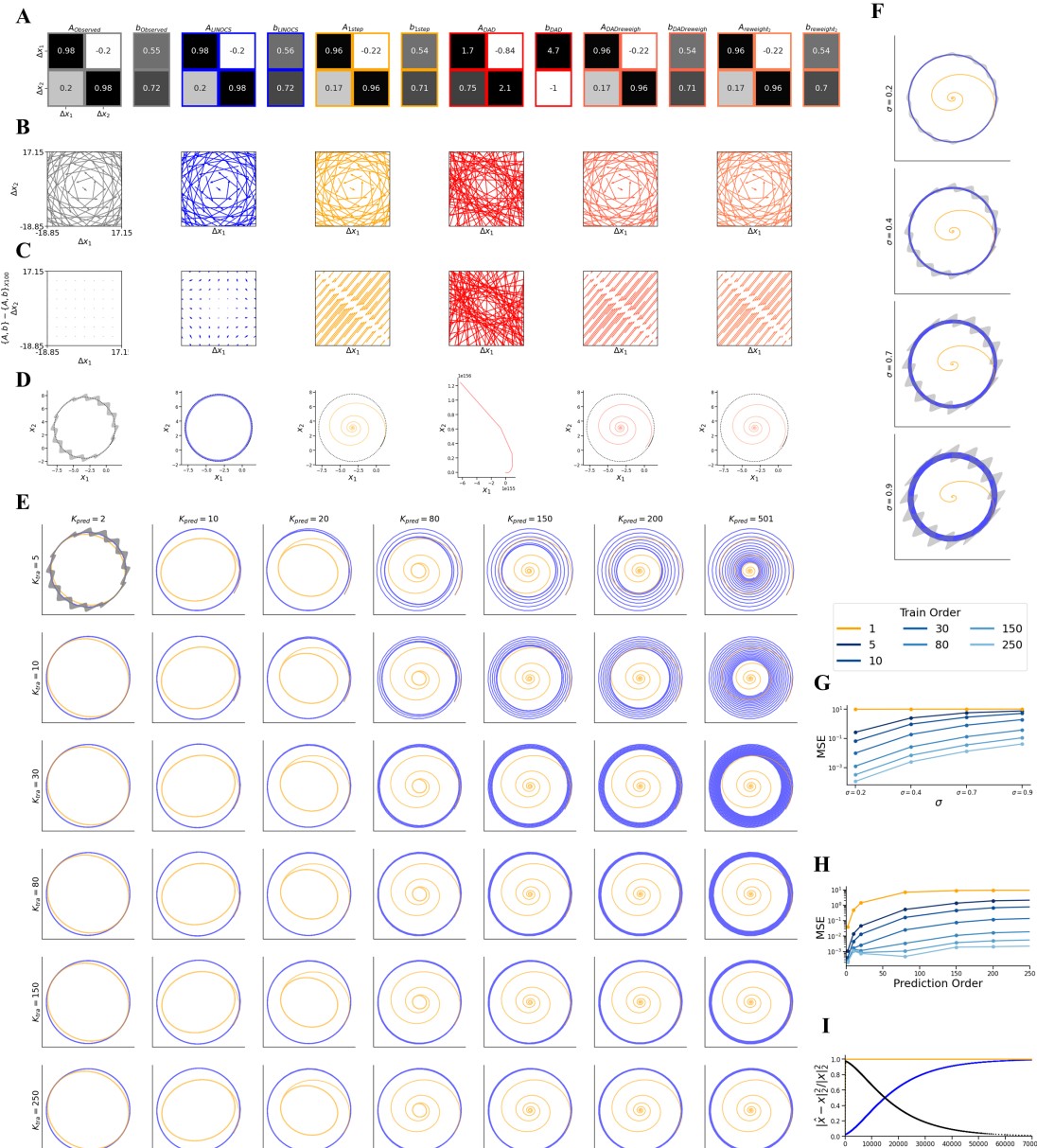

Figure 13: **Linear System under Structured Noise. A:** Real vs. identified operators and offsets. **B:** Quiver plots of real and identified operators (lower 2x2 sub-matrix, i.e., focusing on the rotational part of $\boldsymbol{x}_2$-$\boldsymbol{x}_3$) present patterns that appear similar, rendering it challenging to discern differences when examined in isolation. **C:** The differences in effects between real operators and inferred operators highlight how minor distinctions in dynamic operators gain prominence during lookahead reconstruction (calculation details in A.1). **D:** Full lookahead reconstruction (ground truth vs. baselines) shows swift convergence to the circle's center for the one-step optimization results due to small differences in dynamic values (mid-yellow subplot) and divergence for DAD-based results (three most-right subplots). **E, H:** MSE under increasing prediction orders. LINOCS achieves better (lower) MSE compared to 1-step optimization. **F, G:** LINOCS reconstruction compared to 1-step optimization under increasing noise values reveals that LINOCS maintains good reconstruction even under extreme noise conditions. **I:** By propagating identified operators until a relative reconstruction error of $\sim 1$, LINOCS enables future predictions of $\sim 70,000$ time points ($\sim 4280$ full rotations), contrasting with immediate convergence in one-step optimization. Black indicates error differences between one-step optimization and LINOCS.

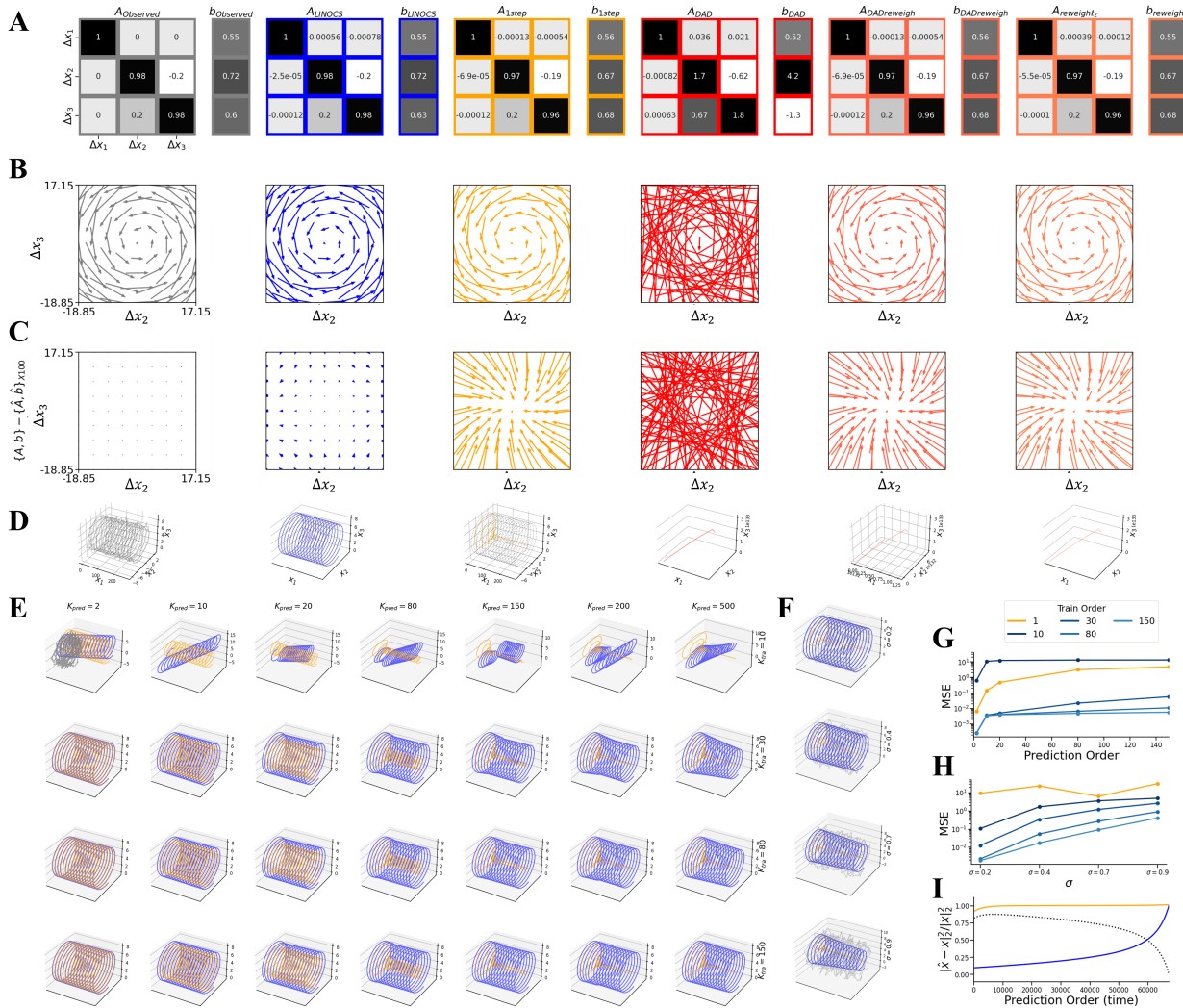

Figure 14: **Linear System for** $3D$ **Cylinder. A:** Real vs. identified operators and offsets. **B:** Quiver plots of real and identified operators present patterns that appear similar, rendering it challenging to discern differences when examined in isolation. **C:** The differences in effects between real operators and inferred operators highlight how minor distinctions in dynamic operators gain prominence during lookahead reconstruction (calculation details in A.1). **D:** Full lookahead reconstruction (ground truth operators vs. baselines) shows swift convergence to the cylinder's center for the 1-step optimization results due to small differences in dynamic values (yellow) and divergence for DAD-based results (three most-right subplots). **E, G:** MSE under increasing prediction orders. LINOCS achieves better (lower) MSE compared to 1-step optimization with perfect full lookahead reconstruction under high-enough training order (**E** right bottom). **F,H:** LINOCS reconstruction compared to 1-step optimization under increasing noise values reveals that LINOCS maintains good reconstruction even under extreme noise conditions. **I:** Propagating the identified operators until reaching a relative reconstruction error of $\sim 1$ shows that LINOCS identifies operators that enable a future prediction of $\sim 70{,}000$ time points ($\sim 2680$ full rotations) before converging, unlike one-step optimization that converges immediately. black: error difference between one-step optimization and LINOCS.

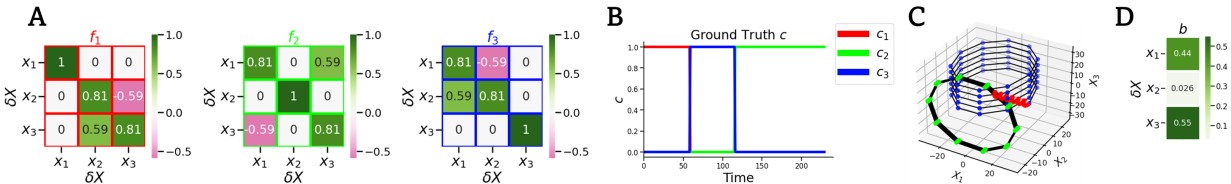

Figure 15: **Ground Truth operators and coefficients for the SLDS experiment. A:** The ground truth basis dynamics operators $\{\boldsymbol{f}_j\}_{j=1}^{J}$ consist of rotational matrices oriented in various directions. **B:** Ground truth operators' coefficients ($\boldsymbol{c}$). **C:** Ground truth state $\boldsymbol{x}$. **D:** offset applied to the operators.

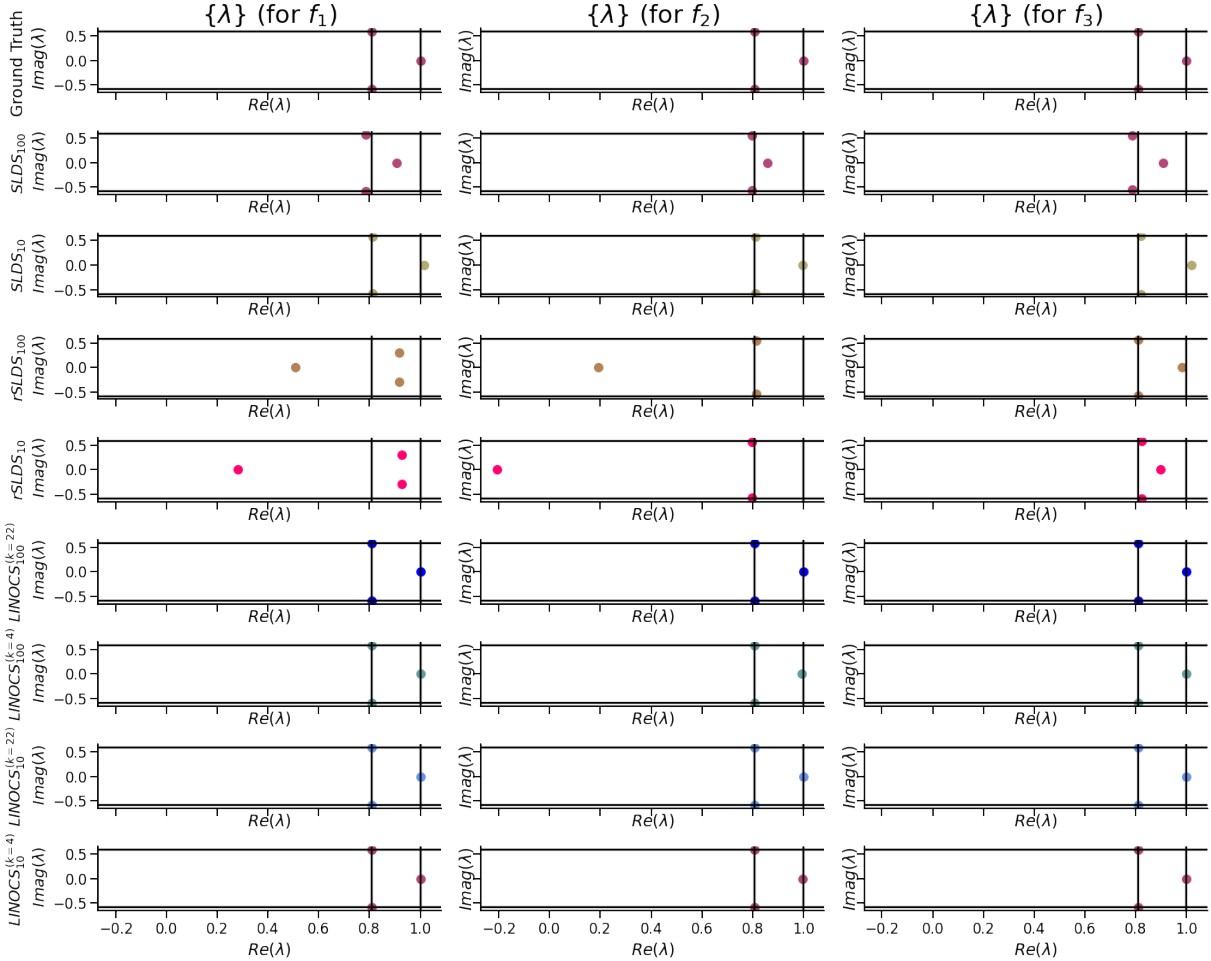

Figure 16: **Eigenvalues of identified operators by the different systems compared to the eigenvalues of the ground truth operators.** Rows represent different methods, with LINOCS (with different parameter combination) in the four last rows. Columns represent the three eigenvalues of each of the three different $3 \times 3$ linear operators. LINOCS enabled the identification of almost perfect eigenvalues while the other methods found at least one wrong eigenvalue per operator, explaining the decaying/divergence of their reconstruction.

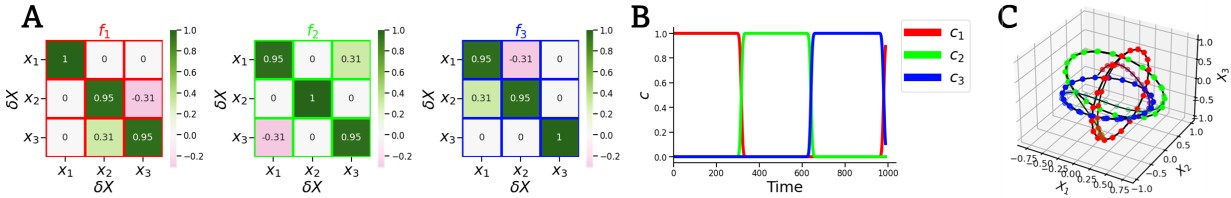

Figure 17: **Ground Truth operators and coefficients for the "pseudo-switching" dLDS experiment.** **A:** The ground truth basis dynamics operators $\{\boldsymbol{f}_j\}_{j=1}^J$ consist of rotational matrices oriented in various directions. **B:** Ground truth operators' coefficients ($\boldsymbol{c}$). **C:** Ground truth state $\boldsymbol{x}$. The colors of the markers are a time-changing re-weighted average of the active operators.

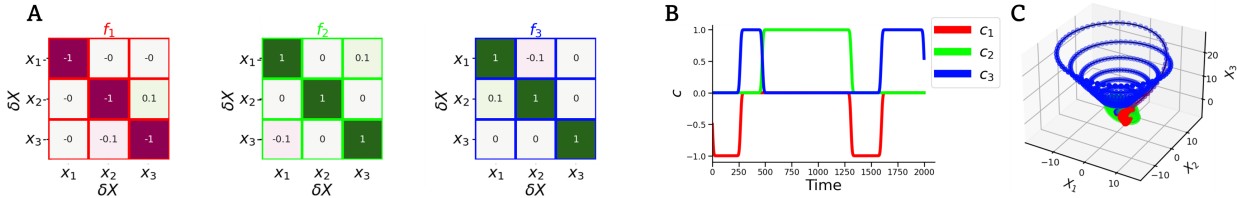

Figure 18: **Ground Truth operators and coefficients for the second dLDS experiment. A:** The ground truth basis dynamics operators $\{\boldsymbol{f}_j\}_{j=1}^J$ consist of rotational matrices oriented in various directions. **B:** Ground truth operators' coefficients ($\boldsymbol{c}$). **C:** Ground truth state $\boldsymbol{x}$.

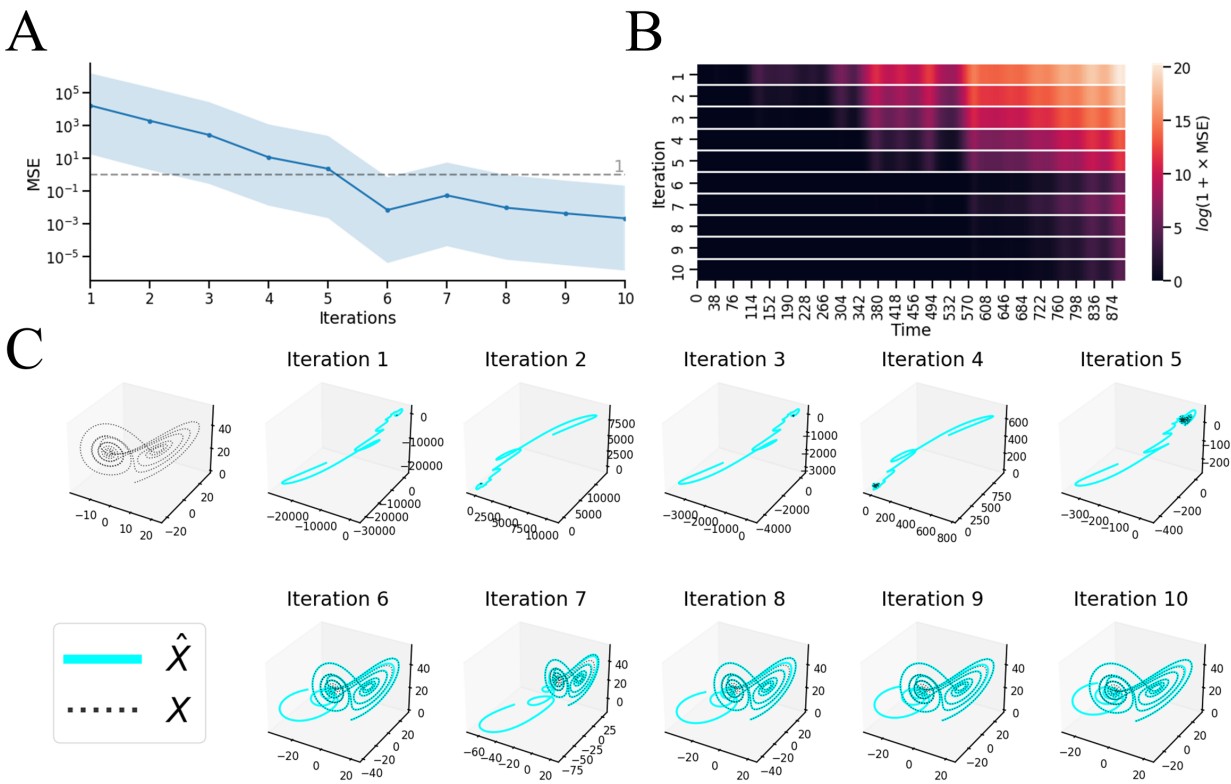

Figure 19: **Demonstration of LINOCS' effect over iterations for Lorenz attractor with 900 time points ($K_{\mathbf{train}} = 19$).** **A:** MSE over iterations (curve corresponds to median values; shade represents 25%-75% percentiles over time). **B:** MSE over time-points (horizontal) and training iterations (vertical). **C:** Full lookahead reconstruction based on operators identified under different iterations. Dotted black: (as exemplified in the top left) ground truth; Solid Cyan: Full lookahead LINOCS reconstruction.

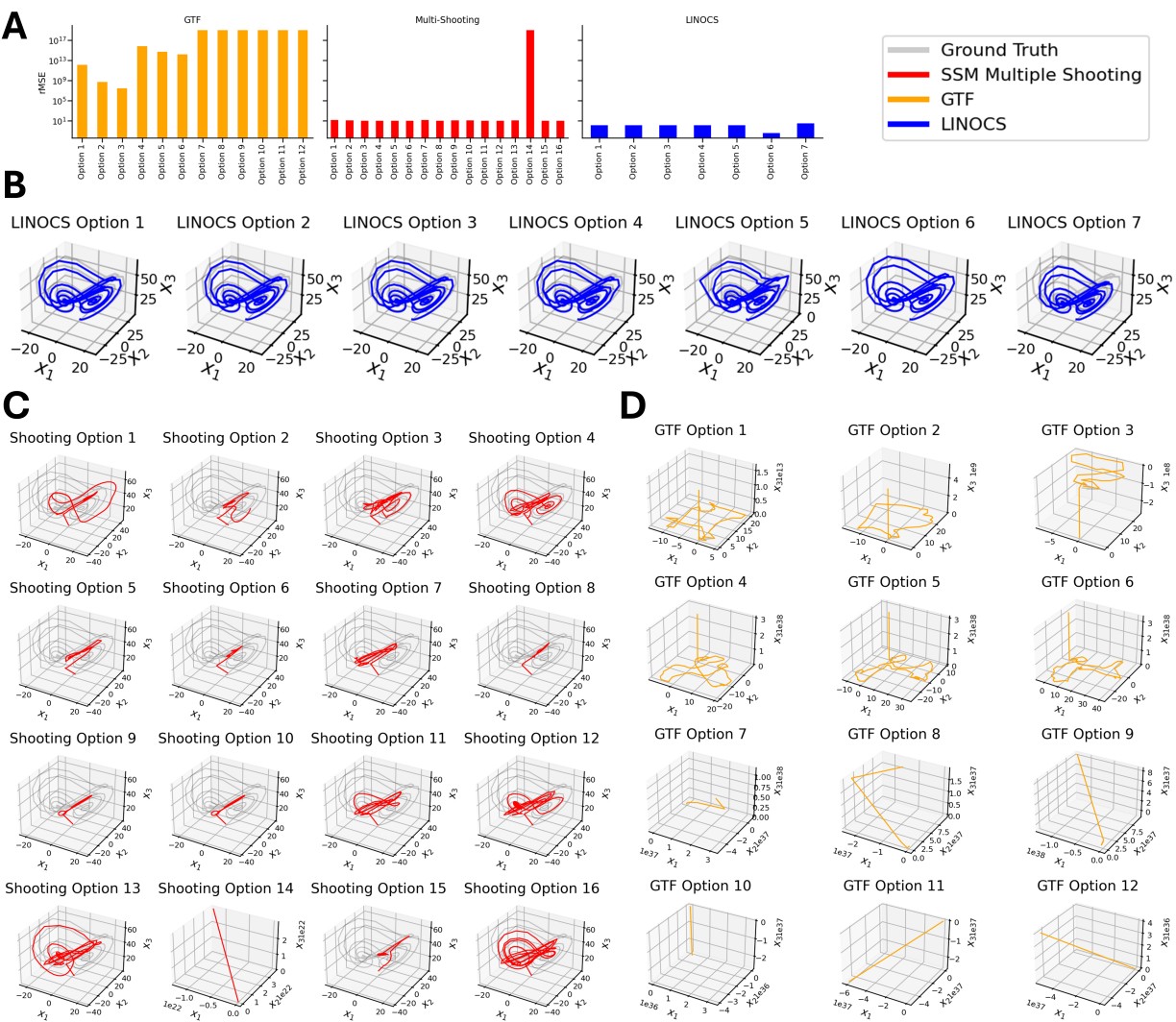

Figure 20: **Comparison of LINOCS Lorenz locally linear time-varying results to other multi-step approaches. A:** rMSE of full lookahead reconstruction by LINOCS (blue) compared to the other baselines (Jordana et al., 2021) (Shooting, red), and (Hess et al., 2023) (GTF, orange). **B:** LINOCS full lookahead reconstruction under different hyper-parameter settings, with the options described in Table 10. **C:** Shooting (Jordana et al., 2021) full lookahead reconstruction under different hyper-parameter settings, with the options described in Table 9. **D:** GTF (Hess et al., 2023) full lookahead reconstruction under different hyper-parameter settings, with the options described in Tables 7 and 8.

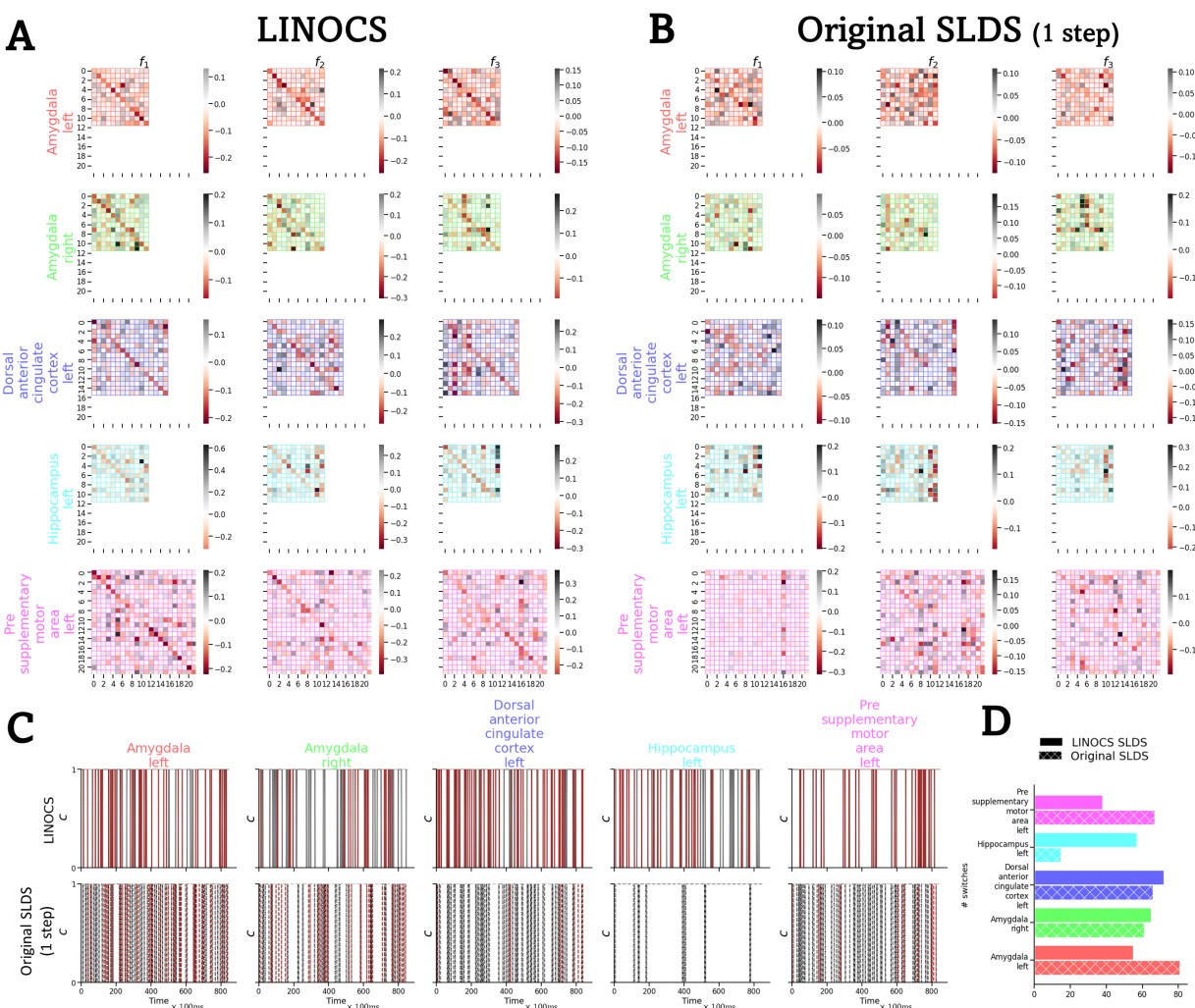

Figure 21: **Operators and reconstructions by the LINOCS-driven SLDS compared to classical SLDS. A:** Dynamical operators identified by LINOCS-driven SLDS. **B:** Full lookahead reconstruction.

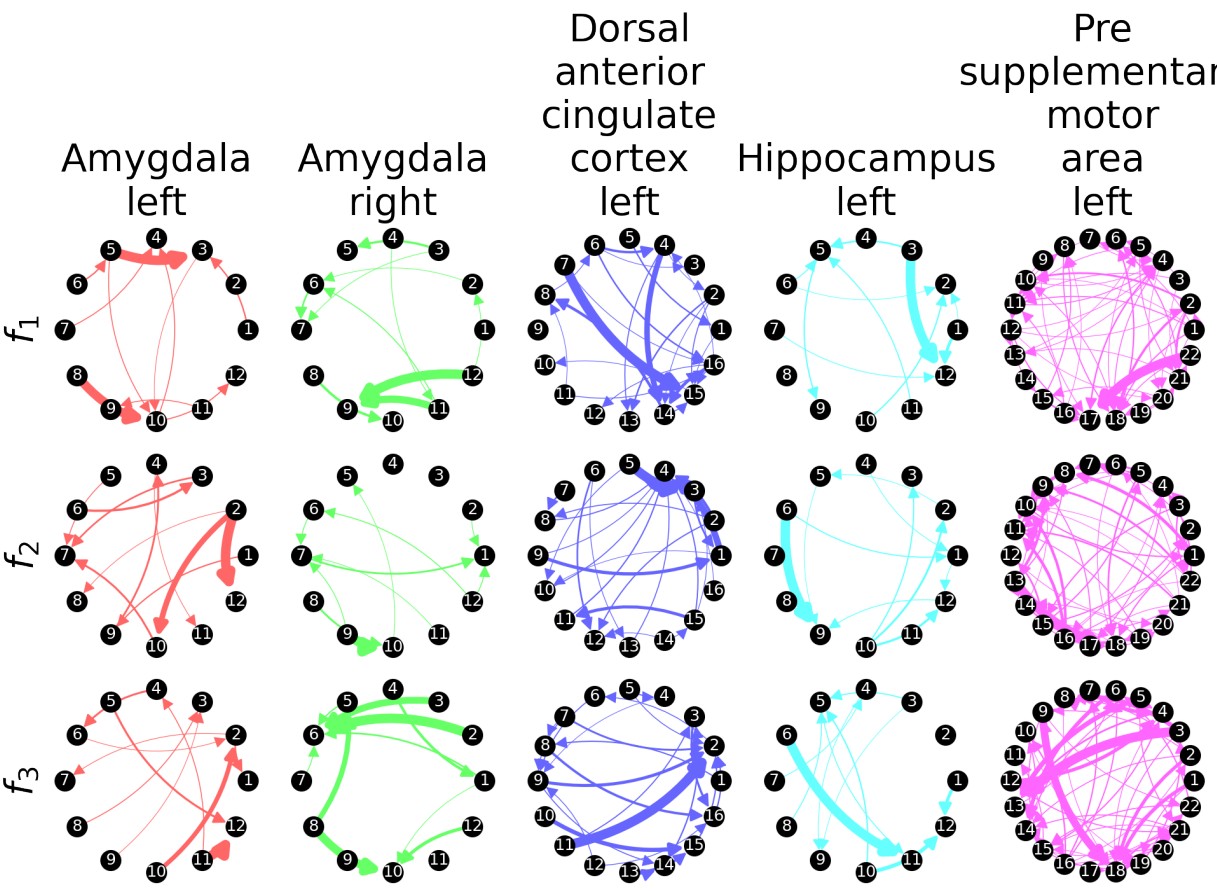

Figure 22: **Classical SLDS results (non LINOCS) on the real world data**. The identified networks ($\{\boldsymbol{f}_j\}_{j=1}^{J}$) by the non-LINOCS SLDS code (Linderman et al., 2020), for each region in the real world data (Kyzar et al., 2024).

Figure 23: **SLDS results on real data. A:** Identified operators per region by LINOCS. **B:** Identified operators per region by classical SLDS. **C:** Switch times by LINOCS-SLDS vs classical SLDS. **D:** Number of switches for LINOCS-SLDS vs. classical SLDS.

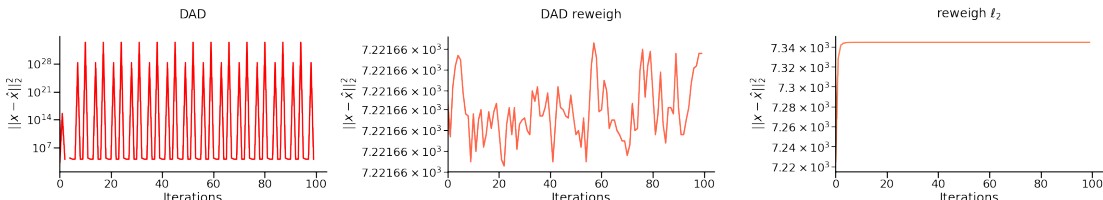

Figure 24: Lookahead prediction error ($\|\boldsymbol{x} - \widehat{\boldsymbol{x}}\|_2^2$) of DAD-algorithms over training iterations (for the white noise linear experiment).

