# OpenReview forum: "LINOCS: Lookahead Inference of Networked Operators for Continuous Stability"
_TMLR — Accepted by TMLR_

### Review · Reviewer_GGPj · 2024-06-08

**Summary Of Contributions:**

The paper introduces a learning procedure, LINOCS, for linear dynamical systems to produce accurate long-term predictions and identify interactions in noisy time-series data. LINOCS integrates multiple-step predictions with adaptive weights to accurately capture long-term trends. The authors demonstrate the effectiveness of LINOCS in recovering ground-truth dynamic operators in various linear dynamical system models, including time-invariant linear, switching linear, decomposed linear, and regularized linear systems. They also demonstrated the application on real-world neural recordings, identifying meaningful interactions between multiple regions.

**Audience:**

Yes

**Broader Impact Concerns:**

No broader impact concerns.

**Claims And Evidence:**

Yes

**Requested Changes:**

- discussion and citations to existing methods with similar ideas
- presentation of computational cost at different orders
- compare or discuss with other methods aiming for long-term predictions, e.g. generalized teacher forcing or any other methods to the authors knowledge.

**Strengths And Weaknesses:**

Strengths
- Introducing multi-step predictions cost improves long-term reconstruction
- LINOCS is applicable to various existing dynamical system models
- LDS with accurate system identification produce reliable and interpretable interaction between state dimensions (element)
- The real-world neural recoding example demonstrates the practicality of LINOCS.

Weaknesses
- The scope is limited to linear systems. Though the motivation of LDS is partially interpretability, nonlinear systems would also benefit from multiple-step cost. In addition, the expressive power of LDS is limited to such as fixed points and spiral.
- The idea of multiple-step prediction was also used by other studies, e.g. Hocker 2017 (10.1109/NER.2017.8008426).
- Lack of complexity analysis, i.e., how much more time the multiple-step prediction costs the training.
- Lack of comparison or discussion with other methods aiming for long-term predictions.

---

> ### Author Response · Authors · 2024-08-05
> **Response to reviewer GGPj**
>
> We thank the reviewer for their helpful feedback and for recognizing the practicality and applicability of LINOCS.
> We address the specific points below:
>
> 1) **The reviewer was concerned that the scope is limited to linear systems with limited expressive power.**
>
> We appreciate the opportunity to discuss this point. Our demonstrations focused on locally linear approximations to dynamical systems that can be nonstationary and thereby also represent nonlinear systems that may present more complex structures than, e.g., fixed points and spirals. Particularly, as discussed in the “Background and Terminology” section, “Our focus on locally linear dynamics is supported by the fact that even highly nonlinear functions can be well approximated over small time intervals using local linearization”.
> Given that this problem has not been completely solved even for locally linear models, and considering the widespread usage and need for such models, we believe it is crucial to first focus on this simpler locally linear case. Additionally, we found it important to demonstrate LINOCS on switching and decomposed architectures, as these are prevalent in neural dynamic models.
> To isolate the effect of LINOCS on Switching and Decomposed dynamics, we aimed to preserve the original architectures of SLDS and dLDS (which are time-changing locally linear) without introducing additional (nonlinear) transformations. However, we agree that incorporating a nonlinear local transformation would be an exciting future direction, which we’ve added to the future work section in our updated PDF, and believe that implementing such functions should be a straightforward extension.
>
> 2) **The reviewer was concerned about the need to compare the model to additional baselines:**
>
> Our comparisons focused on benchmarking to the most relevant current methods we knew of, specifically variations of DAD and DAgger models for the linear case, non-LINOCS SLDS and non-LINOCS rSLDS for the switching case, and non-LINOCS dLDS for the decomposed case. The paper that the reviewer mentioned (Hocker and Park (2017)) is indeed relevant, and we have added a discussion of it to the related work section. However, a direct comparison to their method is less relevant because their focus on a Poisson-GLM model differs from the dynamical models we demonstrated LINOCS on in our paper. While applying LINOCS to such a Poisson architecture could be an interesting direction for future research, it falls beyond the scope of this paper. Additionally, the absence of shared code for their multistep Poisson-GLM poses a challenge for such a comparative analysis.
>
> Yet, we understand the need for additional comparisons with multi-step methods. Hence, we have added comparisons of LINOCS to two new multi-step dynamic approaches [1,2] under varying hyperparameters in the updated PDF for the locally time-varying dynamic model.
>
> [1] Florian Hess, et al.  ``Generalized teacher forcing for learning chaotic dynamics''. ICML 2023.
>
> [2] Armand Jordana, et al. ``Learning dynamical systems from noisy sensor measurements using multiple shooting''. arXiv preprint arXiv:2106.11712, 2021.
>
>
> 3) **The reviewer asked to add complexity analysis:**
>
> We agree that a complexity analysis would benefit this work. Hence, we have added this analysis to the appendix in the updated PDF.
>
>
> 4) **The reviewer suggested comparing LINOCS with more long-term prediction methods:**
>
> In addition to the existing baselines we included, we have added a discussion of Generalized Teacher Forcing (GTF, Hess et al. (2023)) as well as of Hocker and Park (2017) that was mentioned before (Sec. 3, “Prior Relevant Approaches”). As for GTF, we point out that their method is tailored towards training RNNs with stationary or piecewise transition functions and thus does not support non-stationary dynamics like time-changing decomposed systems.
> Additionally, their method does not currently support constraints (e.g. sparsity) on the transition matrix which may be required in certain applications.
> Moreover, GTF relies on adjusting the desired Jacobian matrix, which requires either pre-defined knowledge of the Jacobian or its estimation. GTF also requires estimating the ideal forcing parameter α which may strongly affect the results.
>
> In the updated paper, we added a comparison of LINOCS to the GTF model with varying parameter choices; see Figure 8 and the options for parameter choice for GTF in Tables 7 and 8.

---

> > ### Comment · Reviewer_GGPj · 2024-08-12
> >
> > Thank the authors for the responses. My concerns are addressed. I appreciate the efforts.
> >
> > I had no doubt that the locally linear dynamics would produce good prediction empirically. Yet, I believed it's valuable for readers to understand why the scope is limited to linear systems, even though it may not seem necessary.
> >
> > I did not ask for a comparison (with any metric) with Hocker 2017. That was to point out the idea of multistep-prediction loss had been used for improving long-term prediction in a general way since the application in neuroscience was mentioned in the paper. The additional discussion is helpful.
> >
> > The discussion of GTF is helpful. Though I'm not sure why GTF looks too bad in Fig.8 comparing to Fig S4 in Hess 2023, I will not ask for more.

---

### Review · Reviewer_MKNA · 2024-06-16

**Summary Of Contributions:**

The authors present a new method, LINOCS, to analyze and model non-stationary and non-linear dynamical systems. They demonstrate their method on a variety of simulations, which outperforms previous methods in prediction accuracy. Particularly, it focuses on combining multiple multi-step predictions to create robust estimates of the states, allowing for robust modeling and forecasting. They further demonstrate it's utility on a published neural dataset, where it identifies possible underlying neural connections.

**Audience:**

Yes

**Claims And Evidence:**

Yes

**Requested Changes:**

Critical:
1. Add citations to "common" methods. For example, there is a whole host of methods, such as DMD (OPT-DMD, BOP-DMD, NS-DMD, MR-DMD, etc.), TVART, PLSO, to name a few.
2. Background "(e.g., if $x_t | x_{t−1}$ has a variance of $v_t$, then $x_t|x_{t−K}$ will have a variance of $\sum_{k=0}^{K-1} v_{t-k}$ due to the property of variance summation)." Doesn't this assume that all samples are independent from the previous ones? In which case this isn't quite right.
3. Sec 2.1, Mention SLDS and others, but this should be cited
4. Sec 2.1.1 "...Koopman operator for improved long-term prediction. However, these methods often require extensive learning data and remain uninterpretable in the “network” sense, as non-linear state measurements in Koopman operators can obscure understanding of pairwise state interactions." Koopman operator methods (e.g. something like data-driven DMD approaches) specifically target finding interpretability and understanding of pairwise state interactions. This should be clarified.
5. Sec 2.1.1 "...such representation does not provide explicit insight into the time-changing interactions between the state elements." It does if you choose the correct equations, but this is hard to do. Again, this should be clarified.
6. Sec 2.1: "...“network” interpretability. Expand upon this. What does network interpretability mean here?
7. Sec 2.1: "...observation space, essentially enforcing the transition to the latent space to be the identity operator." First time talking about observation and latent spaces; should expand on what these are.
8. Fig 1. Is this a cartoon? Or how is this generated?
9. Sec 3. "[LINOCS]...yields not only a more accurate..." This model is "better" compared to what?
10. "Specifically, it gradually increases the weight of the best lookahead reconstruction until convergence conditions are satisfied." From the text, it looks like $w$ is being learned while simultaneously finding A? Please add an algorithm to help aid with these descriptions.
11. Fig 1. Could use more description. E.g. what are x1 and x2? Which models are you using that perform well on 1-step predictions? B, An algorithm approach could be extremely helpful to explain this. C, What is being shown? What is "Iter?"
12. Eq 9: When do you update c vs f vs w (if included)? Since $F_{all}$ depends on $c_t$, while $c_t$ depends on $F_{all}$? An algorithm may be useful here too.
13. Figures are extremely small, such as figures 4, 7, 11, 12, etc. These should either be enlarged or condensed for better readability. E.g. in Fig 7A, it's extremely hard to tell by eye if one trace is better than another.
14. Sec 4.3 "This underscores the necessity of multi-step reconstructions for accurately estimating dynamical system dynamics." This just shows that this method works while some others don't, but this doesn't prove that multi-step reconstructions are "necessary." Might want to reword.
15. Sec 4.4 "We compared LINOCS-LTV with several other LTV solvers with varying constraints, including smoothness and sparsity." These should be listed.
16. Sec 4.5 Please provide a little bit more description of the task for the monkeys. E.g. "screening task?" Also, the results presented should be task aligned; what alignment are you using? E.g. in Fig 8, when does this timing occur? Is there more data that is excluded from analysis?
17. Sec 4.5 "..this suggests that LINOCS may better capture the linear approximation of brain activity than the common 1-step optimization." This is really big claim without the necessary evidence to back it up. Ideally, would want to compare to standard neuro methods and show that your method can recover standard results (e.g. from granger causality) while something like 1-step optimization cannot. Since this dataset was already published, is there some result that can be recovered?
18. Fig 8A: Might want to make it clear in the text that each node is a neuron. And how does the region to region connectivity work? Is it averaged across neurons? Is there any normalization applied?
19. "LINOCS not only achieves more precise full lookahead reconstruction compared to baseline methods but also successfully retrieves ground truth operators in synthetic data..." This seems to imply that other methods do NOT recover ground truth operators, which isn't necessarily true. Might want to rewrite this to clarify this point.
20. "[LINOCS]...may help mitigate issues such as vanishing or exploding gradients." This is a fairly big claim, what evidence is there to support this so far?
21. Fig 16/17/19: Given that all three rotational matrices (all in different directions) occur, shouldn't there be THREE "rings"?  Why are there only two? As a slightly more general test, the three matrices could be rotations in three non-orthogonal directions, to show that the transitions are found regardless if there's an overlap in transition matrices.
22. Fig 20: What is the prediction and training order?
23. The following items should be added/discussed in the discussion section:
24. "Additionally, our framework allows custom weight functions that suit their specific needs. In the experiments presented in this paper, we concentrate on showcasing three specific options for the weights..." How does one choose a weight function?
25. You mention 4 different methods in Sec 3. (pure linear, switching linear, etc.). How does one choose which method to choose? What happens if you choose a more general method on a simpler case?
26. For the neuro applications, which method should be used? You applied all four methods, but does one seem like the most appropriate for neuroscience? Does each one provide different information?
27.For the SLDS like methods and applications, how does one choose the number of modes? Especially in the neural application, wouldn't this number affect results?
28. How does one choose hyperparameters in general. For example, how should one choose K?

To strengthen work:
1. The term "often" (and other similar terms) is overused and citations should be added to help the reader. E.g. Intro P2: "often rely on either “black-box." Intro P2: "often result in undesired divergence away from the system." Background P6 "they often struggle to fully reconstruct the dynamics over longer time-scales."
2. Intro "...in cases where the dynamics are non-stationary, non-linear, or otherwise constrained in ways that reflect real-world system behavior." What other ways exist?
3. Intro "...achieving significantly improved accuracy in operator identification and long-term predictions." Should say compared to what. Also, it may help to reference later sections (e.g. Sec. X) so one can easily find this information.
4. Sec 2.1: "One common method for determining A (and b if it is assumed that an unknown offset exists)." In a lot of methods (e.g. PCA), the data is normalized such that "b" doesn't exist. Could say something about this decision. Also, this was referred to earlier in the manuscript as a "forcing term," but now this is just an offset?
5. Eq 6. Is $w_k$ normalized?
6. Fig 2. Far right panel, do the coefficients have to be roughly constant? Or can they fluctuate in more general forms?
7. "...however LINOCS can be easily adjusted to other noise statistics." How?
8. Eq in Sec. 3.1: A is raised to an exponent right? Not an index, could be helpful to specify this.
9. Sec 3.2: SSM framework? And this could use a little bit more description. This first uses SLDS to find the boundaries of linear periods, and only then finds the linear dynamics? Can you summarize the method here? How do you deal with Kth order predictions, which may extend past the boundary?
10. Comment on Fig 3: not to add another model, but OPT-DMD should also work extremely well here.
11. Fig 3D right 3 panels: Can you interpret this? From Fig 3B, I would expect the values to still "rotate." Why does it diverge?
12. Fig 3E: What is theoretically best considering the level of noise?
13. Sec 4.1: you mention using DAD with 100 iterations. Is this enough to converge? What would happen if you tested with more iterations?
14. Comment on the statement: "...instead decay to zero (away from the real system)." In fact, this is known in DMD (Dawson 2016), and is fixed with OPT-DMD.
15. Sec 4.1: "...approximately 35,000 time points." What is the inherent time scale? E.g. how many rotations is this for in the plots, or what is delta t / the sampling rate?
16. "We further tested LINOCS on linear systems with structured noise (Fig. 12)." This may be better included as SI material instead of the main text? Also, the cylinder case. What about using a much larger system? E.g. around 100 dimensions?
17. Fig 4:  It looks like SLDS and rSLDS improve with training epochs. If training for much longer, would they find the right transitions? And if they do find the right transitions, how do the algorithm speeds compare?
18. End of section 4.2. This is the first point when the importance of the eigenvalue spectrum is mentioned, which feels out of place at this point. Could specify that this is an objective earlier?
19. Sec 4.3 "Training with orders...resulted in high-quality full reconstruction". While impressive, I'm not sure this is a "perfect" full reconstruction. May want to reword.
20. "In addition, we analyzed operators identified across various training iterations of LINOCS to assess their proficiency in achieving lookahead reconstruction (Fig. 20)." What are the prediction and training orders? Hard to tell how impressive the result is without this information.
21. Fig 6H, which prediction order is used? Also, a similar plot would be interesting for looking at predictions when switches occur, since a switch could be impossible to predict. Or, perhaps a plot of the prediction order vs training order matrix. I would expect to see the error increase with lower training orders and larger prediction orders.
22. Fig 6A, Why does K pred of 50 look worse than 800 or 1000?
23. Fig 8C: Why does the 1-step approach fail to capture the average activity? Shouldn't this be an area where the 1-step approach works well? Also, typically in neuro applications, the data is normalized; how does that affect things here?
24. Fig 9C: Doesn't this figure say that SLDS does well during parts of the time period? In which case, how do the results compare during these times?
25. Fig 11: What new information is D telling us?
26. Fig 18: Is there a different way to show the same info? This doesn't seem accurate by eye, but how good is it actually?
27. Fig 22: What does this plot tell us in terms of the differences between classical SLDS and LINOCS SLDS? E.g. what does the number of switches tell us?
28. Possible discussion items:
29. For the SLDS model, why does your setup lead to the correct boundary transitions in Fig 4 while SLDS does not? I would assume that for a simple setup this should be doable with SLDS.
30.The simulated cases here are very small. What would happen for large matricies, which is fairly common for a lot of systems?

Typos or small fixes:
1. Sec 2.1: typo: Switching, not switched.
2. HMM should be expanded at least once and cited.
3. In descriptions of methods (3.1, 3.2...) it could be nice to reference associated figures. This is done for 3.2, but not any others.
4. Fig 3A: Titles have "F" and "b," but shouldn't "F" be "A" according to the text?
5. Fig 3A-D: Ylabels are confusing (Values? Quiver? y hat is never used?).
6. Fig 3F/I are not referenced in the text and the colors aren't labeled.
7. Fig 3 text "...increasing noise levels..." put sigma here.
8. First eq in 3.4: Write out the regularization term at least once.
9. "...The operators are initialized with a regularized 1-step optimization (Fig 3 cyan)..." Cyan should be orange I think? Also, check which figures are referenced to which others.
10. "...even under very high sigma noise levels..." Sigma should be labeled inside the Gaussian noise definition? Otherwise, this is the first time seeing this.
11. Sec 4.1 "...even extreme Gaussian noise levels, sigma = 0.9 (Fig. 3D, blue)." This should be Fig 3H I think
12. Sec 4.2 "Additionally, the offset for each state was set to a random..." This offset is not shown in Fig 16.
13. Sec 4.3 "...the high-order LINOCS-dLDS achieved a structure much closer to the ground truth (Fig. 18)." There is nothing but LINOCS? So there is no comparison?
14. Fig 4: Mention that correlation is rho in plots vector
15. Fig 5: A: Specify in the figure itself for which ones are made by non-LINOCS and LINOCS? B: The colors could be aligned with other figures.
16. Fig 7: What is lambda? Some hyperparameter? Also should specify somewhere how l0 regularization is implemented. Also, maybe use a log scale or something else for B? With the current scale, it's hard to tell how much better one is compared to another (e.g. the red ones).
17. Fig 8: units for time. Also, it might make sense to align the patterns between B and C (e.g. LINOCS be solid for both instead of solid for B and dashed for C).
18. Fig 9C: Maybe do logscale? Can't tell how good or bad the solid line is. Also "(full curve)" should be "solid curve."
19. Fig 10B: Colors are hard to distinguish. Units needed.
20. Fig 10D: The real and imag part should be on different plots given that the scale is completely different. Should also say lambda in the text for easier readability.
21. Fig 11: A: wasn't "w" called "lambda" before? Either way, labels should be consistent. B/E, consider fixing the color scale across plots.
22. App B was previously said in the main text and doesn't add new information.
23. Fig 12: Similar comments as fig 4. Also, it may help to order the plots the same way.
24. Fig 13: Similar comments to fig 4. Some numbers don't fit in their respective boxes. Also, for D, x1 / x2 (which is shown as the circular bit in B), is not the circular bit here; so the axes are not consistent.
25. Fig 16: "b" is not shown. Inconsistent axes (x,y,z notation for C and $x_0$,$x_1$,$x_2$ notation for A). Time is cutoff in B. It would be handy in C to show the temporal evolution, to see the initial conditions and how it evolves. Initial conditions are not specified.
26. Fig 18: What do the colors mean? And in the text you mention a comparison, but it doesn't look like the comparison is there.
27. Fig 20: Ground truth doesn't exist, despite being labeled?
28. Fig 22: White space could be avoided. Also (C), isn't SLDS used, not 1 step?

**Strengths And Weaknesses:**

Strengths:\
They provide a whole host of evidence and figures demonstrating the utility of their method. They apply their method on a wide variety of simulations as well as one real dataset. They provide code for the community to recreate results.

Weaknesses:\
Could be more concise. The figures especially are quite small and hard to read. It is also lacking in some references. Specifically, there is a whole host of methods analyzing dynamical systems: DMD (OPT-DMD, BOP-DMD, NS-DMD, MR-DMD, etc.), TVART, PLSO, to name a few.

---

> ### Author Response · Authors · 2024-08-05
> **Response number 1 to Reviewer MKNA**
>
> We deeply thank the reviewer for highlighting LINOCS’s capabilities and acknowledging our thorough simulations. We greatly appreciate the reviewer’s detailed and nuanced feedback and address each point below.
>
> 1) **The reviewer recommended discussing additional methods including DMD, TVART, and PLSOV:**
> Thank you and we agree that these works should be discussed. We have added a discussion of these additional methods, including DMD and its variants, TVART, and PLSOV (Section 3), and further discuss also HMMs. Additionally, we include in the future work section that LINOCS can be extended to these models as well.
>
> 2) **The reviewer was concerned by the increase in variance as mentioned in the background:**
> We thank the reviewer for raising this point. We now further clarified this point in the text (see updated PDF).
>
> To summarize:
>
> This statement refers to the conditional variance, i.e., the variance of each state given the previous state.
> For example, in the linear case, let:
>
> $\tilde{x}_t$ =
>
> $A \tilde{x}_{t-1} + \epsilon$
>
> where
> $ \epsilon \sim \mathcal{N}(0, \sigma^2 I) $.
>
>
> Then:
>
> $\tilde{x}_{t+1}$ =
>
>  $ A^2 \tilde{x}_{t-1} + A \epsilon + \epsilon$
>
> meaning the new noise term is
> ${A}\epsilon + \epsilon \sim \mathcal{N}(0, \sigma^2 ({A}{A}^T + I)) $.
>
> Hence, for the prediction of multiple (>2) time-steps into the future, the accumulated noise variance over $K$ steps results in:
> $\sum_{k=0}^{K-1} \sigma^2 {A}^k ({A}^T)^k.$ which, while depends on the eigenvalues of ${A}$, is a monotonically increasing function of the noise variance.
>
> Similar ideas hold for the nonstationary linear dynamical systems, while also accounting for the time index of ${A}$.
>
> To clarify this point in the paper, we applied the following changes:
> - We changed the notation ${x}$ to the noisy observations  $\tilde{x}$ in the sentence mentioned by the reviewer.
> - We added a more detailed explanation about the above (see the Background section).
>
>
> 3) **The reviewer raised concerns regarding the mention of SLDS and other methods in Section 2.1 without proper citation:**
> We would like to draw the reviewer’s attention to the multiple variants of SLDS discussed in Section
> 2.1 (now Section 4 in the updated paper, under “Switching Linear Dynamical Systems (SLDS)"). In this section,
> we explore various variants of the model and cite the pertinent literature. If the reviewer’s query
> pertains to the introductory part of this section, where we enumerate different types of systems,
> we have not included citations there because we were referring to the general switching concept rather than a
> specific learning algorithm/implementation. However, we agree with the reviewer and to reduce misunderstanding in the updated manuscript,
> we have updated this section to include these references to ensure all discussions are adequately
> supported with citations.
>
>
> 4) **The reviewer asked for clarification regarding “network interpretability'' and interpretability of other
> methods we referred to (e.g.,methods based on the Koopman operator):**
> We appreciate the feedback and clarified our use of the term “network interpretability'' in the
> paper (see Section 2, third paragraph, in the updated paper).
> In our work, we focus on the aspect of interpretability
> that involves understanding the interactions between individual state elements in the observation
> space. In this context, we refer specifically to the influence of each element $i$ on element $j$ in
> the next time-step. This interpretation of network interpretability thus focuses on direct, pairwise
> state interactions without the effects introduced by first transforming the data through non-linear or
> multivariate functions.
>
> However, Koopman theory and some other methods, find the linear dynamics
> underlying observed variables **after subjecting them to a (potentially) nonlinear transformation**, which may
> obscure the discovery of pairwise interactions between the observed units themselves.
> Importantly, we do not claim that these methods are less interpretable in the general sense, but
> specifically in the aspect of pairwise interactions.
>
> Notably, beyond this interpretability gap, LINOCS addresses a different aspect in dynamical system modeling
> than the Koopman operator. Particularly, LINOCS is a *learning procedure* to improve learning of
> existing dynamical models by integrating re-weighted lookahead optimization. Conversely, Koopman operators and some other suggested methods are *models* of dynamical systems, which like LDS, SLDS, dLDS, etc. mathematically
> formulate assumptions about the temporal evolution of a system that can be fitted to data. Thus, LINOCS **complements** these models and can improve their fit to data. Applying LINOCS to additional models beyond those presented in the paper indeed represents an exciting direction for future work.

---

> > ### Author Response · Authors · 2024-08-05
> > **Response number 2 to Reviewer MKNA**
> >
> > 5. **The reviewer noted that in Sec 2.1, we mentioned '...observation space, essentially enforcing the transition to the latent space to be the identity operator.' This is the first instance where we discuss observation and latent spaces; hence, he/she  suggested we should expand on what these terms mean.**
> >
> > In the context where we introduced the SLDS and dLDS models, we mentioned the latent versus
> > observation space to highlight that these models (SLDS and dLDS) traditionally include a step of moving to a latent
> > space—a step we do not include in the current version of LINOCS. This detail is not the main
> > focus of our paper but serves to clarify the application of LINOCS to these models. Fixing
> > the observation matrix to the identity operator allows us to learn the dynamics directly in the
> > observation space.
> >
> > While this is not the primary focus of LINOCS, we understand the potential
> > confusion and have added a supplementary section to explain this concept (Appendix C.4 in the
> > updated manuscript).
> >
> > 6) **The reviewers asked for clarification whether Figure 1 is a cartoon:**
> > This is indeed an illustration (as mentioned in the caption, i.e. not real results). We will change
> > the word ''llustration'' to ``scheme'' for clarity. We generated it to exemplify the problem and how LINOCS
> > addresses it.
> >
> > 7) **The reviewer asked about the sentence ‘This approach yields not only a more accurate full-lookahead
> > post-learning reconstruction but also operators that are more closely aligned with the ground truth”.**
> > The claim addresses two main points discussed in the paper: 1) the importance of considering
> > lookahead prediction capabilities when evaluating a dynamical systems model, with the integration
> > of LINOCS improving this aspect, and 2) accurate lookahead predictions also suggest more precise
> > operators, as demonstrated throughout the paper.
> >
> > Thus, integrating LINOCS proves beneficial for
> > ``more accurate full-lookahead post-learning reconstruction'' across various systems, and it facilitates
> > the recovery of more accurate operators (e.g., Figures 3 and 4). We hope this explanation is helpful;
> > otherwise, please let us know if there are any additional specific concerns.
> >
> > 8) **The reviewer pointed out that the text 'Specifically, it gradually increases the weight of the best lookahead reconstruction until convergence conditions are satisfied.' requires clarification. Additionally, they requested the inclusion of an algorithm to aid with these descriptions.**
> >
> > We agree that this point can be clarified. In the original paper, we provided more details about the
> > parameters and weight choice for each experiment in the supplementary Tables, which could have
> > been more clearly referenced the main paper.
> >
> > The overall options for the weights we focused on in
> > this paper are described in the bullets that appear just after the above sentence (“Specifically, it
> > gradually increases...”). Particularly, we enabled different options for weight choice and the choice
> > between them is an input to the method. Regarding the suggestion to add algorithms, we agree that including
> > it will be helpful, and hence we have **added algorithms to all experiments** to ensure that the processes are
> > clear.
> >
> > 9) **The reviewer raised some confusions regarding the illustration in Fig 1.**
> >
> > Figure 1 illustrates the
> > current modeling gap and our approach. It demonstrates how 1-step predictions under some arbitrary
> > dynamical systems identification method can appear accurate while long-term predictions diverge.
> > “Iter” refers to the iteration number. $x_1$ and $x_2$ are the state components to be consistent with the
> > remainder of the paper.
> >
> > To ensure clarity, we have also applied the following changes to the figure in the updated PDF:
> > 1. Clarified “iter” == iterations and added a colorbar for the iteration number.
> > 2. Clarified x = [x1; x2].
> > 3. Improved the aesthetics of the legend in subplot C.
> > 4. Enhanced the caption of subplot C with additional details.
> >
> > 10) **The reviewer asked, with respect to Eq 9, when do we update $c$ vs $f$ vs $w$, and suggested to add an
> > algorithm.**
> > Thank you for raising this point, we added an algorithm to all experiments, including dLDS (Alg. 3). LINOCS, just as in the original
> > presentation of dLDS in Mudrik et al. (2024), iteratively updates the dynamic operators and their
> > time coefficients. Specifically, in each iteration, we first infer the coefficients and then update the
> > dynamic operators.

---

> > > ### Author Response · Authors · 2024-08-05
> > > **Response number 3 to Reviewer MKNA**
> > >
> > > 11) **The reviewer was concerned that the Figures are too small, as figures 4, 7, 11, 12.**
> > >
> > > Thank you for bringing this to our attention. We have made the following changes:
> > > * In Fig. 4 we increased the text size in the heatmap annotations to improve readability.
> > > * In Fig. 7A (top): The 1-step prediction performance across methods is indeed very similar,
> > > and it is not expected for the reader to observe meaningful differences at the 1-step prediction
> > > level. The seemingly identical accuracy in 1-step prediction is central to the figure’s purpose,
> > > which aims to illustrate that despite seemingly accurate 1-step predictions, significant differences
> > > exist in the methods’ abilities to fully reconstruct the system’s dynamics. Figure 7 aims to
> > > accomplish this by pairing the 1-step prediction plots with full lookahead prediction plots to
> > > contrast similar short term accuracy with vastly different long-term accuracy.
> > > * In Fig. 11 we reorganized the figure to increase the size of the subplots, and updated all figure
> > > references.
> > > * In Fig. 12 we were unsure if there was a particular subplot that was concerning to the reviewer. The figure is already enlarged to the largest possible size that can fit on a single page
> > > without the need to split it to 2 figures.
> > >
> > >
> > > 12) **The reviewer was concerned about the following sentence from Sec 4.3 “This underscores the necessity
> > > of multi-step reconstructions for accurately estimating dynamical system dynamics”**.
> > >
> > > We agree that the term “necessary” may be too strong. We revised the
> > > sentence to: “This implies that multi-step reconstructions might be important for more accurately
> > > estimating dynamical system dynamics, as errors could be obscured when assessing only the single-
> > > step reconstruction.”
> > >
> > > 13) **The reviewer recommended listing the LTV constraints we mentioned in Sec 4.4.**
> > >
> > > We agree and have now listed all regularizers (see updated paper).
> > >
> > > 14) **Sec 4.5, the reviewer asked to provide a little more description of the task for the monkeys, and asked about the timing of the task and the data.**
> > >
> > > We have included a more detailed description of the task and the pre-processing applied.
> > > While we agree that a more thorough analysis of the scientific merits would be interesting, our goal in this paper was
> > > to demonstrate that LINOCS can learn operators that achieve better long-term prediction
> > > on real neural data that is high-dimensional, noisy, and nonstationary. For the demonstration, we
> > > randomly chose a random session of the data and trained LINOCS on it.
> > >
> > > We have also added more
> > > details about the data in Sec. C.5 of the updated paper.
> > >
> > > 15) **Regarding Sec 4.5, the reviewer was concerned about the claim “..this suggests that LINOCS may better capture the linear approximation of brain activity
> > > than the common 1-step optimization.” and asked about comparison to neural results.**
> > >
> > > We recognize the importance of comparing neural results to neuroscience methods rather than dynamical systems methods.
> > > However, the
> > > primary goal of presenting LINOCS with real data in our study was to illustrate its ability to
> > > address real-world challenges (e.g. noise, non-stationarity, etc.), not to extract deeper scientific meaning at this point.
> > > While we
> > > agree that it would be extremely interesting to perform such comparisons, we believe this is beyond
> > > the scope of our current paper.
> > > Regarding the big claim, we agree that we should softer it, and
> > > rephrased it to “This suggests that LINOCS may provide a more nuanced linear approximation
> > > of brain activity compared to 1-step optimization, though further comparisons with
> > > established neuroscience methods are needed to fully validate this advantage”.
> > >
> > >
> > > 16) **With respect to Fig 8A, the reviewer suggested to make it clear in the text that each node is a neuron, and asked how the region to region connectivity work.**
> > >
> > > Each node represent a neural unit from the spike sorting results (ideally a neuron). Colors of nodes represent
> > > their estimated brain region.
> > > The region-to-region connectivity displayed in Fig.
> > > 8 is achieved by
> > > running LINOCS on the mean activity per region. We added clarification of both of the above points
> > > in the updated paper. As for normalization, in Appendix C.6 of the updated PDF, we added explanation of the pre-processing we applied.
> > >
> > > 17) **The reviewer asked about the statement "LINOCS not only achieves more precise full lookahead reconstruction compared to baseline methods
> > > but also successfully retrieves ground truth operators in synthetic data...".**
> > >
> > >
> > > Thank you for highlighting this.
> > > We do demonstrate in the results section (e.g., Fig.
> > > 4e, Fig.
> > > 3 ABC) that LINOCS recovers more accurate operators compared to other methods.
> > > However,
> > > to avoid confusion, we will revise the wording of this statement to: “LINOCS achieves more precise full lookahead
> > > reconstruction and more accurately retrieves ground truth operators in synthetic data compared to
> > > baseline methods.”.

---

> > > > ### Author Response · Authors · 2024-08-06
> > > > **Response number 4 to Reviewer MKNA**
> > > >
> > > > 18) **The reviewer was concerned about the following future work idea "[LINOCS]...may help mitigate
> > > > issues such as vanishing or exploding gradients".**
> > > >
> > > > This is mentioned as part of future work, as we believe that extending LINOCS ideas can be helpful
> > > > to better capture dynamics also in other settings beyond these presented in the paper. We rephrased
> > > > that sentence to clarify the above.
> > > >
> > > > 19) **The reviewer asked about the number of rings in Figures 16,17,19.**
> > > >
> > > > Regarding the number of rings, we updated the plot in the updated PDF to include a marker of the
> > > > most active sub-dynamic in each time point. This will highlight the presence of the less apparent
> > > > small-radius circles (e.g.
> > > > in Fig. 16, the circle that goes towards the y-direction, which is less
> > > > noticeable due to its closer-to-0 initial conditions). With respect to testing rotations with increasing
> > > > overlap levels, this could be an exciting exploration to analyze the robustness of the dLDS model against increasing operator overlap.  However, it is beyond the scope of this paper.
> > > >
> > > > 20) **With respect to Fig 20, the reviewer asked about the prediction and training order.**
> > > >
> > > > This is a full lookahead prediction (i.e. prediction order =  T), with a training order of K_train = 19. Thank you for paying attention, we clarified it in the updated paper.
> > > >
> > > > 21) **The reviewer asked about how one chooses a weight function.**
> > > >
> > > > The choice of weight functions in our framework can be guided by both data-related constraints
> > > > (e.g., non-stationarity, noise level) and computational factors (e.g., desired computation time). For
> > > > example, a model demanding more computational power might use higher-order weight function (i.e.,
> > > > higher K) to achieve faster performance, while scenarios with limited computational resources that
> > > > can accommodate longer computation times may use lower-order (i.e., lower K) weight functions,
> > > > provided they result in more equations than variables.
> > > >
> > > > 22) **The reviewer asked about how one chooses which model to choose, and what happens if you choose a more general model on a simpler case.**
> > > >
> > > > LINOCS is a *learning procedure* designed to enhance inference in existing models, and we demonstrate
> > > >  its applicability through several models in the paper. The choice between these models is
> > > > not related to LINOCS but pertains more generally to the scientific goals of the data analysis. For
> > > > instance, a linear model would be suitable for simple systems we assume are stationary. An SLDS
> > > > would be preferable for scenarios where the dynamics are assumed to change abruptly at specific
> > > > times. dLDS is ideal for capturing multiple, simultaneous underlying processes. Finally, a locally Linear
> > > > Time Varying model would be appropriate for tracking a slowly changing dynamic process.
> > > >
> > > > It is thus
> > > > up to the annalist to select the appropriate model, and outside the scope of this work which focuses
> > > > on the learning procedure.
> > > >
> > > > As for the question of what would happen if a more general method
> > > > is applied to a simpler case, the outcomes can vary. For example, if one chooses an SLDS for a
> > > > system that is in practice linear, they might find a system with a single active operator and no
> > > > transitions. Similarly, applying the LTV model to a system with multiple co-occurring processes
> > > > may be effective, but using dLDS could additionally reveal the latent co-active processes underlying
> > > > the system.
> > > >
> > > > 23) **The reviewer asked which model should be used in the neuroscience applications.**
> > > >
> > > > We demonstrated LINOCS on neural data to demonstrate that it can cope with real
> > > > world data challenges. While a full scientific analysis of the dataset is beyond the scope of this work, we do
> > > > note that each model reveals different nuances in the data. For instance, the linear dynamics model
> > > > captures simple background behavior, while dLDS distinguishes different co-occuring neural processes.
> > > >
> > > >
> > > > 24) **The reviewer asked, for the SLDS like methods and applications, how one chooses the number of modes.**
> > > >
> > > > The number of discrete states in SLDS will indeed affect the results, but it is important to note
> > > > that this is a property of the model itself, not the learning procedure (e.g., LINOCS). Related
> > > > work on each method addresses such questions, for example, in SLDS/rSLDS a non-parametric
> > > > stick breaking method can be used (as in Linderman et al. (2016)), and in dLDS, regularization penalizes
> > > > extraneous dynamical systems (Mudrik et al. (2024)). We added a note in the manuscript to refer to
> > > > each model’s original formulation for parameter selection (see under the section “Specific models considered in
> > > > this work” in the updated PDF).

---

> > > > > ### Author Response · Authors · 2024-08-06
> > > > > **Response number 5 to Reviewer MKNA**
> > > > >
> > > > > 25) **The reviewer asked how one chooses hyperparameters in general. For example, how should one choose K?**
> > > > >
> > > > > We would like to distinguish between two classes of parameters:
> > > > > * learning parameters.
> > > > > * model parameters.
> > > > >
> > > > > The learning parameters are related to the learning procedure and include, e.g., the
> > > > > weight function and the look-ahead training order ($K$). For how to choose $K$, please refer to our former answer from bullet 21.
> > > > >
> > > > > For model parameters, these are dependent on the selected model, rather than the learning procedure (e.g., LINOCS), and so users should
> > > > > refer to guidelines given by the model developers, e.g.
> > > > > the number of states in rSLDS and dLDS
> > > > > should follow the explanations in (Linderman et al., 2016; Mudrik et al., 2024).
> > > > >
> > > > >
> > > > > ## Ideas raised to strengthen work
> > > > > 1)  **The reviewer was concerned that the word “often” is over-used.**
> > > > >
> > > > > Thank you for the recommendation, we have replaced some instances of the word “often” by “frequently”, “regularly”, “commonly”, “typically”, or “usually”.
> > > > >
> > > > > 2) **Intro, the reviewer asked about what other ways exist with respect to "...in cases where the dynamics are non-stationary, non-linear, or otherwise constrained in ways that reflect real-world system behavior."**
> > > > >
> > > > > Some other constraints that may arise in modeling include dynamics that are constrained by e.g.,
> > > > > temporal smoothness, sparsity, some other matrix regularizations.
> > > > >
> > > > > 3) **Intro, the reviewer asked to clarify compared to what "...achieving significantly improved accuracy...", and asked to reference later sections.**
> > > > >
> > > > > Thank you for the suggestion, we will add “compared to 1-step optimization” and reference the relevant sections.
> > > > >
> > > > > 4) **The reviewer asked about the use of the terms 'forcing term' vs. 'offset' and inquired about the choice to include an offset compared to normalizing the data.**
> > > > >
> > > > > The decision to normalize data or include an offset term ($b$) is based on the model’s structural
> > > > > requirements, not on the operator learning procedure that LINOCS addresses. We agree, however,
> > > > > that a potential demonstration of LINOCS could involve first normalizing the data and then applying
> > > > > the linear LINOCS without an offset term. The current code implementation supports this option,
> > > > > making it a matter of preprocessing choice.
> > > > >
> > > > > Regarding the terms “offset” and “forcing term”’, they have similar mathematical effects and are often used interchangeably, depending on the terminology
> > > > > preferred in the literature.
> > > > > In this work, we have used “offset”, “bias”, and “forcing term” interchangeably to refer to a constant component added to the system. Both terms relate to how inputs affect system dynamics. However, for consistency in our updated paper,
> > > > > we have changed to “offset” only.
> > > > >
> > > > >
> > > > > 5) **The reviewer asked, with respect to Eq. 6, if the weights are normalized.**
> > > > >
> > > > > In Equation 6, the normalization of
> > > > > $w_k$
> > > > > does not influence the optimization of
> > > > > $A_t$,
> > > > > as the objective
> > > > > is to minimize the weighted sum.
> > > > > As constant multipliers do not change the location of function
> > > > > minimum, it makes no difference whether
> > > > > $w_k$
> > > > > is normalized or not.
> > > > >
> > > > >
> > > > > 6) **The reviewer asked, about Fig 2 right panel, if the coefficients have to be roughly constant or can fluctuate**.
> > > > >
> > > > > In dLDS, the coefficients do not need to be constant and can also include negative values. The only
> > > > > constraint is the sparsity of the coefficients at any time-point. Please refer to Mudrik et al. (2024)
> > > > > for more details about dLDS.
> > > > >
> > > > >
> > > > > 7) **The reviewer asked about a future direction statement "...however LINOCS can be easily adjusted to other noise statistics."**
> > > > >
> > > > > In this paper, we implicitly assumed Gaussian statistics by utilizing the $\ell_2$
> > > > > or Frobenius norm of
> > > > > the differences to minimize the loss, which is derived from the negative log-likelihood of Gaussian
> > > > > distributions.
> > > > > However, considering Poisson statistics would require different cost functions that
> > > > > LINOCS could potentially integrate, and hence it is mentioned as future work.
> > > > >
> > > > >
> > > > > 8) **The reviewer asked about the Eq. in Sec. 3.1, and if $A$ is raised to an exponent.**
> > > > >
> > > > > Thank you for raising this point, we clarified this in the updated paper. It is indeed an exponent.
> > > > >
> > > > > 9) **The reviewer asked for clarification about the SSM framework mentioned in Sec 3.2.**
> > > > >
> > > > > We added clarification in the updated paper that the SSM framework is the coding framework proposed by Lin-
> > > > > derman et al. (2020) for training rSLDS and other State Space Models.

---

> ### Author Response · Authors · 2024-08-06
> **Response number 6 to Reviewer MKNA**
>
> 10) **The reviewer asked about the process of training LINOCS on the SLDS model, and how we deal with
> scenarios where the lookahead order extends the boundary.**
>
> In their original implementation, rSLDS
> iteratively alternates between finding switch times and updating the operators, (see Section 4 in the updated PDF). In
> our implementation of LINOCS to the switching model, we only modified the dynamics update step
> which updates the operators within each linear period. We have now clarified this in Algorithm 2
> in the updated PDF. We treat the end of each period as if it were the end of the measurements,
> limiting the order at each starting point as to not exceed the current period and not to combine
> periods. While considering measurements from post or prior periods can be an exciting extension
> for future work, we found that the current approach works well without such additional complexity.
>
> 11) **The reviewer asked about the interpretation of the 3 rightmost panels of Fig 3D right 3 panels.**
>
> Whether the reconstructed system diverges away from the real system or converges to zero depends
> on the differences in the eigenvalues of the identified transition matrix
> $A$
> compared to the eigenvalues
> of the actual system.
>
> Specifically, if the real part of the leading eigenvalue of the identified
> $A$
> is
> greater than that of the real
> $A$, the reconstruction will diverge.
> Conversely, if it is smaller, the
> system will converge to zero. In our baselines, based on the random seed of the initialization and the
> specific inaccuracies in
> $A$, the system may either diverge or converge to zero. We have now updated
> the baselines to a different random seed where the reconstruction converges to zero, away from the
> real system’s dynamics.
>
>
> 12) **With respect to Fig 3E, the reviewer asked what is theoretically best considering the level of noise.**
>
> We expect LINOCS to perform better than other methods, especially under increased noise levels.
> Since higher noise levels introduce uncertainties in the reconstruction process, using more approximations of different orders will help mitigate the effect of individual sample noise, with increased
> training orders leading to more robust LINOCS results, although this comes at the expense of greater
> computational complexity. We anticipate that the 1-step optimization will be the most sensitive to
> noise. In comparison, DAD models should perform slightly better, with their performance depending
> on whether the noise distribution is more uniform at the beginning or increases towards the end of
> the time series. LINOCS is expected to be more robust to noise, and this robustness will improve
> with increased training order.
>
> 13) **The reviewer asked about Sec 4.1, and if 100 iterations are enough for DAD to converge.**
>
> We have added a graph of the error over training iterations for the three DAD models (see Fig. 24 in
> the updated version). As shown, the DAD algorithm does not converge properly, i.e., it is not a mere
> function of iteration number.
> This is possibly due to its inability to consider multiple prediction
> orders simultaneously in order to “close” the dynamics gap gradually.
>
> 14) **Comment to the comment that the statement: "...instead decay to zero (away from the real system)." is
> known in DMD (Dawson 2016).**
>
> We agree and do not pretend to claim that it is something unexpected, but we rather highlight
> the result to build intuition in the LDS case before progressing to the later SLDS and dLDS cases
> (which to the best of our knowledge are not solved with OPT-DMD).
>
>
> 15) **The reviewer asked about the inherent time scales of the rotations mentioned in Sec 4.1.**
>
> This is synthetic data, so the time points are in arbitrary units. However, for better clarity, we now specified how many complete
> rotations these time points correspond to in the plots.
> Please see below and also in the updated paper.
>
> * For the white noise linear experiment: The angle between each pair of consecutive points is∼14.3deg. and hence 35,000
> points corresponds to around ∼1380 circles.
> * For structured noise: The angle between each consecutive points is ∼22deg. Hence, 70,000 samples
> is ∼4280 circles.
> * For cylinder: The angle between each consecutive points (across the y-z axes) is ∼13.7deg. Hence,
> 70,000 samples is ∼2680
> circles.
>
> 16) **The reviewer suggested to include some experiments in the appendix and asked about larger linear systems.**
>
> Thank you for the suggestion, we believe that discussing these examples in the main text builds
> key intuition for the capabilitiesof LINOCS. Thank you for suggesting the use of a larger system.
> We have already included three synthetic linear demonstrations, which we believe comprehensively
> cover synthetic linear systems. To demonstrate our method’s effectiveness in higher-dimensional settings, we
> therefore included the example with the real-world high-dimensional neural data that highlights LINOCS’ applicability and
> effectiveness in more complex high-dimensional scenarios.

---

> > ### Author Response · Authors · 2024-08-06
> > **Response number 7 to Reviewer MKNA**
> >
> > 17) **The reviewer asked whether training SLDS and rSLDS for a longer period would allow them to find the right transitions.**
> >
> > We believe the effectiveness of further training depends on the nature of the noise in the system,
> > This sets a limit on the minimal error that can be achieved using 1-step optimization methods, especially within a finite timeframe. However, this limitation can be overcome by considering multiple
> > predictions, as LINOCS does.
> >
> > 18) **The reviewer asked about the importance of eigenvalues as mentioned at the end of Section 4.2.**
> >
> > While the eigenspectrum of the operators indeed captures their accuracy, we chose to analyze the
> > eigenvalues in Section 4.2 to better understand the differences between operators, which are challenging to quantify based solely on the operators’ entries. This analysis is not the main focus of the
> > paper but serves as an extended exploration that we found relevant for demonstrating the results of
> > the SLDS experiments.
> >
> > 19) **The reviewer suggested rewording this statement from Sec 4.3 "Training with orders...resulted in high-quality full reconstruction".**
> >
> > Thank you for your feedback. The phrase ’high-quality full reconstruction’ was intended to convey
> > the effectiveness of our approach with training orders greater than 35 on our synthetic dataset. To
> > avoid any misunderstanding, we have revised the wording to “highly accurate full reconstruction”.
> >
> > 20) **The reviewer asked about the training and prediction order in Fig. 20.**
> >
> > This figure aims
> > to show the evolution of performance over training iterations, rather than presenting new results
> > about LINOCS’ capabilities. The training order was
> > $K = 19$
> > , and we added this information to the
> > text in the updated version.
> >
> > 21) **The reviewer asked, in Fig 6H, which prediction order is used and about the effect of the switching times.**
> >
> > Figure 6H pertains to dLDS (decomposed), which,
> > unlike SLDS (switching), does not involve switching times, hence there are no switches to predict.
> > The training order used in this analysis was detailed in Table 5. Yet, we acknowledge the importance
> > of making this information more readily accessible and have now included the training order directly
> > in the main text.
> >
> > 22) **The reviewer asked why in Fig 6A, K pred of 50 look worse than of 800 or 1000.**
> >
> >
> > We would like to draw the reviewer’s attention to the axes of the subplots. Although we agree that
> > the structure at
> > $K_{pred} = 50$
> > may appear visually worse compared to
> > $K_{pred} = 800$ or $K_{pred}= 2000$
> > ,
> > the error, measured by the
> > $\ell_2$
> > norm, is actually lower for
> > $K_{pred} = 50$.
> > For instance, at
> > $K_{pred}= 800$,
> > the z-axis reaches much higher values (
> > ∼40) than the ground truth, indicating a larger deviation.
> > Additionally, both
> > $K_{pred} = 800$ and $K_{pred} = 2000$
> > exhibit a more convex structure than the ground
> > truth, which further contributes to the increased
> > $\ell_2$
> > error compared to $K_{pred} = 50$.
> >
> >
> > 23) **With respect to Fig 8C, the reviewer asked why the 1-step approach fails to capture the average activity as well as about data normalization**
> >
> > Figure 8C follows the point wedemonstrate in the synthetic linear experiment, that even very simple
> > linear systems may diverge in the long run when in the 1-step optimization approach.The reason
> > we see the activity decay in this figure, which shows the full lookahead activity, is that even a
> > slight deviation in the operators’ values can result in the decay.Regarding normalization, we have
> > added information about the pre-processing in Section C.6. We indeed normalized the data and we
> > recognize that this may affect the results. However, as demonstrated throughout the paper, LINOCS
> > is robust beyond a specific choice of normalization.
> >
> >
> > 24) **The reivewer asked, regarding Fig 9C (now 10C), if SLDS does well during parts of the time period.**
> >
> > Yes, you are correct; SLDS does perform well during certain time periods. However, in dynamical
> > modeling, we generally aim for a model that maintains consistent performance throughout the
> > duration.
> > Notably, in the three rightmost subplots (blue, cyan, pink regions), SLDS diverges in
> > multi-step predictions. To compare with LINOCS, please compare the solid curves (LINOCS) to the
> > dashed lines (1-step) during these periods.

---

> > > ### Author Response · Authors · 2024-08-06
> > > **Response number 8  to Reviewer MKNA**
> > >
> > > 25) **The reviewer asked, regarding Fig 11, what new information subplot D is telling us.**
> > >
> > > Subplot D in Figure 11 examines the differences in the dynamic operators
> > > $A$
> > > across learning methods, unlike the other subplots that focus on the learned trajectories $X$.
> > > This subplot aims to determine the similarity and dis-similarity between operators identified by different methods,
> > > including LINOCS. In other subplots (e.g., B, E), we observe that LINOCS provides better lookahead
> > > predictions. Unfortunately, since this is real data, and as is often the case with real-world data, we do not have ground truth
> > > for the operators to compare against. Therefore, it becomes particularly interesting to examine how
> > > similar the operators identified by LINOCS are to those identified by other methods and to each
> > > other. This subplot is thus helpful as it allows us to gauge what specifically LINOCS captures that
> > > enables it to predict further into the future more effectively than other methods.
> > >
> > > 26) **The reviewer was concerned about the clarity of supplementary figure 18 (of the original submission).**
> > >
> > > This figure was intended as a follow-up analysis to the dLDS results in Figure 5 that aimed to
> > > demonstrate how the structure of the leading eigenvalues improves with training order. Given that
> > > Figure 5 provides a clearer exploration, we now adhered to that figure, and removed Figure 18 in the updated PDF.
> > >
> > > 27) **The reviewer asked about what Fig 22 tell us in terms of the differences between classical SLDS and LINOCS-
> > > SLDS.**
> > >
> > > This supplementary figure explores the SLDS results in real-world applications. We were interested
> > > in determining whether there are fundamental differences in the switching times or patterns between
> > > LINOCS and the 1-step approach, following our findings with synthetic data (where we saw clear
> > > differences).
> > > It appears that there is no consistent difference in the number of switches between
> > > LINOCS and 1-step SLDS. LINOCS does not consistently find fewer or more switches in all cases,
> > > suggesting that the differences between the methods are not solely due to the number of switches.
> > > Since this is real data and we do not know the ’real’ operators, we cannot determine the correct
> > > switching pattern, however.
> > >
> > >
> > > 28) **The reviewer asked about why LINOCS captures Fig. 4 better than classical-SLDS.**
> > >
> > > We believe that since SLDS and rSLDS are optimized mainly based on 1-step, recovering accurate
> > > boundary transitions can be challenging due to: 1) the added noise, 2) similarities between operators
> > > that make them harder to distinguish, and 3) a limited number of iterations (equivalent to those
> > > enabled with LINOCS, allowing for comparison).
> > > As seen in Figure 4, SLDS 10 captures some
> > > periods correctly, such as the pink long period at the end.
> > > Meanwhile, rSLDS 10 and rSLDS 100
> > > capture trends correctly but show frequent transitions or "jumps" between operators within these
> > > periods. LINOCS overcomes these challenges by considering multiple orders simultaneously, which
> > > helps reduce noise and improve boundary transition accuracy.
> > >
> > >
> > > 29) **The reviewer asked what would happen for larger systems.**
> > >
> > > Indeed, our demonstration with synthetic data focused on simpler
> > > cases to clearly illustrate the concept. This is why we also provided examples using real-world, high-
> > > dimensional neural data from regions of varying dimensions to showcase its applicability in more
> > > complex scenarios.done
> > >
> > >
> > > 30) **The reviewer emphasized the preferred usage of the word 'switching' over 'switched' (section 2.1 in the original submission).**
> > >
> > > Thank you for pointing this out. While indeed some works
> > > (e.g., Linderman et al. (2016); Nassar et al. (2018)) used the term “Switching” for this model, other
> > > works Berger et al. (2022); Klamka and Niezabitowski (2013); Diepold and Eid (2011) described this
> > > context using the word “Switched”. Since in that Section we are discussing the general model rather
> > > than a specific implementation, we believe that both “Switched” and “Switching” should be valid.
> > > However, for consistency, we agree with the reviewer and have updated the paper to use only “Switching” throughout.
> > >
> > > [1]  Scott W Linderman, et al. Recurrent switching linear dynamical systems. arXiv preprint arXiv:1610.08466, 2016.
> > >
> > > [2] Josue Nassar, et al. Tree-structured recurrent
> > > switching linear dynamical systems for multi-scale modeling. arXiv preprint arXiv:1811.12386, 2018
> > >
> > > [3] Guillaume Berger, et al. An algorithm for learning switched linear dynamics from data. Advances in Neural Information
> > > Processing Systems, 35:30419–30431, 2022.
> > >
> > > [4]  Jerzy Klamka and Michał Niezabitowski. Controllability of switched linear dynamical systems. In 2013
> > > 18th International Conference on Methods & Models in Automation & Robotics (MMAR), pages 464–467.
> > > IEEE, 2013
> > >
> > > [5] Klaus J Diepold and Rudy Eid. Guard-based model order reduction for switched linear systems. In Methoden
> > > und Anwendungen der Regelung-stechnik, Erlangen-Münchener Workshops 2009 und 2010, pages 67–78,
> > > 2011.

---

> > > > ### Author Response · Authors · 2024-08-06
> > > > **Response number 9 to Reviewer MKNA**
> > > >
> > > > 31) **The reviewer suggests that HMM should be explained.**
> > > >
> > > >
> > > > We have added a mention of Hidden Markov
> > > > Models (HMMs) in the ’Prior Relevant Approaches’ section of the updated paper.
> > > >
> > > > 32) **The reviewer suggested referencing associated figures at the beginning of sections 3.1, 3.2, etc.**
> > > >
> > > > This has been addressed in the updated PDF.
> > > >
> > > > 33) **The reviewer gave feedback to improve several aspects of Figure 3.**
> > > >
> > > > Thank you, we have fixed all these points in the updated PDF.
> > > >
> > > > 34) **With respect to the equation in Section 3.4, the reviewer suggested writing the regularization term more explicitly.**
> > > >
> > > > Importantly, we wanted to keep
> > > > the regularization as a user choice, and hence presented different options in the section opening.
> > > > However, we understand the reviewer's point so further specified it in the updated version.
> > > >
> > > > 35) **The reviewer suggested labeling $\sigma$ more explicitly in the text.**
> > > >
> > > > We agree and clarified it in the update PDF.
> > > >
> > > > 36) **The reviewer was concerned that we did not present the random offset in supplementary Figure 16 in the original paper.**
> > > >
> > > > We added a
> > > > heatmap of the offset to the figure (subplot D).
> > > >
> > > >
> > > > 37) **The reviewer asked about the comparison in supplementary Figure 18**
> > > >
> > > > We would like to emphasize that
> > > > $K_{train} = 1$
> > > > is equivalent to
> > > > $1−step$
> > > > optimization and is the
> > > > traditional (non-LINOCS) dLDS. We have now further emphasized this in the text.
> > > >
> > > > 38) **The reviewer suggested clarifying that $\rho$ refers to correlation in Fig. 4.**
> > > >
> > > > We thank the reviewer for the suggestion, and clarified this in the figure caption.
> > > >
> > > > 39) **Comments about colors and comparison in Fig 5**
> > > >
> > > > A) The goal of this figure is to compare different training orders of LINOCS. They are all
> > > > using LINOCS, except for
> > > > $K_{train} = 1$, which by definition, is non-LINOCS.
> > > > B) The goal of this figure is to explore the effect of different training orders on full lookahead
> > > > prediction performance.
> > > > Particularly, this figure comes to emphasize the gap between 1-step and
> > > > full lookahead
> > > > prediction
> > > > and how LINOCS can help mitigate this. Hence, the colormaps used in
> > > > former figures (e.g. blue for LINOCS and orange for 1-step) will not be very relevant here.
> > > >
> > > >
> > > > 40) **The reviewer asked about the meaning of $\lambda$ and suggested using log-scale for some figures.**
> > > >
> > > > We thank the reviewer for the comment. We added a clarification about what is
> > > > $\lambda$
> > > > in the LTV description. Particularly, it is the weight of
> > > > the regularization term (please refer to Eq. 13 in the new PDF). We also further clarified this in the caption now.
> > > >
> > > > As for Fig. 7A, the differences between the 1-step prediction results are indeed very small across the models, and
> > > > this is partially what the figures aims to convey. The emphasis should be on the difference between
> > > > the multi-step ability (bottom of 7A), not the 1-step (where results may look similar despite some being much better
> > > > full-dynamic predictors than other).
> > > >
> > > > 41) **The reviewer suggested adding units for the time in Figure 8 (Figure 9 in the update PDF).**
> > > >
> > > > We are thankful for the suggestion, we now added the time units to that figure and other figures.
> > > >
> > > > 42) **The reviewer suggested using a log scale in Fig. 9C and asked to replace the term 'full curve' with 'solid curve.'**
> > > >
> > > > We have replaced 'full' with 'solid.' Regarding the scale, we believe the figure's emphasis is on showing that the existing SLDS diverge while LINOCS does not. Therefore, small nuanced differences in the exact values are less relevant, as the diverging trend is clearly visible.

---

> > > > > ### Author Response · Authors · 2024-08-06
> > > > > **Response number 10 to Reviewer MKNA**
> > > > >
> > > > > 43) **The reviewer suggested updates to Fig 10.**
> > > > > We added clearer time units to the Figure, clarified $\lambda$ in the caption, and separated the real and imaginary parts of the eigenvalues to different subplots.
> > > > > Regarding the colors, the challenge here involves the use of two different color bars. We acknowledge that the colors of
> > > > > the coefficients are close to each other, but they are distinguishable. It was more important for us
> > > > > to use more distinct colors for the areas in the network-style subplot A, as it would be harder to
> > > > > distinguish them there.
> > > > >
> > > > >
> > > > > 44) **The reviewer gave comments on figure 11 (now 12).**
> > > > >
> > > > > Thank you for the comments, we now adhered to using $\lambda$ only as the regularization term.
> > > > > Regarding the suggestion to fix the color scale across plots,
> > > > > although it is standard practice to
> > > > > maintain consistent vmin and vmax across heatmaps for uniform data presentation, this is less ideal
> > > > > in this specific figure, as the color scale varies a lot across subplots (especially in the last columns). Hence,
> > > > > fixing the scale would obscure details in the heatmaps with smaller ranges.
> > > > >
> > > > > 45) **The reviewer suggested that appendix B is redundant.**
> > > > >
> > > > > We agree, and removed it in the updated PDF.
> > > > >
> > > > > 46) **Suggestions about figures 12 & 13 (now 14)**.
> > > > >
> > > > > We thank the reviewer for their ideas, and we applied most of them (labels, titles, scale, numbers to fit in boxes, etc).  Regarding the suggestion to re-organize the figure, while we generally agree that it is ideal to order figures consistently, for Figure 12, we aimed to
> > > > > maximize the coverage of space within the figure. These different demonstrations (compared to Fig. 4) required varying
> > > > > sizes for the subplots which resulted in the different, optimized organization.
> > > > >
> > > > >
> > > > > 47) **Comments to figure 16 (now 18)**
> > > > >
> > > > > We now added $b$, fixed axes in subplot C, fixed cutoff, showed temporal evolution in subplot C.
> > > > >
> > > > >
> > > > > 48) **The reviewer asked about the ground truth in Fig 20 (now 21)**
> > > > >
> > > > > We thank the reviewer for pointing out this visualization issue. The Ground Truth does exist but is obscured
> > > > > by the LINOCS results and excessive transparency (e.g., see subplot of iteration 7).
> > > > > Specifically, in subplots 1-5, where the range of divergence in the predictions is much larger, the Ground Truth
> > > > >  have z values much smaller than those presented in the top subplots and are thus hard to
> > > > > notice. Similarly, in subplots 6-10, the LINOCS reconstruction is very close to the Ground Truth,
> > > > > making it also difficult to discern.
> > > > >
> > > > > Hence, we re-created this figure (Figure 20 originally, now 21) with the following adjustments for clarity:
> > > > >
> > > > > 1. Clarified training order.
> > > > > 2. Made sure that the LINOCS results are more distinguishable from the ground truth by changing
> > > > > the colors.
> > > > > 3. Add the ground truth as an example in its own subplot.
> > > > >
> > > > > 49) **The reviewer had questions regarding SLDS labeling and white space in Fig 22 (now 23).**
> > > > >
> > > > > In the context of '1-step' optimization, we are referring to the learning procedure for the dynamics update step,
> > > > > not the model itself.
> > > > > In that figure, we compared the original SLDS learning update rule (which
> > > > > optimizes based on 1-step optimization) against the  multi-step (LINOCS-driven) SLDS. Hence, we used “1-step” as
> > > > > the label for classical SLDS in that figure.
> > > > > However, we understand the confusion and updated the label in the updated version.
> > > > >
> > > > > Regarding the white space, we are unsure exactly which white space the reviewer is referring to. The
> > > > > white space around each sub-box represents a neuron, and we need to maintain the same size for
> > > > > each neuron representation, regardless of the number of neurons in different areas.

---

> > > > > > ### Comment · Reviewer_MKNA · 2024-09-02
> > > > > >
> > > > > > I thank the authors for their detailed responses and edits. The changes (i.e. additional literature, algorithms, and updated figures) have improved the manuscript immensely

---

### Review · Reviewer_nQ8x · 2024-07-08

**Summary Of Contributions:**

In this work the authors introduce LINCOS, a multi-step ahead training procedure for training dynamical systems. While in principle LINCOS could be applied to any arbitrary dynamical system, the authors focus on linear dynamical systems, switching linear dynamical systems and time-varying linear dynamical systems.

**Audience:**

Yes

**Claims And Evidence:**

Yes

**Requested Changes:**

1. Discussing similarities and differences between LINCOS and previous IMS training procedures. [Critical]
2. Including comparisons to previous IMS training procedures. [Critical]

**Strengths And Weaknesses:**

# Strengths

In general, I think the paper is well-written and very easy to follow. Moreover, I'm *very* impressed the breadth of the experimental section which I think speaks to the broad applicability of LINCOS.

# Weaknesses

I think the two biggest weaknesses is the novelty and positioning of the paper; I will first focus on the novelty aspect. Training dynamical systems---and state-space models in general---using Iterative Multi-Step Prediction (IMS) is not novel and has been done before. For instance, in [1] IMS was used to train deterministic dynamical systems while in [2] and [3] IMS was used to train state-space models. Thus, it is imperative that the authors discuss the differences between the LINCOS and the studies I have linked as I think that there is substantial overlap between them. Moreover, I think it is imperative that the some/all of the methods should be compared against LINCOS in the experiments.

Next, I think the positioning of the paper needs to be revamped. LINCOS is a general learning algorithm that can be applied to arbitrary dynamical systems, though the authors focus on linear dynamical systems and its extensions. This is important to note because the authors state in the introduction that "LINOCS also avoids relying on massive amounts of data (like RNNs, for example, require)", where the massive amount of data is really a comment about the parameterization of the dynamical system (RNN) and NOT about the learning algorithm used. For instance, LINCOS can be applied to RNNs but still might require a massive amount of data due to the sheer number of parameters in the RNN. I am also making this point as methods proposed in [2] and [3] can also be used for linear dynamical systems as they are, just like LINCOS, learning algorithms that can be applied to arbitrary dynamical systems. Thus, what needs to be made clear in the paper is what is LINCOS solving that the previous papers have not? For instance, it might be the case that previous approaches don't empirically perform well on linear dynamical systems and that is what LINCOS solves. Or it might be the case that adaptive weighting in LINCOS allows for more stable training compared to previous works.


# References
[1] *Learning Dynamical Systems from Noisy Sensor Measurements using Multiple Shooting*. Armand Jordana, Justin Carpentier, Ludovic Righetti. https://arxiv.org/abs/2106.11712

[2] *Learning Latent Dynamics for Planning from Pixels*. Learning Latent Dynamics for Planning from Pixels.* Danijar Hafner, Timothy Lillicrap, Ian Fischer, Ruben Villegas, David Ha, Honglak Lee, James Davidson. https://arxiv.org/abs/1811.04551

[3] *RNN with Particle Flow for Probabilistic Spatio-temporal Forecasting.* Soumyasundar Pal, Liheng Ma, Yingxue Zhang, Mark Coates. https://arxiv.org/abs/2106.06064

---

> ### Author Response · Authors · 2024-08-06
> **Response to reviewer nQ8x**
>
> We acknowledge the reviewer’s feedback and their recognition of the broadness and depth of the experiments. We would like to address the reviewers’ concerns as follows:
>
> 1. We believe there are three unique properties of our paper:
>
> * Due to its broadness, LINOCS can be applied to more specialized systems like Switching Systems and Decomposed Systems, as we have demonstrated. We will ensure that this is emphasized in the paper.
>
> * LINOCS introduces adaptive weights for multiple starting and ending points, which are not present in other methods. This is advantageous in cases where the starting or adjustment points are very noisy. For example, the work by Jordana et al. (2021) that the reviewer mentioned uses multiple shooting with multi-step prediction. Although their paper introduces multiple starting points, if one of the shooting nodes is very noisy, the entire prediction will be biased since the same noisy node is used for the prediction period. LINOCS’s adaptive weights consider multiple starting points and find the optimal representation, reducing the impact of noise.
>
> * Most other methods rely on RNN architectures, which, while advantageous when fitting RNNs, introduce additional challenges like gradient descent-related issues (e.g., sensitivity to step size) and complex parameterization, making them difficult to train. Our method is not specific to RNNs and is therefore relevant to hierarchical models that represent more structured and nuanced dynamic interactions, such as dLDS and SLDS.
>
> 2. Changes we applied to the paper:
>
> * We clarified LINOCS’s advantages over existing methods at the end of the Introduction.
>
> * We added more comparisons of LINOCS to multi-step approaches. Specifically, we benchmarked the LTV-LINOCS to two additional multi-step methods [1,2] under various hyperparameters (figure 8 in updated PDF).
>
> [1] Florian Hess, et al. "Generalized teacher forcing for learning chaotic dynamics". ICML 2023.
>
> [2] Armand Jordana, et al. "Learning dynamical systems from noisy sensor measurements using multiple shooting". arXiv preprint arXiv:2106.11712, 2021.
>
>
> * In the updated PDF, we now further mention and discuss several other multi-step methods in section 3 ("Prior relevant approaches"), including those the reviewer mentioned and additional ones.

---

### Comment · Action_Editor_N7sM · 2024-08-15
**Reminder to reviewers**

Hi reviewers,

Thank you for your patience. Now that the authors have replied to your reviews, please read their responses and let them know if you have any remaining questions. If you do not, please update/add your official recommendation.

Best,

AE

---

### Decision · Action_Editor_N7sM · 2024-09-08

**Recommendation:** Accept as is

**Comment:**

All reviewers were initially positive about the submission, but had questions of clarity on how LINOCS was different from other methods and how specific technical details were conveyed. The authors provided a detailed and thorough rebuttal and revision, greatly enhancing the quality of their original submission. Due to this lengthy set of revisions, and due to the fact that the reviewers seemed satisfied with them, I am recommending accept as is.

Note that one of the three reviewers was not able to change their original recommendation, which was made prematurely before the authors provided their responses. However, given that both of the other two reviewers recommend acceptance, and because I believe the authors have sufficiently addressed the comments of that specific reviewer, I am making my decision now (so as to not unnecessarily prolong the process).

**Audience:**

The growing integration of the computational neuroscience, dynamical systems, and machine learning communities (which this work sits at the intersection of) makes TMLR an ideal journal to publish this work, as it attracts readers from all three areas.

**Claims And Evidence:**

All reviewers agree that the claims and evidence presented in the paper are supported. Additionally, all reviewers agreed that the use of the proposed method (LINOCS) on real-world data was a real strength of the paper. The lengthy revisions and rebuttal provided by the authors further solidifies the paper's clarity and impact.